# Local to Global: Learning Dynamics and Effect of Initialization for Transformers

**Ashok Vardhan Makkuva** *
EPFL

**Marco Bondaschi** *
EPFL

**Chanakya Ekbote**
EPFL

**Adway Girish**
EPFL

**Alliot Nagle**
UT Austin

**Hyeji Kim**
UT Austin

**Michael Gastpar**
EPFL

## Abstract

In recent years, transformer-based models have revolutionized deep learning, particularly in sequence modeling. To better understand this phenomenon, there is a growing interest in using Markov input processes to study transformers. However, our current understanding in this regard remains limited with many fundamental questions about how transformers learn Markov chains still unanswered. In this paper, we address this by focusing on first-order Markov chains and single-layer transformers, providing a comprehensive characterization of the learning dynamics in this context. Specifically, we prove that transformer parameters trained on next-token prediction loss can either converge to global or local minima, contingent on the initialization and the Markovian data properties, and we characterize the precise conditions under which this occurs. To the best of our knowledge, this is the first result of its kind highlighting the role of initialization. We further demonstrate that our theoretical findings are corroborated by empirical evidence. Based on these insights, we provide guidelines for the initialization of single-layer transformers and demonstrate their effectiveness. Finally, we outline several open problems in this arena. Code is available at: https://github.com/Bond1995/Markov.

## 1 Introduction

Transformers have been at the forefront of recent successes across various fields including natural language processing [34]. To obtain insights into their impressive sequential modeling capabilities, a notable emerging theme among several recent works is to model the input data as a Markov process.

Using this Markovian perspective, works such as [24, 14, 8], among others, study the in-context learning capabilities of transformer. [23] analyzes the loss-landscape for the next-token prediction task, while [18] shows an equivalence between the attention mechanism and Markov models. Although these works reveal interesting insights about transformers and their capabilities, many fundamental questions about their learning dynamics remain unanswered. In particular, a comprehensive characterization of their training dynamics vis-á-vis the data distributional properties and the role of initialization is still missing.

To address this gap, in this paper, we focus on the canonical setting of first-order Markov chains and single-layer transformers and analyze the learning dynamics in this context. Specifically, we prove (Thms. 2, 3, and 8) that the input data properties and the parameter initialization play a significant role in the convergence of the transformer parameters to either local or global minima on the loss surface. Further, we precisely characterize (Figs. 1 and Fig. 2) the specific data characteristics and the region of initialization under which this convergence occurs. Based on these insights, we provide guidelines

---

*Equal contribution. Correspondence to ashok.makkuva@epfl.ch.

38th Conference on Neural Information Processing Systems (NeurIPS 2024).

for the initialization of transformer parameters and empirically corroborate our theoretical findings. On the theoretical front, our analysis provides a novel gradient flow analysis of the transformer parameters, capitalizing on their low-rank structure during training. Our main contributions can be summarized as follows:

- **Theoretical analysis:** We precisely characterize the loss landscape and gradient flow dynamics for single-layer transformers with first-order Markov chains (Secs. 3 and 4). We demonstrate that transformer parameters trained on next-token prediction loss can converge to global or local minima, depending on the initialization and the Markovian data properties, and determine the exact conditions under which this occurs (Thms. 2, 3, and 8). To the best of our knowledge, this is the first result of its kind.

- **Insights into initialization:** Our theoretical analysis underscores the crucial role of initialization in transformer parameter training. Specifically, we demonstrate how the standard Gaussian initialization scheme can lead the convergence to local or global minima depending on the Markovian data properties (Thms. 2 and 8, Figs. 1 and 2).

- **Guidelines:** Based on these insights, we provide practical guidelines for parameter initialization, corroborated by empirical evidence demonstrating their effectiveness (Sec. 5.2).

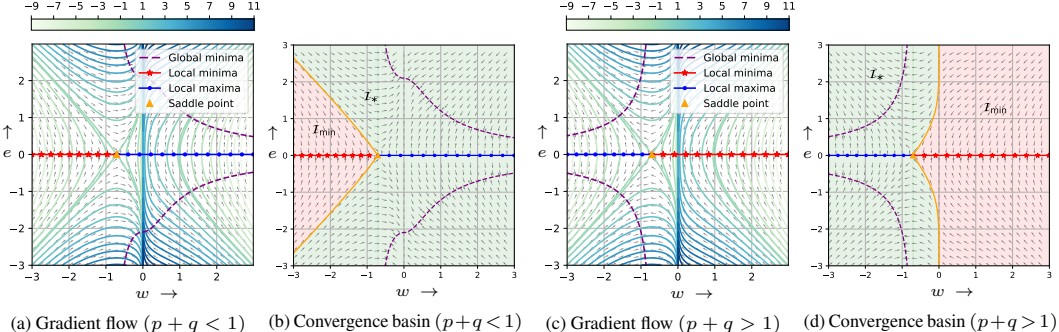

(a) Gradient flow $(p + q < 1)$    (b) Convergence basin $(p+q<1)$    (c) Gradient flow $(p + q > 1)$    (d) Convergence basin $(p+q>1)$

Figure 1: Gradient flow dynamics and initialization effect for single-layer transformers. $(p, q)$ are Markov switching probabilities, and $(e, w)$ are the embedding and weight parameters (Sec. 2). (a), (c): The flow is aligned along energy contour lines, converging to local or global optima. (b), (d): $\mathcal{I}_\star$ is the basin of convergence for global minima, $\mathcal{I}_{\min}$ for the local minima, and yellow asymptotes for the saddle point. Notice the contrasting behavior for Gaussian initialization around origin for $p + q \lessgtr 1$.

**Notation.** We denote scalars by italic lower case letters like $x, y$ and Euclidean vectors and matrices in bold: $\boldsymbol{x}, \boldsymbol{y}, \boldsymbol{M}$, etc. $\|\cdot\|$ denotes the $\ell_2$-norm for Euclidean vectors and Frobenius norm for matrices. $[k] \triangleq \{1, \ldots, k\}$, and for a sequence $(x_n)_{n \geq 1}$, define $x_k^m \triangleq (x_k, \ldots, x_m)$ if $k \geq 1$ and $(x_1, \ldots, x_m)$ otherwise. For $z \in \mathbb{R}$, the sigmoid $\sigma(z) \triangleq 1/(1+e^{-z})$, $\mathrm{ReLU}(z) \triangleq \max(0, z)$ and the convex logistic loss $\ell_{\log}(z) \triangleq \log(1 + \exp(-z)) \in (0, \infty)$. For events $A$ and $B$, $\mathbb{P}(A)$ denotes the probability of $A$ whereas $\mathbb{P}(A \mid B)$ the conditional probability. Let $(x, y)$ be a pair of discrete random variables on $[k] \times [k]$ with the probability mass function (pmf) of $x$ being $\boldsymbol{p}_x = (p_1, \ldots, p_k) \in [0, 1]^k$. Then its Shannon entropy is defined as $H(x) = H(\boldsymbol{p}_x) \triangleq -\sum_{i \in [k]} p_i \log p_i$. The conditional entropy is defined to be $H(y|x) \triangleq H(x, y) - H(x)$. The entropy rate of a stochastic process $(x_n)_{n \geq 1}$ is defined as $\lim_{n \to \infty} H(x_1^n)/n$. We simply write $x = y$ to mean $\mathbb{P}(x = y) = 1$. We also use the shorthand $\mathbb{P}(y = j \mid x)$ for $\mathbb{P}(y = j \mid x = x)$ as a function of the random variable $x$. For $p \in (0, 1)$, the binary entropy function $h(\cdot)$ is defined as $h(p) \triangleq -p \log p - (1 - p) \log(1 - p)$.

## 2 Problem Setting

We formally define the problem setting for analysis of single-layer transformers with Markovian data, following [23].

**Input data.** We assume that the input word sequence $\{x_n\}_{n=1}^N \in \{0, 1\}^N$ is a first-order time-homogenous Markov chain with a fixed kernel $\boldsymbol{P} = (\boldsymbol{P}_{ij})$. That is, the transition probability

$P_{ij} \triangleq \mathbb{P}\left(x_{n+1} = j \mid x_n = i\right) = \mathbb{P}\left(x_{n+1} = j \mid x_n = i, x_1^{n-1}\right)$, for any $x_1^{n-1}, i, j \in \{0, 1\}$. In particular, we consider $\boldsymbol{P} = [1-p, p; q, 1-q]$ where $p = \boldsymbol{P}_{01} = \mathbb{P}\left(x_{n+1} = 1 \mid x_n = 0\right)$ and $q = \boldsymbol{P}_{10} = \mathbb{P}\left(x_{n+1} = 0 \mid x_n = 1\right)$ denote the switching probabilities from the states 0 and 1 respectively. We call $p + q$, the *switching factor*. We assume that the process is already mixed, i.e. $x_n \sim \boldsymbol{\pi}$ for all $n$, where $\boldsymbol{\pi} \triangleq (\pi_0, \pi_1) = (q, p)/(p+q)$ is the stationary distribution satisfying $\boldsymbol{\pi} = \boldsymbol{\pi P}$. Succinctly, $(x_n)_{n \geq 1} \sim (\boldsymbol{\pi}, \boldsymbol{P})$. For this process, the entropy rate, $H(x_{n+1}|x_n) = \frac{1}{p+q}\left(q\,h(p) + p\,h(q)\right)$, and the entropy of the marginal, $H(x_n) = H(\boldsymbol{\pi})$, are both constant in $n$.

**Transformer architecture.** We consider a single-layer transformer with a single-head attention and ReLU non-linearity. Given an input sequence $\{x_n\}_{n=1}^N$, it performs the following mathematical operations at each $n \in [N]$ to predict the next-token probability $f_{\boldsymbol{\theta}}(x_1^n)$:

$$x_n \in \{0, 1\} \xrightarrow{\text{Embedding}} \boldsymbol{x}_n \xrightarrow{\text{Attention}} \boldsymbol{y}_n \xrightarrow{\text{Feed-forward}} \boldsymbol{z}_n \xrightarrow{\text{Linear}} \text{logit}_n \xrightarrow{\text{Prediction}} f_{\boldsymbol{\theta}}(x_1^n),$$

where

$$\boldsymbol{x}_n = x_n \boldsymbol{e} + \boldsymbol{p}_n \in \mathbb{R}^d, \tag{Embedding}$$

$$\boldsymbol{y}_n = \boldsymbol{x}_n + \sum_{i \in [n]} \underbrace{\text{att}_{n,i}}_{\in (0,1)} \cdot \boldsymbol{W}_V \boldsymbol{x}_i \in \mathbb{R}^d, \tag{Attention}$$

$$\boldsymbol{z}_n = \boldsymbol{y}_n + \boldsymbol{W}_2 \text{ReLU}(\boldsymbol{W}_1 \boldsymbol{y}_n) \in \mathbb{R}^d, \tag{Feed-forward}$$

$$\text{logit}_n = \langle \boldsymbol{a}, \boldsymbol{z}_n \rangle + b \qquad \in \mathbb{R}, \tag{Linear}$$

$$f_{\boldsymbol{\theta}}(x_1^n) \triangleq \mathbb{P}_{\boldsymbol{\theta}}\left(x_{n+1} = 1 \mid x_1^n\right) = \sigma(\text{logit}_n) \in [0, 1]. \tag{Prediction}$$

Here $\boldsymbol{\theta} \triangleq (\boldsymbol{e}, \{\boldsymbol{p}_n\}_{n=1}^N, \dots, \boldsymbol{W}_1, \boldsymbol{W}_2, b, \boldsymbol{a}) \in \mathbb{R}^D$ denotes the full list of the transformer parameters from the embedding layer till the linear layer (§ A details them). While the underlying data $\{x_n\}_{n=1}^N$ is Markovian, i.e. $\mathbb{P}\left(x_{n+1} = 1 \mid x_1^n\right) = \mathbb{P}\left(x_{n+1} = 1 \mid x_n\right)$, the transformer is agnostic to this fact and it can potentially utilize the full past $x_1^n$ in the Attention layer, via the attention weights $\text{att}_{n,i}$, to predict the next-symbol probability $f_{\boldsymbol{\theta}}(x_1^n) = \mathbb{P}_{\boldsymbol{\theta}}\left(x_{n+1} = 1 \mid x_1^n\right)$. Note that it suffices to estimate the symbol 1 probability as the vocabulary is binary. We also refer to the above architecture as "full model".

**Loss and training.** The transformer parameters $\boldsymbol{\theta}$ are usually initialized according to standard Gaussian distribution $\mathcal{N}(0, \sigma^2 \boldsymbol{I})$ with a small variance $\sigma^2$ [26] and are trained using gradient-based methods to minimize the cross-entropy loss on the next-token prediction, i.e.

$$\min_{\boldsymbol{\theta}} L(\boldsymbol{\theta}), \quad L(\boldsymbol{\theta}) \triangleq -\frac{1}{N} \sum_{n \in [N]} \mathbb{E}_{x_1^{n+1}}[x_{n+1} \cdot \log f_{\boldsymbol{\theta}}(x_1^n) + (1 - x_{n+1}) \cdot \log(1 - f_{\boldsymbol{\theta}}(x_1^n))]. \tag{1}$$

When the input sequence $\{x_n\}_{n=1}^N \sim (\boldsymbol{\pi}, \boldsymbol{P})$, the minimal loss equals its entropy-rate, i.e. $L_\star \triangleq \min_{\boldsymbol{\theta}} L(\boldsymbol{\theta}) = H(x_{n+1}|x_n)$ [11].

**Loss landscape.** A key surprising observation in [23] is that the loss function $L(\cdot)$ admits both the global and local minima depending on the switching factor $p + q$ of the Markovian data, and the weight-tying of the embedding and linear weights ($\boldsymbol{e} = \boldsymbol{a}$) of the transformer. In particular, they show that

(i) for all $(p, q) \in (0, 1)^2$, there exists a global minimum $\boldsymbol{\theta}_\star$ for the loss $L$ such that its prediction matches the Markov kernel, i.e. $\mathbb{P}_{\boldsymbol{\theta}_\star}\left(x_{n+1} = 1 \mid x_1^n\right) = \mathbb{P}\left(x_{n+1} = 1 \mid x_n\right)$.

(ii) if $p + q > 1$ and the weights are tied ($\boldsymbol{e} = \boldsymbol{a}$), there exists a bad local minimum $\boldsymbol{\theta}_{\min}$ for $L$ whose prediction equals the marginal, i.e. $\mathbb{P}_{\boldsymbol{\theta}_\pi}\left(x_{n+1} = 1 \mid x_1^n\right) = \mathbb{P}\left(x_{n+1} = 1\right)$.

In view of these results, we focus on the weight-tying scenario and hence let $\boldsymbol{e} = \boldsymbol{a}$ to be a single parameter in $\mathbb{R}^d$. Thus, $\boldsymbol{\theta} = (\boldsymbol{e} = \boldsymbol{a}, \{\boldsymbol{p}_n\}_{n=1}^N, \dots, \boldsymbol{W}_1, \boldsymbol{W}_2, b)$. We interchangeably refer to $\boldsymbol{\theta}$ as both the transformer and the set of parameters.

**Our objective.** While the aforementioned results detail the static landscape of the loss, they do not characterize the learning dynamics on the loss surface and the effect of initialization, which plays a central role in training machine learning models [5]. In view of these shortcomings, the main objective of this paper is to address the following question:

> *(Q.1): Can we explain how the initialization and learning dynamics affect the convergence of the transformer parameters $\boldsymbol{\theta}$ to the local or global optima?*

# 3 Canonical Low-rank Parameterization

**Motivation.** Given the complexity of the transformer architecture and the non-convex loss function, it is challenging to analyze the learning dynamics directly [24, 14]. To tackle this, we capitalize on the following empirical observation [23] which is the motivating idea behind our approach: when trained by gradient-based methods, the weight matrices $(\boldsymbol{W}_V, \ldots, \boldsymbol{W}_1, \boldsymbol{W}_2)$ at the optima $\boldsymbol{\theta}_\star$ and $\boldsymbol{\theta}_{\min}$ exhibit *rank-one* structure, whose eigenvector is the same direction in which the both the token embedding $\boldsymbol{e}$ and the positional embeddings $\boldsymbol{p}_n$ are all aligned in. Interestingly, such low-rank solutions can also be shown to be theoretically optimal (§ B details these structures). While these observations illustrate the implicit bias towards low-rank solutions at the final convergence, a natural question arises: *if we initialize with low-rank parameters, will they remain low-rank during training?* In Sec. 5.1, we affirmatively address this based on a thorough empirical evaluation for single-layer transformers and inspired by these empirical phenomena, without loss of generality, we restrict our attention to these low-rank manifolds to characterize the learning dynamics. This is similar in spirit to [24], where they assume special attention matrix structure for learning induction heads.

**Parameterization.** More specifically, we consider a special low-rank parameterization that is empirically observed and capitalize on it to address *(Q.1)*. Interestingly, along this low-rank manifold, it suffices to consider a reduced set of parameters $\boldsymbol{\theta} \in \mathbb{R}^2$ or $\boldsymbol{\theta} \in \mathbb{R}^3$ given by:

$$\boldsymbol{\theta} = (e, w) \in \mathbb{R}^2, \text{ or } \boldsymbol{\theta} = (e, w, a) \in \mathbb{R}^3. \qquad \text{(Reparameterization)}$$

Here $e$ denotes the *embedding* scalar, $w$ the *weight*, and $a$ the *attention* parameter respectively. Now we describe the parameterization of the transformer vis-á-vis these scalars and refer to § C for a more detailed descripton. Let the input $\{x_n\}_{n=1}^N$ be a first-order Markov chain as in Sec. 2 and let $n \in [N]$ be fixed. Then we have

$$\text{Embedding} : \boldsymbol{e} = e \cdot \boldsymbol{\alpha}, \, \boldsymbol{p}_n = \left(-\frac{e}{2}\right) \cdot \boldsymbol{\alpha} \to \boldsymbol{x}_n = e\left(x_n - \frac{1}{2}\right)\boldsymbol{\alpha}, \quad e \in \mathbb{R}, \boldsymbol{\alpha} \in \{\pm 1\}^d/\sqrt{d},$$

$$\text{Attention} : \boldsymbol{W}_V = \boldsymbol{\alpha}\,\boldsymbol{v}^\top \to \boldsymbol{y}_n = e\left(x_n - \frac{1}{2}\right)\boldsymbol{\alpha} + \underbrace{\langle \boldsymbol{v}, \boldsymbol{\alpha}\rangle}_{\propto a \approx 0}\left(\sum_{i \in [n]} \text{att}_{n,i} \cdot e\left(x_i - \frac{1}{2}\right)\right)\boldsymbol{\alpha}, \boldsymbol{\alpha}, \boldsymbol{v} \in \mathbb{R}^d.$$

The scalar $a$ is the product of $\langle \boldsymbol{v}, \boldsymbol{\alpha}\rangle$ and the scaling in the attention weights $\text{att}_{n,i}$ (Eq. (39)), which is empirically close to zero for first-order Markov chains. Hence for the ease of exposition, we first let $a = 0$ and analyze the general case when $a \in \mathbb{R}$ in Sec. 4.1. We continue:

$$\text{Feed-forward} : \boldsymbol{W}_1 = \frac{|w|}{\sqrt{d}}\mathbf{1}\,\boldsymbol{\alpha}^\top, \boldsymbol{W}_2 = \frac{w}{\sqrt{d}}\boldsymbol{\alpha}\,\mathbf{1}^\top \to \boldsymbol{z}_n = e\left(x_n - \frac{1}{2}\right)(1 + 4w|w|x_n)\boldsymbol{\alpha}, w \in \mathbb{R}.$$

$\mathbf{1}$ is the all-one vector in $\mathbb{R}^r$ with $r = 4d$ typically in practice. Substituting this $\boldsymbol{z}_n$ in the linear layer with $\boldsymbol{e} = \boldsymbol{a}$ and bias $b \in \mathbb{R}$, the logits and the probabilities simplify to:

$$\text{Linear} : \text{logit}_n(e, w, b) = e^2(1 + 2w|w|)\, x_n + b - \frac{e^2}{2} \in \mathbb{R}, \qquad (2)$$

$$\text{Prediction} : f_{(\boldsymbol{\theta}, b)}(x_1^n) = \sigma\left(\text{logit}_n\right) \in (0, 1), \quad \boldsymbol{\theta} \triangleq (e, w). \qquad (3)$$

Finally, using the equivalence between the cross-entropy loss and the logistic loss $\ell_{\log}(\cdot)$, the loss function in Eq. (1) can be compactly written as (Lemma 6):

$$L(\boldsymbol{\theta}, b) = \frac{1}{N}\sum_{n \in [N]}\mathbb{E}[\ell_{\log}\left((2x_{n+1} - 1) \cdot \text{logit}_n(\boldsymbol{\theta})\right)], \quad \boldsymbol{\theta} \in \mathbb{R}^2, b \in \mathbb{R}. \qquad (4)$$

Due to convexity of $\ell_{\log}(\cdot)$, it follows that $L(\boldsymbol{\theta}, b)$ is convex in the bias $b$ for any fixed $\boldsymbol{\theta}$, whose minimizer, $b_\star(\boldsymbol{\theta}) = \text{argmin}_{b \in \mathbb{R}} L(\boldsymbol{\theta}, b)$, has a closed form expression (Lemma 5). Hence, without loss of generality, we consider the loss with this optimal bias $b_\star$:

$$L(\boldsymbol{\theta}) \triangleq L(\boldsymbol{\theta}, b_\star) = \frac{1}{N}\sum_{n \in [N]}\mathbb{E}\left[\ell_{\log}\left((2x_{n+1} - 1)\left(e^2(1 + 2w|w|)\, x_n + b_\star - \frac{e^2}{2}\right)\right)\right]. \qquad (5)$$

Empirically, this roughly translates to running the gradient-based algorithm for the bias for more steps at each $\boldsymbol{\theta}$. In practice, one additional step is usually sufficient (see Sec. 5). Eq. (5) resembles the standard logistic regression loss [31] whose binary labels are $2x_{n+1} - 1 \in \{\pm 1\}$ and the logits given by $e^2(1 + 2w|w|) x_n + b_\star - e^2/2$, for each $n \in [N]$. The key difference here is that the logits are a non-linear function of the parameters $(e, w)$ unlike in the standard setting.

We briefly summarize our assumptions below.

**Assumption 1** (Canonical parameterization). For our theoretical analysis, we assume that the effective transformer parameters are canonically parameterized as $\boldsymbol{\theta} = (e, w, a) \in \mathbb{R}^3$. First we study the scenario when $a = 0$ with $\boldsymbol{\theta} = (e, w)$ and build upon these observations to study the general setting of $\boldsymbol{\theta} = (e, w, a)$ in Sec. 4.1.

### 3.1 Loss Landscape with Canonical Parameterization

With the new set of parameters $\boldsymbol{\theta} = (e, w) \in \mathbb{R}^2$, we are now ready to analyze the loss $L(\cdot)$ in Eq. (5). First we recall the definition of a critical point [20]. A point $\boldsymbol{\theta}_\star \in \mathbb{R}^2$ is a critical or a stationary point for $L$ if $\nabla L(\boldsymbol{\theta}_\star) = 0$. A critical point $\boldsymbol{\theta}_\star$ is a *local minimum* if there exists a neighborhood $U$ around $\boldsymbol{\theta}_\star$ such that $L(\boldsymbol{\theta}_\star) \leq L(\boldsymbol{\theta})$ for all $\boldsymbol{\theta} \in U$, and a *local maximum* if $L(\boldsymbol{\theta}_\star) \geq L(\boldsymbol{\theta})$. If the neighborhood $U$ is whole of $\mathbb{R}^2$, it is a *global minimum/maximum*. On the other hand, a critical point is a saddle point if for all neighborhoods $U$ around $\boldsymbol{\theta}_\star$, there are $\boldsymbol{\theta}_1, \boldsymbol{\theta}_2 \in U$ such that $L(\boldsymbol{\theta}_1) \leq L(\boldsymbol{\theta}_\star) \leq L(\boldsymbol{\theta}_2)$.

Thm. 1 below provides a complete characterization of the loss landscape in terms of the aforementioned critical points.

**Theorem 1** (All critical points). *Let the input sequence be $\{x_n\}_{n=1}^N \sim (\boldsymbol{\pi}, \boldsymbol{P})$, the transformer parameters $\boldsymbol{\theta} = (e, w) \in \mathbb{R}^2$, and the next-token prediction loss $L(\cdot)$ be as in Eq. (5). Then for any $(p, q) \in (0, 1)^2$ with $p + q \neq 1$ and $N \in \mathbb{N}$,*

(i) *the set of all global minima is given by*

$$\boldsymbol{\Theta}_\star(p, q) \triangleq \left\{ (e, w) \in \mathbb{R}^2 : e^2(1 + 2w|w|) = \log \frac{(1 - p)(1 - q)}{pq} \right\}, \tag{6}$$

(ii) *the set of all local minima is given by*

$$\boldsymbol{\Theta}_{\min}(p, q) \triangleq \left\{ (e, w) \in \mathbb{R}^2 : e = 0, (p + q - 1)(1 + 2w|w|) > 0 \right\}, \tag{7}$$

(iii) *the set of all local maxima is given by*

$$\boldsymbol{\Theta}_{\max}(p, q) \triangleq \left\{ (e, w) \in \mathbb{R}^2 : e = 0, (p + q - 1)(1 + 2w|w|) < 0 \right\}, \tag{8}$$

(iv) *and the set of all saddle points is*

$$\boldsymbol{\Theta}_{\mathrm{sad}}(p, q) \triangleq \left\{ (0, -1/\sqrt{2}) \right\}. \tag{9}$$

*Thus the set of all critical points is*

$$\left\{ \boldsymbol{\theta} \in \mathbb{R}^2 : \nabla L(\boldsymbol{\theta}) = 0 \right\} = \boldsymbol{\Theta}_\star \cup \boldsymbol{\Theta}_{\min} \cup \boldsymbol{\Theta}_{\max} \cup \boldsymbol{\Theta}_{\mathrm{sad}}. \tag{10}$$

*In addition, for any $\boldsymbol{\theta}_\star \in \boldsymbol{\Theta}_\star, \boldsymbol{\theta}_{\min} \in \boldsymbol{\Theta}_{\min}, \boldsymbol{\theta}_{\max} \in \boldsymbol{\Theta}_{\max}$, and $\boldsymbol{\theta}_{\mathrm{sad}} \in \boldsymbol{\Theta}_{\mathrm{sad}}$, the loss values satisfy*

$$H(x_{n+1} \mid x_n) = L(\boldsymbol{\theta}_\star) < L(\boldsymbol{\theta}_{\min}) = L(\boldsymbol{\theta}_{\max}) = L(\boldsymbol{\theta}_{\mathrm{sad}}) = H(x_{n+1}).$$

*Proof.* We refer to § E. $\square$

Fig. 1 illustrates the loci of these critical points for $p + q < 1$ and $p + q > 1$. Motivated by empirical observations, while [23] characterizes local minima for $p + q > 1$, it is interesting to note that our Thm. 1 shows that local minima also exist for $p + q < 1$ (Eq. (7) and Fig. 3a). So why did they find the minima only when $p + q > 1$? The answer to this, and more broadly to question *(Q.1)* lies in the learning dynamics and initialization for $\boldsymbol{\theta}$, which we study in the next section.

# 4 Learning Dynamics

Capitalizing on the loss landscape in terms of the critical points in Thm. 1, we now focus on the convergence of gradient-based algorithms to these points. In this regard, we analyze the dynamics of the gradient-flow (GF), which can be viewed as a continuous-time analogue of gradient-descent [6]. The gradient-flow of the parameters, $(\boldsymbol{\theta}_t)_{t\geq 0}$, on $L$ is governed by

$$\frac{\mathrm{d}\boldsymbol{\theta}_t}{\mathrm{d}t} = -\nabla L(\boldsymbol{\theta}_t), \quad \boldsymbol{\theta}_t = (e_t, w_t) \in \mathbb{R}^2, \, t \geq 0, \tag{GF}$$

where $\boldsymbol{\theta}_t \triangleq \boldsymbol{\theta}(t)$ is a $C^1$ (continuously differentiable) curve in $\mathbb{R}^2$ starting with a randomly initalized $\boldsymbol{\theta}_0$. To characterize these trajectories, we define an *energy function* $\mathcal{E}(\cdot, \cdot)$, which plays a crucial in the GF dynamics. It is defined as

$$\mathcal{E}(e, w) \triangleq e^2 - (w^2 + \mathrm{sign}(w) \cdot \log|w|), \quad \forall (e, w) \in \mathbb{R}^2 \setminus \text{e-axis}, \tag{11}$$

where e-axis $\triangleq \{(e, w = 0)\}$. Note that $\mathcal{E}$ is well-defined and finite for all the points in its domain. On the other hand, $\lim_{w\to 0^-} \mathcal{E}(e, w) = -\infty$ whereas $\lim_{w\to 0^+} \mathcal{E}(e, w) = \infty$ for any fixed $e \in \mathbb{R}$. Thus the e-axis corresponding to $w = 0$ serves as an energy barrier for the flow. Figs. 3a and 3b illustrate this by visualizing the energy contour lines. The utility of the energy function is captured in the following lemma.

**Lemma 1** (Constant energy along the flow). *For any $(p, q) \in (0, 1)^2$ and initialization $\boldsymbol{\theta}_0 = (e_0, w_0) \in \mathbb{R}^2$, let $(\boldsymbol{\theta}_t)_{t\geq 0}$ be the corresponding GF trajectory starting from $\boldsymbol{\theta}_0$. If $\boldsymbol{\theta}_0 \in \mathbb{R}^2 \setminus \text{e-axis}$, the energy stays constant along the trajectory, i.e.*

$$\mathcal{E}(\boldsymbol{\theta}_t) = e_t^2 - (w_t^2 + \mathrm{sign}(w_t) \cdot \log|w_t|) = \mathcal{E}(\boldsymbol{\theta}_0), \quad \forall t \geq 0. \tag{12}$$

*On the other hand, if $\boldsymbol{\theta}_0 \in \text{e-axis}$, we have that $\boldsymbol{\theta}_t \in \text{e-axis}$ for all $t \geq 0$ with $w_t = w_0 = 0$, i.e. if we initialize on the e-axis, the trajectory always stays there.*

We are now ready to present the main results of our paper. Specifically, Thm. 2 and Thm. 8 highlight the role of the switching factor of the Markovian data, $p + q$, and the parameter initialization, $\boldsymbol{\theta}_0$, in deciding whether the GF converges to local optima or global optima. First we define the energy value $\mathcal{E}_{\text{sad}} \triangleq \mathcal{E}(e = 0, w = -1/\sqrt{2}) = -(1 + \log 2)/2$.

**Theorem 2** (GF dynamics for $p + q > 1$). *Let $(p, q) \in (0, 1)^2$ with $p + q > 1$, the input sequence be $\{x_n\}_{n=1}^N \sim (\boldsymbol{\pi}, \boldsymbol{P})$, and $(\boldsymbol{\theta}_t)_{t\geq 0}$ be the corresponding GF trajectory starting from $\boldsymbol{\theta}_0$. Then for all initializations $\boldsymbol{\theta}_0 \in \mathbb{R}^2$, the gradient flow converges to a critical point of the loss $L$. That is, there exists a $\boldsymbol{\theta}_{\text{lim}} \in \mathbb{R}^2$ such that $\lim_{t\to\infty} \boldsymbol{\theta}_t = \boldsymbol{\theta}_{\text{lim}}$ and $\nabla L(\boldsymbol{\theta}_{\text{lim}}) = 0$. In particular, $\boldsymbol{\theta}_{\text{lim}}$ is a*

(i) *a local minimum if*

$$\boldsymbol{\theta}_0 \in \mathcal{I}_{\text{min}} \triangleq \left\{ (e, w) : w \in (-1/\sqrt{2}, 0), \, e \in (-g(w), g(w)), \, g(w) = \sqrt{w^2 - \log(-w) + \mathcal{E}_{\text{sad}}} \right\}$$
$$\cup \{ (e, w) : w \geq 0 \},$$

(ii) *a saddle point if $\boldsymbol{\theta}_0 \in \mathcal{I}_{\text{sad}} \triangleq \left\{ (e, w) : w \in [-1/\sqrt{2}, 0), \, e = \pm\sqrt{w^2 - \log(-w) + \mathcal{E}_{\text{sad}}} \right\}$,*

(iii) *a local maximum if $\boldsymbol{\theta}_0 \in \mathcal{I}_{\text{max}} \triangleq \{ (e, w) : e = 0, \, w < -1/\sqrt{2} \}$,*

(iv) *and a global minimum if $\boldsymbol{\theta}_0 \in \mathcal{I}_\star \triangleq \mathbb{R}^2 \setminus (\mathcal{I}_{\text{min}} \cup \mathcal{I}_{\text{sad}} \cup \mathcal{I}_{\text{max}})$.*

*Consequently, when $p + q > 1$, if we use the standard initialization $\boldsymbol{\theta}_0 \sim \mathcal{N}(0, \sigma^2 \boldsymbol{I}_2)$ with $\sigma^2 \ll 1/\sqrt{2}$, $\boldsymbol{\theta}_{\text{lim}}$ will be a local minimum with high probability. If $p + q < 1$, under the same initialization scheme, $\boldsymbol{\theta}_{\text{lim}}$ will be a global minimum with high probability.*

*Proof sketch.* The main idea behind the proof is to show that if we do not initialize on the e-axis, the flows stays on the constant energy contour (Lemma 1) and hence converges to a critical point of the loss $L$, which is at the intersection of the contour line and the set of critical points (Lemmas. 10 and 11). By determining where these intersections occur, the corresponding basins of convergence $\mathcal{I}_{\text{min}}, \ldots, \mathcal{I}_\star$ are obtained by showing that an initialization in a specific set leads to the said critical point (Thm. 1). The proof for $\boldsymbol{\theta}_0 \in \text{e-axis}$ is similar. $\square$

Figs. 1b and 1d illustrate these initialization sets corresponding to the convergence basins for $p = q = 0.9$ and $p = q = 0.1$ respectively. An analogous result about GF dynamics for $p + q < 1$ is presented in Thm. 8 (§ F.2). Here a key difference is that small Gaussian initialization around origin leads to a global minimum $\boldsymbol{\theta}_{\lim}$ with high probability (Fig. 1d).

**Key insights.** Together, Thm. 2 and Thm. 8 address our motivating question *(Q.1)* by fully characterizing the GF dynamics in terms of initialization and input data properties. Specifically, our results explain the phenomenon in [23] wherein they observe local minima for $p + q > 1$ more often than for $p + q < 1$, owing to standard Gaussian initialization around origin (Figs. 1b and 1d). However, in practice, we often do not know the input switching factor, raising a natural questions: *is there a data-agnostic initialization that always converges to global minima?* Indeed, as can be seen from Figs. 1b and 1d, there is a common region of initialization in the negative half-plane above the saddle-asymptotes (in yellow) that leads to the global minima convergence irrespective of the switching $p + q$. Mathematically, this region is given by $\mathcal{I}_{\mathrm{common}} \triangleq \{(e, w) : w < 0, |e| > \sqrt{w^2 - \log(-w) + \mathcal{E}_{\mathrm{sad}}}\}$. We empirically corroborate this fact in Sec. 5.2.

## 4.1 Gradient Flow with Attention

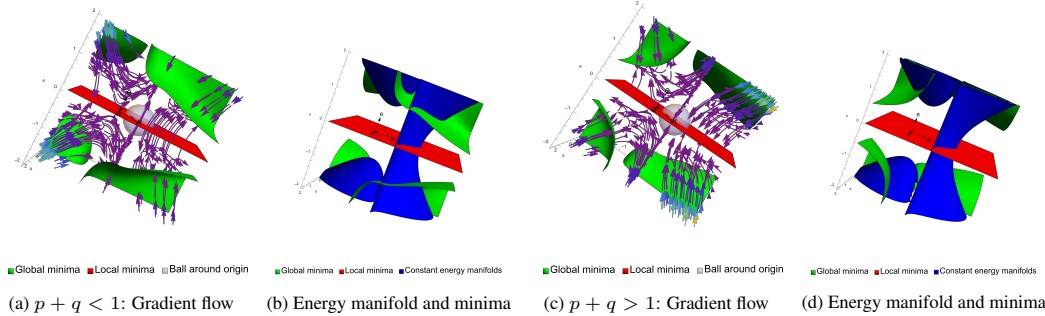

(a) $p + q < 1$: Gradient flow    (b) Energy manifold and minima    (c) $p + q > 1$: Gradient flow    (d) Energy manifold and minima

Figure 2: Gradient flow dynamics for the canonical parameters $\boldsymbol{\theta} = (e, w, a) \in \mathbb{R}^3$ with the attention scalar $a$. Notice the contrasting behavior for Gaussian initialization around origin for $p + q$ smaller and greater than one. For an enhanced view of the flow near the origin, please refer to Fig. 5.

In this section, the consider the attention scalar $a \in \mathbb{R}$ (Sec. 3) and study the gradient flow dynamics with the parameters $\boldsymbol{\theta} = (e, w, a) \in \mathbb{R}^3$. The parameter $a$ captures the overall scaling from the value, key, and query components in the attention layer. Recall that the soft-max attention weights are given by $\mathrm{att}_{n,i} \propto \exp(\langle \boldsymbol{q}_n, \boldsymbol{k}_i \rangle / \sqrt{d})$, where $\boldsymbol{q}_n = \boldsymbol{W}_Q \boldsymbol{x}_n$ and $\boldsymbol{k}_i = \boldsymbol{W}_K \boldsymbol{x}_i$ are the query and key embeddings for any position $i \in [n]$. Using the low-rank structure of the query and key matrices, satisfying $\boldsymbol{W}_Q^\top \boldsymbol{W}_K = (q^2 d) \boldsymbol{\alpha}\boldsymbol{\alpha}^\top$ and the value matrix $\boldsymbol{W}_V = \boldsymbol{\alpha}\boldsymbol{v}^\top$ for some $q \in \mathbb{R}$ and $\boldsymbol{v} \in \mathbb{R}^d$ (§ G), and assuming linear attention $\mathrm{att}_{n,i} \propto \langle \boldsymbol{q}_n, \boldsymbol{k}_i \rangle / \sqrt{d}$, we define a single scalar $a \triangleq \langle \boldsymbol{v}, \boldsymbol{\alpha} \rangle q^2 d^{5/2} / 4$ that captures the essence of the attention layer (Eq. (39)). We note that linear attention weights are a standard assumption in the transformer analysis literature [3, 36]. Using this parameterization, similar to the steps in Sec. 3, we obtain the final loss function to be

$$L(\boldsymbol{\theta}) = \mathbb{E}\left[\ell_{\log}\left((2Y - 1)\left(e^2\left[\left(X - \frac{1}{2}\right)(1 + ae^2)(1 + 2w|w|) + w|w(1 + ae^2)|\right] + b_\star\right)\right)\right],$$

where $\boldsymbol{\theta} = (e, w, a)$ and $b_\star$ is the corresponding optimal bias. $L$ recovers the loss in Eq. (5) when $a = 0$. In Thm. 10, we determine the set of all critical points of $L$ in terms of global minima and local optima in closed-form expressions, analogous to Thm. 1. Capitalizing on this characterization, we now shift our focus to the analysis of the gradient flow in $\mathbb{R}^3$. To this end, let $(\boldsymbol{\theta}_t)_{t \geq 0}$ be a $C^1$ curve in $\mathbb{R}^3$ governed by

$$\frac{\mathrm{d}\boldsymbol{\theta}_t}{\mathrm{d}t} = -\nabla L(\boldsymbol{\theta}_t), \quad \boldsymbol{\theta}_t = (e_t, w_t, a_t) \in \mathbb{R}^3, \ t \geq 0, \qquad \text{(GF-attn)}$$

starting with a randomly initalized $\boldsymbol{\theta}_0$. We define the *energy function* $\mathcal{E}(\cdot, \cdot, \cdot)$ as

$$\mathcal{E}(e, w, a) \triangleq e^2 - (w^2 + \mathrm{sign}(w) \cdot \log|w|) - 2a^2, \quad \forall (e, w, a) \in \mathbb{R}^3 \setminus \text{ea-plane}, \qquad (13)$$

where ea-plane $\triangleq \{(e, w = 0, a)\}$. It is similar to its counterpart in Eq. (11), except for the $2a^2$ term. Fig. 2 visualizes this energy surface and the set of critical points, which reveal close resemblance to that of Fig. 1 in $\mathbb{R}^2$. Capitalizing on the energy function, we now present our main result with the attention.

**Theorem 3** (GF dynamics with attention). *For any* $(p, q) \in (0, 1)^2$ *and initialization* $\boldsymbol{\theta}_0 \in \mathbb{R}^3$, *let* $(\boldsymbol{\theta}_t)_{t \geq 0}$ *be the corresponding GF-attn trajectory starting from it. Then for all* $\boldsymbol{\theta}_0 \in \mathbb{R}^3$, *the gradient flow converges to a critical point of the loss* $L$. *That is, there exists a* $\boldsymbol{\theta}_{\lim} \in \mathbb{R}^3$ *such that* $\lim_{t \to \infty} \boldsymbol{\theta}_t = \boldsymbol{\theta}_{\lim}$ *and* $\nabla L(\boldsymbol{\theta}_{\lim}) = 0$. *Further,*

(i) *if* $\boldsymbol{\theta}_0 \in \mathbb{R}^3 \setminus$ *ea-plane, we have* $\mathcal{E}(\boldsymbol{\theta}_{\lim}) = \mathcal{E}(\boldsymbol{\theta}_t) = \mathcal{E}(\boldsymbol{\theta}_0)$ *for all* $t \geq 0$. *Hence* $\boldsymbol{\theta}_{\lim}$ *is at the intersection of the energy contour line* $\mathcal{E} = \mathcal{E}_0$ *with that of the set of critical points.*

(ii) *if* $\boldsymbol{\theta}_0 \in$ *ea-plane, we have* $\boldsymbol{\theta}_t \in$ *ea-plane for all* $t \geq 0$ *and hence* $\boldsymbol{\theta}_{\lim} \in$ *ea-plane.*

*Proof.* We refer to § G and § N.4. $\qquad\qquad\qquad\qquad\qquad\qquad\qquad\qquad\qquad\qquad\qquad\qquad$ □

Thm. 3 shows that the learning dynamics with attention closely resemble those without it (Thms. 2 and 8). While the set of all critical points of $L$, and thus the limit points of the flow, has a closed-form expression (Thm. 10), deriving the same for the initialization sets $\mathcal{I}_{\min}$ and $\mathcal{I}_\star$ to determine the basin of convergence is technically challenging (see discussion in § G). Nonetheless, empirical observations with the standard Gaussian initialization around origin reveal a similar picture as in the two-dimensional setting for both the $p + q < 1$ and $p + q > 1$ cases (Fig. 2). We believe it's an interesting direction of future research to theoretically characterize this, analogous to Thms. 2 and 8. We refer to § G for additional details and proofs.

## 5 Empirical Results

We empirically validate our canonical parameterization $\boldsymbol{\theta} \in \mathbb{R}^3$ (Sec. 3) by demonstrating full model convergence to low-rank parameters through both qualitative and quantitative evidence. Qualitatively, we visualize weight matrices across iterations; quantitatively, we plot the percentage of energy captured by the top-rank components across iterations. We then demonstrate the generalization of our theoretical findings on local optima and initialization with canonical parameters to the full model $\boldsymbol{\theta} \in \mathbb{R}^D$. We conclude with a discussion on higher-order and multi-state Markov chains.

### 5.1 Low-rank Parameters

**Full model converges to low-rank.** We let the input Markov sequence to be $\{x_n\}_{n=1}^N \sim (\boldsymbol{\pi}(p, q), \boldsymbol{P}(p, q))$ for $p = 0.2, q = 0.3, N = 1024$ and consider the full model as defined in Sec. 2 with embedding dimension $d = 8$. First, we initialize the parameters $\boldsymbol{\theta} = (e = a, \{\boldsymbol{p}_n\}_{n=1}^N, \ldots, \boldsymbol{W}_1, \boldsymbol{W}_2, b)$ using the standard Gaussian initialization with standard deviation 0.001 [26] and train them using SGD on a batch size $B = 16$ and for $t = 800$ iterations. In Fig. 6, we track the value matrix $\boldsymbol{W}_V \in \mathbb{R}^{d \times d}$ and the weight matrix $\boldsymbol{W}_1 \in \mathbb{R}^{4d \times d}$ across iterations. We observe that at convergence both $\boldsymbol{W}_V$ and $\boldsymbol{W}_1$ are approximately rank-one with one of their components being same as the embedding vector (the row in $\boldsymbol{W}_V$ and column in $\boldsymbol{W}_1$). Further, the embedding vector has all entries in $\{\pm 1\}$ up to a scaling. We observe the same conclusion for other weight matrices $\boldsymbol{W}_{K,Q}, \boldsymbol{W}_2$ and for all values of $(p, q) \in (0, 1)^2$. Fig. 3 also quantitatively demonstrates this.

**Full model initialized at low-rank remains low-rank during training.** Inpsired by the low-rank structure obtained above, we randomly initialize the weight parameters as rank-one matrices and the embeddings on the hypercube $\{\pm 1\}^d$. After the initialization, we train them without any low-rank restrictions, and track them during the course of training. Interestingly, here we observe that the parameters still stay low-rank as illustrated in Fig. 7 and Fig. 3. A similar conclusion holds for the remaining weight matrices. Together these results provide the empirical basis for our canonical parameterization analysis in Sec. 3.

### 5.2 Effect of Initialization: Broader Implications

Now we investigate the findings of Sec. 3 and Sec. 4, derived for the canonical low-rank model, more broadly in the context of full model in Sec. 2. In particular, as shown in Thm. 2 and Fig. 1d

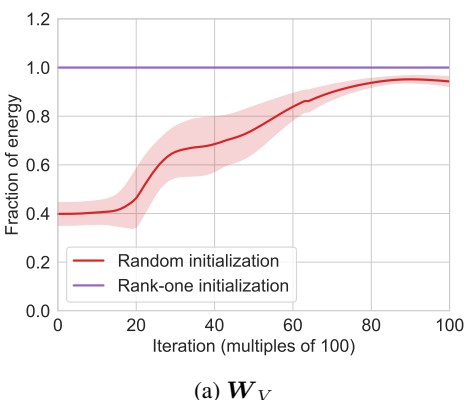

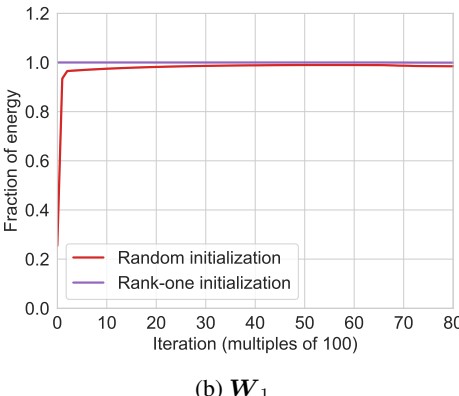

(a) $\boldsymbol{W}_V$                              (b) $\boldsymbol{W}_1$

Figure 3: Convergence to rank one parameters: percentage of energy contained in the first rank component of the weight matrices $\boldsymbol{W}_1$ and $\boldsymbol{W}_V$ across iterations. The percentage is computed as $\frac{\sigma_1^2}{\sum_i \sigma_i^2}$, where the $\sigma_i$'s are the singular values of the matrices in descending order.

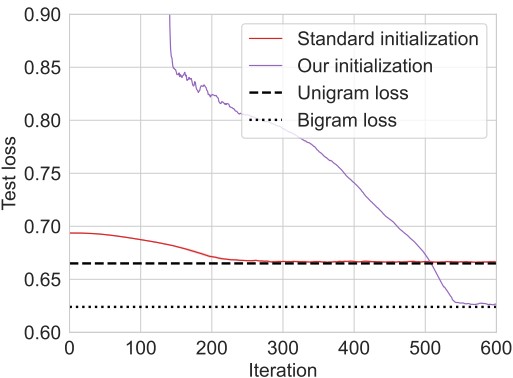

Figure 4: Comparison between the average loss curve for the standard gaussian initialization around 0 and our initialization, for $p = 0.5$ and $q = 0.8$. Starting from the standard initialization, the model converges to a local minimum corresponding to the unigram model. With our initialization, it converges to the global minimum corresponding to the bigram model.

for $p + q > 1$, any small initialization around zero would lead a local minima convergence. To test this hypothesis, we compare the standard initialization where all the transformer parameters $\boldsymbol{\theta} = (\boldsymbol{e} = \boldsymbol{a}, \{\boldsymbol{p}_n\}_{n=1}^N, \ldots, \boldsymbol{W}_1, \boldsymbol{W}_2, b)$ are randomly chosen around zero with small variance $\sigma = 0.02$, with a new initialization based on our results, where we initialize the embedding vector $\boldsymbol{e}$ such that all cordinates are equal to $e = 0.5$, $\boldsymbol{W}_1$ to be constant with the scalar $w_1 = 1$ and $\boldsymbol{W}_2$ constant with $w_2 = -1$ (corresponding to $\mathcal{I}_\star$ in Fig. 1d). We indeed observe that the final test loss matches the unigram loss for the standard initialization, while it converges to the optimal bigram loss for our initialization (see Fig. 4). Together these results indicate that though our analysis used canonical parameterization, the corresponding insights are more general and apply more broadly to the general full model. In a similar spirit, analysis of initialization effects for deeper architectures is an interesting avenue of future research.

### 5.3 Higher-Order and Multi-State Markov Chains

While the primary focus of this paper has been on binary first-order Markov chains, we believe it's possible to extend our analysis to both multi-state and higher-order settings. On the multi-state front, akin to the binary case, [23] already demonstrates the effect of switching probability and weight-tying on the final model convergence. Here, first characterizing the loss landscape and then the associated learning dynamics in line with our approach is an interesting direction. On the other hand, a recent work [28] establishes a surprising result that any $k^{\text{th}}$-order Markov chain can be represented by a

three layer transformer with just one head per layer, relying on induction head mechanism. Analyzing gradient flow dynamics using appropriate canonical parameterization (cf. [24]) in this scenario is also a fruitful direction of research.

# 6 Related Works

The recent success of transformer models in deep learning has sparked significant interest and active research in understanding them [38, 25, 16, 27, 15, 37, 40, 32]. In relation to our paper, they can be broadly classified into two topics: (i) **In-context learning (ICL):** ICL refers to the ability of transformers learn and reason from information present in their context [10, 13, 4, 35, 39, 7, 21, 17]. Along this thread, the works most relevant to ours are [8, 14, 24], which use Markovian input data to understand the ICL mechanism. [8, 14] heuristically show how gradient-based updates can learn an induction-head mechanism using a simplified transformer architecture with frozen encodings, query matrices and linear activations. On the other hand, we consider the canonical parameterization, capitalizing on inherent low-rank parameters, to provide a full characterization of the learning dynamics. [24] demonstrates how two-layer transformers with GD learn induction head mechanism when the input has a causal tree dependency, such as in Markov chains. In this work, we focus on the GF dynamics for single-layer transformers and show how they can also converge to local optima, further highlighting the role of initialization. (ii) **Training dynamics:** On the other hand, numerous works have investigated the training dynamics of transformers. For instance, [9] examines the gradient flow in a simplified single-layer transformer, while [33] studies the process by which self-attention integrates input tokens, assuming the decoder learns faster than the attention layer. Unlike these settings, our focus is on understanding the training dynamics of the full transformer model end-to-end. Other related works include [30], which analyzes gradient dynamics in LSTM Seq2seq models, [19], which shows how Vision Transformers learn spatial structures, and [22], which demonstrates that a single-layer transformer can learn a constrained topic model. A closely related work is [18], which shows that self-attention has a Markovian structure, but our focus is on self-attention's capability in modeling Markov chains and the associated training dynamics.

# 7 Conclusion

In this work, we present a novel characterization of gradient flow dynamics for (weight-tied) single-layer transformers with first-order Markov chains. Specifically, we highlight the significant role of the parameter initialization and inherent properties of the Markovian data in determining the parameter convergence to either global minima or local optima. Drawing upon these insights, we offer practical guidelines for parameter initialization, corroborated by empirical results demonstrating their effectiveness. While our current analysis is limited to single-layer models, uncovering similar results with gradient flow analysis for deeper architectures and higher order Markov chains is open and an interesting avenue for future research.

## Acknowledgments and Disclosure of Funding

Ashok would like to thank Aditya Vardhan Varre for many helpful discussions about the project. This work was supported in part by the Swiss National Science Foundation under Grant 200364.

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

# Contents

# A  Single-layer transformer: architecture and results

We first describe the transformer architecture from Sec. 2:

$$\boldsymbol{x}_n = x_n\,\boldsymbol{e} + \boldsymbol{p}_n \in \mathbb{R}^d, \qquad\qquad\qquad \text{(Embedding)}$$

$$\boldsymbol{y}_n = \boldsymbol{x}_n + \sum_{i\in[n]} \text{att}_{n,i}\cdot\boldsymbol{W}_V\,\boldsymbol{x}_i \in \mathbb{R}^d, \qquad \text{(Attention)}$$

$$\boldsymbol{z}_n = \boldsymbol{y}_n + \boldsymbol{W}_2\,\text{ReLU}(\boldsymbol{W}_1\,\boldsymbol{y}_n) \in \mathbb{R}^d, \qquad \text{(Feed-forward)}$$

$$\text{logit}_n = \langle\boldsymbol{a},\boldsymbol{z}_n\rangle + b \qquad\qquad \in \mathbb{R}, \qquad \text{(Linear)}$$

$$f_{\boldsymbol{\theta}}(x_1^n) \triangleq \mathbb{P}_{\boldsymbol{\theta}}\left(x_{n+1}=1 \mid x_1^n\right) = \underbrace{\sigma\left(\text{logit}_n\right)}_{\in[0,1]}. \qquad \text{(Prediction)}$$

Here $\boldsymbol{\theta} \triangleq (\boldsymbol{e}, \{\boldsymbol{p}_n\}_{n=1}^N, \ldots, \boldsymbol{W}_1, \boldsymbol{W}_2, b, \boldsymbol{a}) \in \mathbb{R}^D$ denotes the full list of the transformer parameters from the embedding layer till the linear layer. In the attention layer, the weight assigned to each value, $\text{att}_{n,i}$, is computed by a compatibility function of the query vector $\boldsymbol{q}_n \triangleq \boldsymbol{W}_Q\,\boldsymbol{x}_n$ and the corresponding key vectors $\boldsymbol{k}_i \triangleq \boldsymbol{W}_K\,\boldsymbol{x}_i$ for all $i \in [n]$. More precisely, $\text{att}_{n,i} \triangleq \text{softmax}((\langle\boldsymbol{q}_n,\boldsymbol{k}_1\rangle,\ldots,\langle\boldsymbol{q}_n,\boldsymbol{k}_n\rangle)/\sqrt{d})_i$. $\boldsymbol{W}_{K,Q,V} \in \mathbb{R}^{d\times d}$ are the respective key, query, and value matrices. For multi-headed attention, the same operation is performed on multiple parallel heads, whose outputs are additively combined.

Finally, the transformer parameters $\boldsymbol{\theta} \triangleq (\boldsymbol{e}, \{\boldsymbol{p}_n\}_{n=1}^N, \ldots, b, \boldsymbol{a})$ are trained via the cross-entropy loss on the next-token prediction:

$$L(\boldsymbol{\theta}) \triangleq -\frac{1}{N}\sum_{n\in[N]} \mathbb{E}_{x_1^{n+1}}[x_{n+1}\cdot\log f_{\boldsymbol{\theta}}(x_1^n) + (1-x_{n+1})\cdot\log(1-f_{\boldsymbol{\theta}}(x_1^n))]. \qquad (14)$$

In this paper, we focus on the weight-tied scenario where $\boldsymbol{e} = \boldsymbol{a}$. Hence we let them be a single parameter with $\boldsymbol{\theta} = (\boldsymbol{e}=\boldsymbol{a}, \{\boldsymbol{p}_n\}_{n=1}^N, \ldots, b) \in \mathbb{R}^D$, where $D$ is the total parameter dimensionality.

## A.1  Loss landscape results

Now we recall the theoretical results from [23] about the loss landscape of $L$ in the form of global and local minima.

**Theorem 4 (Global minimum).** *Let the input sequence be $\{x_n\}_{n=1}^N \sim (\boldsymbol{\pi}(p,q), \boldsymbol{P}(p,q))$ for some fixed $(p,q) \in (0,1)^2$. Then for all $(p,q)$, there exists a $\boldsymbol{\theta}_\star \in \mathbb{R}^D$ with an explicit construction such that it is a global minimum for the population loss $L(\cdot)$ in Eq. (14), i.e.*

(i) *$L(\boldsymbol{\theta}) \geq L(\boldsymbol{\theta}_\star)$ for all $\boldsymbol{\theta} \in \mathbb{R}^D$.*

*Further, $\boldsymbol{\theta}_\star$ satisfies:*

(ii) *$\mathbb{P}_{\boldsymbol{\theta}_\star}\left(x_{n+1}=1 \mid x_1^n\right) = \mathbb{P}\left(x_{n+1}=1 \mid x_n\right)$, the Markov kernel.*
(iii) *$L(\boldsymbol{\theta}_\star) = H(x_{n+1}|x_n)$, the entropy rate of the Markov chain.*
(iv) *$\nabla L(\boldsymbol{\theta}_\star) = 0$, i.e. $\boldsymbol{\theta}_\star$ is a stationary point.*

Let $L_\star \triangleq L(\boldsymbol{\theta}_\star)$ be the global minimal loss from Thm. 4. Now we recall the result on the bad local minimum.

**Theorem 5 (Bad local minimum).** *Let the input sequence be $\{x_n\}_{n=1}^N \sim (\boldsymbol{\pi}(p,q), \boldsymbol{P}(p,q))$ for some fixed $(p,q) \in (0,1)^2$. If $p + q > 1$, there exists an explicit $\boldsymbol{\theta}_{\min} \in \mathbb{R}^D$ such that it is a bad local minimum for the loss $L(\cdot)$, i.e.*

(i) *there exists a neighborhood $\mathcal{B}(\boldsymbol{\theta}_{\min}, r)$ with $r > 0$ such that $L(\boldsymbol{\theta}) \geq L(\boldsymbol{\theta}_{\min})$ for all $\boldsymbol{\theta} \in \mathcal{B}(\boldsymbol{\theta}_{\min}, r)$, with $L(\boldsymbol{\theta}_{\min}) > L_\star$.*

*Further, $\boldsymbol{\theta}_{\min}$ satisfies:*

(ii) *$\mathbb{P}_{\boldsymbol{\theta}_\pi}\left(x_{n+1}=1 \mid x_1^n\right) = \mathbb{P}\left(x_{n+1}=1\right) = \pi_1$, the marginal distribution.*
(iii) *$L(\boldsymbol{\theta}_{\min}) = H(x_{n+1}) = H(\boldsymbol{\pi})$, the entropy of the marginal.*
(iv) *$\nabla L(\boldsymbol{\theta}_{\min}) = 0$, i.e. $\boldsymbol{\theta}_{\min}$ is a stationary point.*

# B Low-rank structure of the optima

Here we recall the low-rank structure for the global minima found by SGD consistently across multiple runs when $p + q < 1$ [23, Appendix C.2]. In particular, it is observed that the token and positional encodings point in the same direction $\boldsymbol{\alpha}$, which is a low-rank factor for the weight matrices in the attention and the feedforward layers, which in turn are all rank-one. Mathematically,

**Embedding.** The embedding vector $\boldsymbol{e}$ obeys

$$\boldsymbol{e} = e \cdot \boldsymbol{\alpha}$$

for some $e > 0$ and $\boldsymbol{\alpha} \in \{\pm 1\}^d$. Further, the positional embeddings $\boldsymbol{p}_n$ are constant across positions $n$ pointing in the same direction albeit with a negative scalar, i.e.

$$\boldsymbol{p}_n = -p \cdot \boldsymbol{\alpha}$$

for $p > 0$ and $p \approx \frac{e}{2}$ such that $e > p$ . Thus from Embedding layer,

$$\boldsymbol{x}_n = (e x_n - p) \cdot \boldsymbol{\alpha}, \tag{15}$$

which ensures that the respective embeddings for the bit $x_n = 0$ and $x_n = 1$ are $\boldsymbol{x}_n = -p \cdot \boldsymbol{\alpha}$ and $\boldsymbol{x}_n = (e - p) \cdot \boldsymbol{\alpha}$, which are roughly anti-podal.

**Attention.** Recall from the Attention layer that the output $\boldsymbol{y}_n$ is given by $\boldsymbol{y}_n = \boldsymbol{x}_n + \boldsymbol{W}_O \sum_{i \in [n]} \mathrm{att}_{n,i} \cdot \boldsymbol{W}_V \boldsymbol{x}_i$, where the attention weights $\mathrm{att}_{n,i}$ are computed according to $\mathrm{att}_{n,i} = \exp\left(\langle \boldsymbol{q}_n, \boldsymbol{k}_i \rangle / \sqrt{d}\right) / \left(\sum_{j \in [n]} \exp\left(\langle \boldsymbol{q}_n, \boldsymbol{k}_j \rangle / \sqrt{d}\right)\right)$ with $\boldsymbol{q}_n = \boldsymbol{W}_Q \boldsymbol{x}_n$ and $\boldsymbol{k}_i = \boldsymbol{W}_K \boldsymbol{x}_i$. Here it is observed that the matrix products are all rank-one with $\boldsymbol{\alpha}$ being a factor, i.e.

$$\boldsymbol{W}_O \boldsymbol{W}_V = \boldsymbol{\alpha} \cdot \boldsymbol{v}^\top \in \mathbb{R}^{d \times d}, \quad \text{for some } \boldsymbol{v} \in \mathbb{R}^d,$$

$$\boldsymbol{W}_Q^\top \boldsymbol{W}_K = (q^2 d)\, \boldsymbol{\alpha} \cdot \boldsymbol{\alpha}^\top \in \mathbb{R}^{d \times d}, \quad \text{for some } q \in \mathbb{R}.$$

Hence,

$$\boldsymbol{W}_V \boldsymbol{x}_i = \langle \boldsymbol{v}, \boldsymbol{\alpha} \rangle (e x_i - p)\, \boldsymbol{\alpha},$$

and

$$\frac{\langle \boldsymbol{q}_n, \boldsymbol{k}_i \rangle}{\sqrt{d}} = \frac{1}{\sqrt{d}} \cdot \boldsymbol{x}_n^\top \boldsymbol{W}_Q^\top \boldsymbol{W}_K \boldsymbol{x}_n = \frac{q^2 d}{\sqrt{d}} \cdot (\boldsymbol{x}_n^\top \boldsymbol{\alpha})(\boldsymbol{x}_i^\top \boldsymbol{\alpha}) \stackrel{(\|\boldsymbol{\alpha}\|^2 = d)}{=} \frac{q^2 d^3}{\sqrt{d}} \cdot (e x_n - p)(e x_i - p)$$

$$= q^2 d^{5/2} \cdot (e x_n - p)(e x_i - p).$$

Thus,

$$\boldsymbol{y}_n = \boldsymbol{x}_n + \sum_{i \in [n]} \mathrm{att}_{n,i} \cdot \boldsymbol{W}_O \boldsymbol{W}_V \boldsymbol{x}_i$$

$$= (e x_i - p)\, \boldsymbol{\alpha} + \sum_{i \in [n]} \mathrm{att}_{n,i} \cdot \langle \boldsymbol{v}, \boldsymbol{\alpha} \rangle (e x_i - p)\, \boldsymbol{\alpha}$$

$$= \left( (e x_n - p) + \langle \boldsymbol{v}, \boldsymbol{\alpha} \rangle \sum_{i \in [n]} \frac{\exp\left(q^2 d^{5/2}\, (e x_n - p)(e x_i - p)\right)}{\sum_{j \in [n]} \exp\left(q^2 d^{5/2}\, (e x_n - p)(e x_j - p)\right)} \cdot (e x_i - p) \right) \boldsymbol{\alpha}. \tag{16}$$

It is further noticed that $\langle \boldsymbol{v}, \boldsymbol{\alpha} \rangle \approx 0$ and hence $\boldsymbol{y}_n = (e x_n - p)\, \boldsymbol{\alpha} = \boldsymbol{x}_n$.

**Feed-forward.** For the Feed-forward layer, both the matrices $\boldsymbol{W}_1 \in \mathbb{R}^{r \times d}$ and $\boldsymbol{W}_2 \in \mathbb{R}^{d \times r}$ exhibit rank-one structure with $\boldsymbol{\alpha}$ being one of the factors,

$$\boldsymbol{W}_1 = w \cdot \boldsymbol{w} \cdot \boldsymbol{\alpha}^\top, \quad \text{for some } \boldsymbol{w} \in \{\pm 1\}^r, w > 0, \tag{17}$$

$$\boldsymbol{W}_2 = \boldsymbol{W}_1^\top. \tag{18}$$

Thus $\boldsymbol{W}_1 \boldsymbol{y}_n = d w (e x_n - p)\, \boldsymbol{w}$. Since $-p < 0$ and $e - p > 0$, corresponding to $x_n = 0$ and $x_n = 1$ respectively, we obtain $\mathrm{ReLU}(\boldsymbol{W}_1 \boldsymbol{y}_n) = d w \left((1 - x_n)p \cdot \mathrm{ReLU}(-\boldsymbol{w}) + x_n (e - p) \cdot \mathrm{ReLU}(\boldsymbol{w})\right)$. Denoting the number of ones in $\boldsymbol{w}$ as $\beta$, i.e. $\beta = \sum_{i=1}^r \mathbb{1}(w_i = 1)$, we further simplify:

$$\boldsymbol{W}_2 \mathrm{ReLU}(\boldsymbol{W}_1 \boldsymbol{y}_n) = \boldsymbol{W}_1^\top \mathrm{ReLU}(\boldsymbol{W}_1 \boldsymbol{y}_n)$$

$$= w^2 d \left( (1 - x_n) p \cdot \langle \boldsymbol{w}, \mathrm{ReLU}(-\boldsymbol{w}) \rangle + x_n (e - p) \cdot \langle \boldsymbol{w}, \mathrm{ReLU}(\boldsymbol{w}) \rangle \right) \boldsymbol{\alpha}$$
$$= w^2 d \left( (1 - x_n) p \cdot (\beta - r) + x_n (e - p) \cdot \beta \right) \boldsymbol{\alpha}$$
$$= w^2 d (e x_n - p) \left( (2\beta - r) x_n + r - \beta \right) \boldsymbol{\alpha}.$$

Hence

$$\boldsymbol{z}_n = \boldsymbol{y}_n + \boldsymbol{W}_2 \mathrm{ReLU}(\boldsymbol{W}_1 \boldsymbol{y}_n) = (e x_n - p) \left( 1 + w^2 d \left( (2\beta - r) x_n + r - \beta \right) \right) \boldsymbol{\alpha}. \qquad (19)$$

**Linear.** Using the fact that $\boldsymbol{e} = \boldsymbol{a} = e \cdot \boldsymbol{\alpha}$ due to weight-tying, we obtain from Linear layer that

$$\mathrm{logit}_n = \langle \boldsymbol{e}, \boldsymbol{z}_n \rangle + b = e d (e x_n - p) \left( 1 + w^2 d \left( (2\beta - r) x_n + r - \beta \right) \right) + b. \qquad (20)$$

**Prediction.** We finally obtain that the prediction probability

$$f_{\boldsymbol{\theta}}(x_1^n) = \sigma(\mathrm{logit}_n) = x_n \cdot \sigma \left( e d (e - p) \left( 1 + \beta w^2 d \right) + b \right) + (1 - x_n) \cdot \sigma \left( -e d p \left( 1 + (r - \beta) w^2 d \right) + b \right).$$

Thus we see that the prediction probability and hence the loss function $L(\cdot)$ in Eq. (14) is influenced only by the scalars $e, p, w, b$ and $\beta$.

## C  Canonical reparameterization

Building on the low-rank strucutre of the transformer parameters described above, we consider a special parameterization for them. A key property of this parameterization is that it covers both the global and local minima from Thm. 4 and Thm. 5 for all $(p, q) \in (0, 1)^2$. Recall that Thm. 5 characterizes local minima only for $p + q > 1$ whereas our special parameterization allows to discover local minima even for $p + q < 1$. Our construction follows the same outline as in Eqs. (15)-(20). First we start with the embedding layer.

**Embedding.** We let $\boldsymbol{e} = e \cdot \boldsymbol{\alpha}$ and $\boldsymbol{p}_n = -p \cdot \boldsymbol{\alpha}$ for all $n$ where $e > 0, p = \frac{e}{2}$ and $\boldsymbol{\alpha} \in \{\pm 1\}^d / \sqrt{d}$. Thus the embedding $\boldsymbol{x}_n$ from Eq. (15) simplifies to

$$\boldsymbol{x}_n = e \left( x_n - \frac{1}{2} \right) \boldsymbol{\alpha} \in \{ \pm \frac{e}{2} \} \boldsymbol{\alpha}.$$

**Attention.** Substituting this $\boldsymbol{x}_n$ in Eq. (16), we have

$$\boldsymbol{y}_n = e \left( x_n - \frac{1}{2} \right) \boldsymbol{\alpha} + \langle \boldsymbol{v}, \boldsymbol{\alpha} \rangle \left( \sum_{i \in [n]} \mathrm{att}_{n,i} \cdot e \left( x_i - \frac{1}{2} \right) \right) \boldsymbol{\alpha}, \qquad (21)$$

where the attention weights $\mathrm{att}_{n,i} = \frac{\exp \left( e^2 q^2 d^{5/2} (x_n - \frac{1}{2})(x_i - \frac{1}{2}) \right)}{\sum_{j \in [n]} \exp \left( e^2 q^2 d^{5/2} (x_n - \frac{1}{2})(x_j - \frac{1}{2}) \right)} \in (0, 1)$ for some $q \in \mathbb{R}$. Since $\langle \boldsymbol{v}, \boldsymbol{\alpha} \rangle \approx 0$, we let $\boldsymbol{v} = 0$ and obtain

$$\boldsymbol{y}_n = \boldsymbol{x}_n = e \left( x_n - \frac{1}{2} \right) \boldsymbol{\alpha}.$$

**Feed-forward**. For the feed-forward layer, we observe from Eq. (17) and Eq. (19) that for any $\boldsymbol{w} \in \{\pm 1\}^r$, only the number of 1's in $\boldsymbol{w}$, $\beta$, matters for the final vector $\boldsymbol{z}_n$ which further interacts with the weight scalar $w$. Hence without loss of generality, we set $\boldsymbol{w}$ to be the all-ones vector: $\boldsymbol{w} = \mathbf{1} \in \mathbb{R}^r$ and hence $\beta = r = 4d$. While we observe from Eq. (17) that $\boldsymbol{W}_2 = \boldsymbol{W}_1^\top$ for $p + q < 1$, we observe from the proof of the Thm. 4 for $p + q > 1$ in [23, Appendix B.2] that we need $\boldsymbol{W}_2 = -\boldsymbol{W}_1^\top$ in this scenario. Hence we consider the following parameterization that covers both these scenarios:

$$\boldsymbol{W}_1 = \frac{|w|}{\sqrt{d}} \mathbf{1} \cdot \boldsymbol{\alpha}^\top \in \mathbb{R}^{4d \times d}, \quad \boldsymbol{W}_2 = \frac{w}{\sqrt{d}} \boldsymbol{\alpha} \cdot \mathbf{1}^\top \in \mathbb{R}^{d \times 4d}.$$

Here $w > 0$ ensures $\boldsymbol{W}_2 = \boldsymbol{W}_1^\top$ whereas $w < 0$, $\boldsymbol{W}_2 = -\boldsymbol{W}_1^\top$. Using this parameterization, substituting $\beta = r = 4d$ and $w \mapsto \frac{w}{d}$ in Eq. (19), we get

$$\boldsymbol{z}_n = e \left( x_n - \frac{1}{2} \right) \left( 1 + 4 w |w| x_n \right) \boldsymbol{\alpha}.$$

**Linear.** Since $e = a = e \cdot \alpha$ due to weight-tying, Eq. (20) simplifies to

$$\text{logit}_n = \langle e, z_n \rangle + b = e^2 \left( x_n - \frac{1}{2} \right) (1 + 4w|w|x_n) + b$$

$$\overset{(x_n = x_n^2)}{=} e^2 \left( x_n + 4w|w|x_n - \frac{1}{2} - 2w|w|x_n \right) + b$$

$$= e^2 (1 + 2w|w|) x_n + b - \frac{e^2}{2}.$$

**Prediction.** The next-token prediction probability is

$$f_{(\theta,b)}(x_1^n) = \sigma \left( e^2 (1 + 2w|w|) x_n + b - \frac{e^2}{2} \right), \quad \theta \triangleq (e, w) \in \mathbb{R}^2. \tag{22}$$

**Loss.** While we assumed $e > 0$ in the beginning, in view of Eq. (22) and the fact that $\alpha \in \{\pm 1\}^d / \sqrt{d}$, we see that $e \in \mathbb{R}$ gives us the same expression for probability. Thus the final probability depends on just the three scalars $(e, w, b) \in \mathbb{R}^3$. Defining $\theta = (e, w) \in \mathbb{R}^2$, we recall the cross-entropy loss $L(\cdot)$ from Eq. (4) in Sec. 2 for this canonical model:

$$L(\theta, b) = -\frac{1}{N} \sum_{n \in [N]} \mathbb{E}_{x_1^{n+1}} [x_{n+1} \cdot \log f_{(\theta,b)}(x_1^n) + (1 - x_{n+1}) \cdot \log(1 - f_{(\theta,b)}(x_1^n))]. \tag{23}$$

It turns out that we can further remove the bias $b$ by minimizing the loss over it which we discuss in App. E. For now in the next section, we analyze when it's present as in Eq. (23).

# D   Analysis of the loss with the bias, $L(\boldsymbol{\theta}, b)$, in Eq. (4) and Eq. (23)

In this section, we analyze the loss function with the bias, $L(\boldsymbol{\theta}, b)$, from Eq. (4) and Eq. (23), which will later be useful for studying $L(\boldsymbol{\theta})$. First we characterize the set of its critical points in $\mathbb{R}^3$. To this end, we define the following sets of points

$$\boldsymbol{\Gamma}_\star(p, q) \triangleq \left\{ (e, w, b) \in \mathbb{R}^3 : e^2(1 + 2w|w|) = \log \frac{(1-p)(1-q)}{pq},\, b - \frac{e^2}{2} = \log \frac{p}{1-p} \right\},$$
(24)

$$\boldsymbol{\Gamma}_{\min}(p, q) \triangleq \left\{ (e, w, b) \in \mathbb{R}^3 : e = 0, (p + q - 1)(1 + 2w|w|) > 0, b = \log \frac{p}{q} \right\},$$
(25)

$$\boldsymbol{\Gamma}_{\mathrm{sad}}(p, q) \triangleq \left\{ (e, w, b) \in \mathbb{R}^3 : e = 0, (p + q - 1)(1 + 2w|w|) \leq 0, b = \log \frac{p}{q} \right\}.$$
(26)

The following result establishes that these sets exhaust all the critical points.

**Theorem 6** (All critical points). *Let the input sequence be $\{x_n\}_{n=1}^N \sim (\boldsymbol{\pi}, \boldsymbol{P})$, the transformer parameters $(\boldsymbol{\theta}, b) = (e, w, b) \in \mathbb{R}^3$, and the next-token prediction loss $L(\cdot)$ be as in Eq. (23). Then all the stationary points of $L$ are either in $\boldsymbol{\Gamma}_\star$, $\boldsymbol{\Gamma}_{\min}$, or $\boldsymbol{\Gamma}_{\mathrm{sad}}$, i.e.*

$$\{(\boldsymbol{\theta}, b) \in \mathbb{R}^3 : \nabla L(\boldsymbol{\theta}, b) = 0\} = \boldsymbol{\Gamma}_\star \cup \boldsymbol{\Gamma}_{\min} \cup \boldsymbol{\Gamma}_{\mathrm{sad}}.$$
(27)

*Proof.* We refer to App. J.1. □

Recall the definitions of local minima & maxima, global minima, and that of all the saddle points from Sec. 3.1. We are now ready to present the main result about the loss landscape of $L(\cdot)$.

**Theorem 7** (Loss landscape with bias). *Let the input sequence be $\{x_n\}_{n=1}^N \sim (\boldsymbol{\pi}, \boldsymbol{P})$, the transformer parameters $(e, w, b) \in \mathbb{R}^3$, and the next-token prediction loss $L(\cdot)$ be as in Eq. (23). Then for any $(p, q) \in (0, 1)^2$ with $p + q \neq 1$ and $N \in \mathbb{N}$,*

 (i) *the set of all global minima of $L$ is given by $\boldsymbol{\Gamma}_\star(p, q)$,*

 (ii) *the set of all bad local minima of $L$ is given by $\boldsymbol{\Gamma}_{\min}(p, q)$,*

(iii) *and the set of all saddle points of $L$ is $\boldsymbol{\Gamma}_{\mathrm{sad}}(p, q)$.*

*Furthermore, for any $\boldsymbol{\gamma}_\star \in \boldsymbol{\Gamma}_\star, \boldsymbol{\gamma}_{\min} \in \boldsymbol{\Gamma}_{\min}$, and $\boldsymbol{\gamma}_{\mathrm{sad}} \in \boldsymbol{\Gamma}_{\mathrm{sad}}$, the losses are ordered as*

$$H(x_{n+1} \mid x_n) = L(\boldsymbol{\gamma}_\star) < L(\boldsymbol{\gamma}_{\min}) = L(\boldsymbol{\gamma}_{\mathrm{sad}}) = H(x_{n+1}).$$

**Remark 1.** Note that a bad local minimum is a local minimum whose loss value is strictly less than that of the global minimum, as is the case here. Interestingly, Thm. 7 highlights that all local minima for the loss $L$ are indeed bad local minima.

*Proof.* We refer to App. J.2. □

## D.1   Technical lemmas

The proofs of both Thm. 6 and Thm. 7 rely on few key lemmas that we present below. First we start with the result that rewrites the loss $L(\boldsymbol{\theta}, b)$ from Eq. (23) in a compact manner using the logistic function $\ell_{\log}(\cdot)$.

**Lemma 2** (Loss as a logistic function). *The next-token prediction loss $L(\cdot)$ in Eq. (23) can be written as*

$$L(\boldsymbol{\theta}, b) = \frac{1}{N} \sum_{n \in [N]} \mathbb{E}[\ell_{\log}((2x_{n+1} - 1) \cdot \mathrm{logit}_n)]$$

$$= \mathbb{E}_{X,Y}\left[\ell_{\log}\left((2Y - 1)\left(e^2(1 + 2w|w|)X + b - \frac{e^2}{2}\right)\right)\right],$$
(28)

*where $(X, Y) \in \{0, 1\}^2$ are distributed according to $(X, Y) \sim (\boldsymbol{\pi}, \boldsymbol{P})$, i.e. $X$ is a Bernoulli random variable with $X \sim \boldsymbol{\pi} \equiv \mathrm{Bern}(p/(p + q))$ and $Y|X \sim \boldsymbol{P}(p, q)$, the Markov kernel.*

The following lemma establishes the gradients of the loss function with respect to the parameters $e, w$, and $b$.

**Lemma 3** (Gradient computation). *For any $(e, w, b) \in \mathbb{R}^3$ and the next-token prediction loss $L(\cdot)$ in Eq. (23), the gradients are given by*

$$\frac{\partial L}{\partial e} = \mathbb{E}_X \left[ (f_1 X + f_2)(2X(1 + 2w|w|) - 1) \right] \cdot e,$$

$$\frac{\partial L}{\partial w} = \mathbb{E}_X \left[ (f_1 X + f_2)X \right] \cdot 4e^2 |w|,$$

$$\frac{\partial L}{\partial b} = \mathbb{E}_X \left[ f_1 X + f_2 \right],$$

*where $X \in \{0, 1\}$ is a Bernoulli random variable with $X \sim \text{Bern}(p/(p+q))$, $f_1 = \sigma\left(2e^2 w|w| + b + \frac{e^2}{2}\right) + q - 1 - \sigma\left(b - \frac{e^2}{2}\right) + p$, and $f_2 = \sigma\left(b - \frac{e^2}{2}\right) - p$.*

**Remark 2.** It is interesting to note that the gradients for both $e$ and $w$ are product of an expectation term and an $e$ factor. Also, except for scaling factors in terms of $(e, w, b)$, all the gradients are governed by the two expectation terms $\mathbb{E}[(f_1 X + f_2)X]$ and $\mathbb{E}[f_1 X + f_2]$. This observation plays a key role in obtaining an ordinary differential equation which yields the energy function $\mathcal{E}$, defined in Eq. (11).

Now we characterize the Hessian at both local-minima and saddle points.

**Lemma 4** (Hessian at local-minima and saddle points). *For the canonical parameterization $\gamma = (b, e, w) \in \mathbb{R}^3$ and the next-token prediction loss $L(\cdot)$ in Eq. (23), the Hessian at any $\gamma_{\min} \in \Gamma_{\min}$ or $\gamma_{\text{sad}} \in \Gamma_{\text{sad}}$ is given by*

$$\nabla^2 L(\gamma)\bigg|_{\gamma = \gamma_{\min}, \gamma_{\text{sad}}} = \pi_0 \pi_1 \begin{bmatrix} 1 & 0 & 0 \\ 0 & 2(p + q - 1)(1 + 2w|w|) & 0 \\ 0 & 0 & 0 \end{bmatrix},$$

*where $\pi_0 = \frac{q}{p+q}$ and $\pi_1 = \frac{p}{p+q}$.*

**Remark 3.** We note that the Hessian is computed with the parameter ordering $(b, e, w)$.

The proofs of the lemmas are presented in App. K.

# E  Analysis of the loss without bias, $L(\boldsymbol{\theta})$, and proof of Thm. 1

The proof of Thm. 1, concerning the loss $L(\boldsymbol{\theta})$ in Eq. (5), is similar to that of Thm. 7 which studies the loss $L(\boldsymbol{\theta}, b)$ with the bias present. The main idea is to establish the analogous set of lemmas, as in App. D, when the bias is substituted with its optimal choice. First we recall the loss function

$$L(\boldsymbol{\theta}) \triangleq L(\boldsymbol{\theta}, b_\star) = -\frac{1}{N} \sum_{n \in [N]} \mathbb{E}_{x_1^{n+1}}[x_{n+1} \cdot \log f_{(\boldsymbol{\theta}, b_\star)}(x_1^n) + (1 - x_{n+1}) \cdot \log(1 - f_{(\boldsymbol{\theta}, b_\star)}(x_1^n))],$$
(29)

$$b_\star = \operatorname*{argmin}_{b \in \mathbb{R}} L(\boldsymbol{\theta}, b).$$

We start with the result that establishes a closed form expression for $b_\star$.

**Lemma 5** (Optimal bias). *For $\boldsymbol{\theta} = (e, w) \in \mathbb{R}^2$ and $b \in \mathbb{R}$, let $L(\boldsymbol{\theta}, b)$ be the next-token prediction loss defined in Eq. (23). Then, for any $\boldsymbol{\theta} \in \mathbb{R}^2$, $L(\boldsymbol{\theta}, b)$ is convex in $b$ and the minimizer $b_\star \triangleq \operatorname{argmin}_{b \in \mathbb{R}} L(\boldsymbol{\theta}, b)$ is given by*

$$\exp\left(b_\star - \frac{e^2}{2}\right) = \frac{1}{2A}\left[\frac{p}{q} - 1 + \sqrt{\left(\frac{p}{q} - 1\right)^2 + 4 \cdot \frac{p}{q} \cdot A}\right], \quad A \triangleq \exp(e^2(1 + 2w|w|)). \quad (30)$$

*Consequently, if $e^2(1 + 2w|w|) = \log \frac{(1-p)(1-q)}{pq}$, then $b_\star - \frac{e^2}{2} = \log \frac{p}{1-p}$. If $e = 0$, then $b_\star = \log \frac{p}{q}$.*

Now we rewrite the loss in terms of the logistic function.

**Lemma 6** (Loss as a logistic function). *For any $\boldsymbol{\theta} \in \mathbb{R}^2$, the next-token prediction loss $L(\boldsymbol{\theta})$ in Eq. (29) can be written as*

$$L(\boldsymbol{\theta}) = \frac{1}{N} \sum_{n \in [N]} \mathbb{E}[\ell_{\log}((2x_{n+1} - 1) \cdot \text{logit}_n)]$$
(31)
$$= \mathbb{E}_{X,Y}\left[\ell_{\log}\left((2Y - 1)\left(e^2(1 + 2w|w|)X + b_\star - \frac{e^2}{2}\right)\right)\right].$$

*where $b_\star$ follows from Eq. (30), $(X, Y) \in \{0, 1\}^2$ are distributed according to $(X, Y) \sim (\boldsymbol{\pi}, \boldsymbol{P})$, i.e. $X$ is a Bernoulli random variable with $X \sim \boldsymbol{\pi} \equiv \text{Bern}(p/(p+q))$ and $Y|X \sim \boldsymbol{P}(p, q)$, the Markov kernel.*

The following lemma establishes the gradients of the loss.

**Lemma 7** (Gradient computation). *For any $\boldsymbol{\theta} = (e, w) \in \mathbb{R}^2$ and the next-token prediction loss $L(\boldsymbol{\theta})$ in Eq. (29), the gradients are given by*

$$\frac{\partial L}{\partial e} = \mathbb{E}_X[(f_1 X + f_2)X] \cdot 2(1 + 2w|w|)e,$$

$$\frac{\partial L}{\partial w} = \mathbb{E}_X[(f_1 X + f_2)X] \cdot 4e^2|w|,$$

*where $X \in \{0, 1\}$ is a Bernoulli random variable with $X \sim \text{Bern}(p/(p + q))$, $f_1 = \sigma\left(2e^2w|w| + b_\star + \frac{e^2}{2}\right) + q - 1 - \sigma\left(b_\star - \frac{e^2}{2}\right) + p$, and $f_2 = \sigma\left(b_\star - \frac{e^2}{2}\right) - p$. Further, $\pi_1 f_1 + f_2 = 0$.*

**Remark 4.** We observe above that the gradients for both $e$ and $w$ are proportional to each other, except for the scaling factors in terms of $e$ and $w$. This forms the basis for the derivation of the energy function discussed in App. F.

The following lemma characterizes the Hessian.

**Lemma 8** (Hessian computation). *Let $\boldsymbol{\gamma} = (b, \boldsymbol{\theta}) \in \mathbb{R}^3$ with $\boldsymbol{\theta} = (e, w) \in \mathbb{R}^2$, and $L(\boldsymbol{\gamma})$ be the next-token prediction loss in Eq. (23) and $L(\boldsymbol{\theta})$ be the one in Eq. (29). Let the Hessian of $L$ at $\boldsymbol{\gamma}$ be*

$$H(\boldsymbol{\gamma}) \triangleq \nabla^2_{\boldsymbol{\gamma}\boldsymbol{\gamma}} L = \begin{bmatrix} H_{bb} & H_{b\boldsymbol{\theta}} \\ H_{b\boldsymbol{\theta}}^\top & H_{\boldsymbol{\theta}\boldsymbol{\theta}} \end{bmatrix} = \begin{bmatrix} \nabla^2_{bb} L & \nabla^2_{b\boldsymbol{\theta}} L \\ (\nabla^2_{b\boldsymbol{\theta}} L)^\top & \nabla^2_{\boldsymbol{\theta}\boldsymbol{\theta}} L \end{bmatrix} \in \mathbb{R}^{3 \times 3}.$$

Then the Hessian of L at $\boldsymbol{\theta} \in \mathbb{R}^2$ is given by

$$H(\boldsymbol{\theta}) \triangleq \nabla_{\boldsymbol{\theta}\boldsymbol{\theta}}^2 L = H_{\boldsymbol{\theta}\boldsymbol{\theta}} - H_{b\boldsymbol{\theta}}^\top \cdot H_{bb}^{-1} \cdot H_{b\boldsymbol{\theta}}. \tag{32}$$

Consequently, for any $\boldsymbol{\gamma} = (b, e, w) \in \boldsymbol{\Gamma}_{\min} \cup \boldsymbol{\Gamma}_{\text{sad}}$, the Hessian $H(\boldsymbol{\theta})$ at $\boldsymbol{\theta} = (e, w)$ is given by

$$H(\boldsymbol{\theta}) = \pi_0 \pi_1 \begin{bmatrix} 2(p+q-1)(1+2w|w|) & 0 \\ 0 & 0 \end{bmatrix}, \tag{33}$$

where $\pi_0 = \frac{q}{p+q}$ and $\pi_1 = \frac{p}{p+q}$.

The proofs of the above lemmas are deferred to App. L. We are now ready to present the proof of Thm. 1.

### E.1 Proof of Thm. 1

*Proof.* Let $\boldsymbol{\theta} \in \mathbb{R}^2$ and $\boldsymbol{\gamma}(\boldsymbol{\theta}) = (\boldsymbol{\theta}, b_\star(\boldsymbol{\theta}) \in \mathbb{R}^3$ be its embedding in $\mathbb{R}^3$ with the optimal bias $b_\star(\boldsymbol{\theta}) = \operatorname{argmin}_{b \in \mathbb{R}} L(\boldsymbol{\theta}, b)$ from Lemma 5. Define the following four sets of points:

$$\boldsymbol{\Theta}_\star(p, q) \triangleq \left\{ (e, w) \in \mathbb{R}^2 : e^2(1 + 2w|w|) = \log \frac{(1-p)(1-q)}{pq} \right\},$$

$$\boldsymbol{\Theta}_{\min}(p, q) \triangleq \left\{ (e, w) \in \mathbb{R}^2 : e = 0, (p+q-1)(1+2w|w|) > 0 \right\},$$

$$\boldsymbol{\Theta}_{\max}(p, q) \triangleq \left\{ (e, w) \in \mathbb{R}^2 : e = 0, (p+q-1)(1+2w|w|) < 0 \right\},$$

$$\boldsymbol{\Theta}_{\text{sad}}(p, q) \triangleq \left\{ (e, w) : e = 0, w = -1/\sqrt{2} \right\}.$$

First we show that any critical point of $L : \mathbb{R}^2 \to \mathbb{R}$ has to lie in one of these sets. Then we characterize that they correspond to the set of all global minima, local minima & maxima, and saddle points respectively.

**(i) Set of all critical points:** Recall from Thm. 7 that for any critical point $\boldsymbol{\gamma} = (\boldsymbol{\theta}, b) = (e, w, b) \in \mathbb{R}^3$ of $L$, $\boldsymbol{\gamma} \in \boldsymbol{\Gamma}_\star \cup \boldsymbol{\Gamma}_{\min} \cup \boldsymbol{\Gamma}_{\text{sad}}$. Here the main observation is that all these critical points are of the form $(\boldsymbol{\theta}, b_\star(\boldsymbol{\theta}))$ where $\boldsymbol{\theta} \in \boldsymbol{\Theta}_\star \cup \boldsymbol{\Theta}_{\min} \cup \boldsymbol{\Theta}_{\max} \cup \boldsymbol{\Theta}_{\text{sad}}$. To see this, let $\boldsymbol{\gamma} \in \boldsymbol{\Gamma}_\star$. Here we have $e^2(1 + 2w|w|) = \log \frac{(1-p)(1-q)}{pq}$ from Eq. (24) and hence $\boldsymbol{\theta} \in \boldsymbol{\Theta}_\star$. Further, by Lemma 5, we have that the optimal bias for this $\boldsymbol{\theta}$ satisfies $b_\star - \frac{e^2}{2} = \log \frac{p}{1-p}$, which is precisely the characterization of the bias $b$ for $\boldsymbol{\gamma} = (e, w, b)$ in Eq. (24). Likewise, if $\boldsymbol{\gamma} \in \boldsymbol{\Gamma}_{\min} \cup \boldsymbol{\Gamma}_{\text{sad}}$, we have $e = 0$ and hence $\boldsymbol{\theta} \in \boldsymbol{\Theta}_{\min} \cup \boldsymbol{\Theta}_{\max} \cup \boldsymbol{\Theta}_{\text{sad}}$. Hence by Lemma 5, $b_\star = \log \frac{p}{q}$, matching that of Eq. (25) and Eq. (26). Thus the set of all critical points of $L$ in $\mathbb{R}^3$ are of the form $(\boldsymbol{\theta}, b_\star(\boldsymbol{\theta}))$ with where $\boldsymbol{\theta} \in \boldsymbol{\Theta}_\star \cup \boldsymbol{\Theta}_{\min} \cup \boldsymbol{\Theta}_{\max} \cup \boldsymbol{\Theta}_{\text{sad}}$. Since $\boldsymbol{\Gamma}_\star \cup \boldsymbol{\Gamma}_{\min} \cup \boldsymbol{\Gamma}_{\text{sad}}$ covers the entirety of stationary points of $L$ in $\mathbb{R}^3$, it follows that the set of all stationary points in $\mathbb{R}^2$ is precisely $\boldsymbol{\Theta}_\star \cup \boldsymbol{\Theta}_{\min} \cup \boldsymbol{\Theta}_{\max} \cup \boldsymbol{\Theta}_{\text{sad}}$. Also, the ordering of losses directly follows from the aforementioned observation.

Now we characterize these critical points in terms of the extrema.

**(ii) Set of global and local minima:** From Eq. (24), for any global minimum $\boldsymbol{\gamma}_\star = (\boldsymbol{\theta}_\star, b_\star(\boldsymbol{\theta}_\star))$ of $L$ in $\mathbb{R}^3$, we have $\boldsymbol{\theta}_\star \in \boldsymbol{\Theta}_\star \subseteq \mathbb{R}^2$. Hence by definition, $\boldsymbol{\Theta}_\star$ is the set of all global minima in $\mathbb{R}^2$. A similar argument holds for $\boldsymbol{\Theta}_{\min}$, which establishes that it is a set of all local minima.

**(iii) Set of local maxima and saddle points:** From Eq. (26), for any saddle point $\boldsymbol{\gamma} = (e, w, b_\star(e, w))$ of $L$ in $\mathbb{R}^3$, we have that $e = 0$ and $(p+q-1)(1+2w|w|) \leq 0$. Hence $\boldsymbol{\theta} = (e, w) \in \boldsymbol{\Theta}_{\max} \cup \boldsymbol{\Theta}_{\text{sad}}$. Suppose $\boldsymbol{\theta} \in \boldsymbol{\Theta}_{\max}$ which implies $e = 0, (p+q-1)(1+2w|w|) < 0$. By Lemma 8, the Hessian at $\boldsymbol{\theta}$ (upto a positive scale) is a diagonal matrix with the entries $(p+q-1)(1+2w|w|) < 0$ and 0, corresponding to the directions of $e$ and $w$ respectively. Though one of the eigenvalue here is zero, using a continuity argument as in the proof of Thm. 7 for local minima, we can establish that $\boldsymbol{\theta}$ is indeed a local maximum. Thus $\boldsymbol{\Theta}_{\max}$ is a set of local minima.

Now suppose $(e, w) \in \boldsymbol{\Theta}_{\text{sad}}$. Thus $e = 0$ and $w = -\frac{1}{\sqrt{2}}$. Since it lies at the intersection of $\boldsymbol{\Theta}_{\min}$ and $\boldsymbol{\Theta}_{\max}$, using a neighborhood argument, it's straightforward to see that $\boldsymbol{\Theta}_{\text{sad}}$ is indeed a set of saddle points.

Finally it follows that $\boldsymbol{\Theta}_{\min}, \boldsymbol{\Theta}_{\max}, \boldsymbol{\Theta}_{\mathrm{sad}}$ are the only set of local minima, maxima, and saddle points from the above fact about the characterization of the set of all critical points in terms of these sets and $\boldsymbol{\Theta}_\star$, the ordering of the losses, and using the same argument as in the final steps of the proof of Thm. 7 with the bias. This concludes the proof.

$\square$

# F   Gradient flow analysis without attention

In this section, we analyze the learning dynamics of the transformer parameters $\boldsymbol{\theta} = (e, w) \in \mathbb{R}^2$ without the attention scalar. First, we present few important lemmas regarding the same, useful for the proofs of Thm. 2 and Thm. 8 later. Recall from Sec. 4 that the trajectory $(\boldsymbol{\theta}_t)_{t \geq 0}$ is governed by

$$\frac{\mathrm{d}\boldsymbol{\theta}_t}{\mathrm{d}t} = -\nabla L(\boldsymbol{\theta}_t), \quad \boldsymbol{\theta}_t = (e_t, w_t) \in \mathbb{R}^2, \, t \geq 0, \tag{GF}$$

starting with a randomly initalized $\boldsymbol{\theta}_0$. The *energy function* $\mathcal{E}(\cdot, \cdot)$ is defined as

$$\mathcal{E}(e, w) \triangleq e^2 - (w^2 + \mathrm{sign}(w) \cdot \log|w|), \quad \forall (e, w) \in \mathbb{R}^2 \setminus \text{e-axis}, \tag{34}$$

where e-axis $\triangleq \{(e, w = 0)\}$ and w-axis $\triangleq \{(e = 0, w)\}$. Note that $\mathcal{E}_{\mathrm{sad}} = \mathcal{E}(0, -\frac{1}{\sqrt{2}}) = -\frac{1 + \log 2}{2}$. We re-present the Lemma 1 from Sec. 4 below for the sake of completeness.

**Lemma 9** (Constant energy along the flow). *For any $(p, q) \in (0, 1)^2$ and initialization $\boldsymbol{\theta}_0 = (e_0, w_0) \in \mathbb{R}^2$, let $(\boldsymbol{\theta}_t)_{t \geq 0}$ be the corresponding GF trajectory starting from $\boldsymbol{\theta}_0$. If $w_0 \neq 0$, then the energy stays constant along the trajectory, i.e.*

$$\mathcal{E}(\boldsymbol{\theta}_t) = e_t^2 - (w_t^2 + \mathrm{sign}(w_t) \cdot \log|w_t|) = \mathcal{E}(\boldsymbol{\theta}_0), \quad \forall t \geq 0. \tag{35}$$

*On the other hand, if $w_0 = 0$, $w_t = 0$ for all $t \geq 0$. Hence, if we initialize on e-axis the trajectory always stays on the e-axis.*

Now we establish that the GF trajectories always converge.

**Lemma 10** (GF convergence). *Let $(\boldsymbol{\theta}_t)_{t \geq 0}$ be a continuously diferentiable GF trajectory starting from $\boldsymbol{\theta}_0$. Then for all initializations $\boldsymbol{\theta}_0 \in \mathbb{R}^2$,*

(i) *$(\boldsymbol{\theta}_t)_{t \geq 0}$ is bounded,*

(ii) *there exists a $\boldsymbol{\theta}_{\lim} \in \mathbb{R}^2$ such that $\lim_{t \to \infty} \boldsymbol{\theta}_t = \boldsymbol{\theta}_{\lim}$ and*

(iii) *$\lim_{t \to \infty} \|\nabla L(\boldsymbol{\theta}_t)\| = \|\nabla L(\boldsymbol{\theta}_{\lim})\| = 0$.*

*Hence $\boldsymbol{\theta}_{\lim}$ is a critical point of $L$.*

The following result characterizes the energy of the limit point.

**Lemma 11** (Energy at the limit point). *Consider the same setting as in Lemma 10. If $\boldsymbol{\theta}_0 \in \mathbb{R}^2 \setminus \text{e-axis}$, then $\mathcal{E}(\boldsymbol{\theta}_{\lim}) = \mathcal{E}(\boldsymbol{\theta}_0)$. Hence $\boldsymbol{\theta}_{\lim}$ lies at the intersection of the contour line $\mathcal{E}(e, w) = \mathcal{E}_0$ with the set of critical points of $L$ in $\mathbb{R}^2$.*

*On the other hand, if $\boldsymbol{\theta}_0 \in \text{e-axis}$, then $\boldsymbol{\theta}_{\lim} \in \text{e-axis}$.*

We now study the energy function on the w-axis which plays a key role in the GF analysis.

**Lemma 12** (Analysis of the energy function). *Let $\mathcal{E}(\cdot, \cdot)$ be the energy function defined in Eq. (48) and $f(w) \triangleq \mathcal{E}(e = 0, w) = -(w^2 + \mathrm{sign}(w) \cdot \log|w|)$ be the energy evaluated on w-axis for $w \in \mathbb{R} \setminus \{0\}$. Then*

(i) *$f : (-\infty, -1/\sqrt{2}] \to (-\infty, \mathcal{E}_{\mathrm{sad}}]$ is monotonically increasing with $\lim_{w \to -\infty} f(w) = -\infty$ and the maximum being $f(-1/\sqrt{2}) = \mathcal{E}_{\mathrm{sad}}$,*

(ii) *$f : [-1/\sqrt{2}, 0) \to [\mathcal{E}_{\mathrm{sad}}, -\infty)$ is monotonically decreasing with $\lim_{w \to 0^-} f(w) = -\infty$,*

(iii) *$f'(-\frac{1}{\sqrt{2}}) = 0$, and*

(iv) *$f : (0, \infty) \to (-\infty, \infty)$ is monotonically decreasing with $\lim_{w \to 0^+} f(w) = \infty$ and $\lim_{w \to \infty} f(w) = -\infty$.*

We are now ready to prove Thm. 2 corresponding to $p + q > 1$.

## F.1   Proof of Thm. 2

*Proof.* Let $\boldsymbol{\theta}_0 = (e_0, w_0) \in \mathbb{R}^2$ be the initialization for the GF trajectory $(\boldsymbol{\theta}_t)_{t \geq 0}$. Recall that

$$\mathcal{I}_{\min} \triangleq \left\{ (e, w) : w \in (-1/\sqrt{2}, 0), \, e \in (-g(w), g(w)), \, g(w) = \sqrt{w^2 - \log(-w) + \mathcal{E}_{\mathrm{sad}}} \right\}$$

$$\cup \{(e, w) : w > 0\} \cup \{(e, w) : w = 0\},$$

$$\mathcal{I}_{\text{sad}} \triangleq \left\{ (e, w) : w \in [-1/\sqrt{2}, 0), \ e = \pm\sqrt{w^2 - \log(-w) + \mathcal{E}_{\text{sad}}} \right\},$$

$$\mathcal{I}_{\text{max}} \triangleq \left\{ (e, w) : e = 0, \ w < -1/\sqrt{2} \right\},$$

$$\mathcal{I}_\star \triangleq \mathbb{R}^2 \setminus \left( \mathcal{I}_{\text{min}} \cup \mathcal{I}_{\text{sad}} \cup \mathcal{I}_{\text{max}} \right).$$

We consider the cases $\boldsymbol{\theta}_0 \in$ e-axis and $\boldsymbol{\theta}_0 \in \mathbb{R}^2 \setminus$ e-axis separately. First recall from Thm. 1 and Eq. (6) that for $p + q > 1$, the loci of the global minima, $e^2(1 + 2w|w|) = \log\frac{(1-p)(1-q)}{pq} < 0$, lies entirely in the negative half-plane corresponding to $w < -\frac{1}{\sqrt{2}}$. On the other hand, all the local minima, maxima and the saddle points span the w-axis corresponding to $e = 0$.

**(i) $\boldsymbol{\theta}_0 \in \mathbb{R}^2 \setminus$ e-axis:** Let $\mathcal{E}_0 = \mathcal{E}(\boldsymbol{\theta}_0) \in \mathbb{R}$. By Lemmas. (9), (10), and (11), we have that the trajectory $(\boldsymbol{\theta}_t)_{t \geq 0}$ always stays on the contour line $\mathcal{E}(e, w) = \mathcal{E}_0$ and converges to the limit $\boldsymbol{\theta}_{\text{lim}}$ which is an intersection of this contour line with the set of critical points of $L$. Hence the crux of the proof is to establish where these intersections occur based on the initialization $\boldsymbol{\theta}_0$ and the initial energy $\mathcal{E}_0$. This gives rise to the set of initializations $\mathcal{I}_{\text{min}}, \mathcal{I}_{\text{max}}, \mathcal{I}_{\text{sad}}$, and $\mathcal{I}_\star$ that correspond to the limit being a local minimum/maximum, a saddle point, or a global minimum.

We characterize them individually below starting with $\mathcal{I}_{\text{min}}$.

**Initializations for local minima, $\mathcal{I}_{\text{min}}$.** For $\boldsymbol{\theta}_0 = (e_0, w_0) \in \mathbb{R}^2 \setminus$ e-axis, assume that $w_0 > 0$. Since $\mathcal{E}_0 \in \mathbb{R}$, there exists an unique $w_\star > 0$ such that $f(w_\star) = \mathcal{E}(0, w_\star) = \mathcal{E}_0$ by Lemma 12, (iv). Further using the fact that the energy contour lines do not cross each other (by definition of a contour line) and the fact they do not intersect the e-axis (it's an energy barrier as discussed in Sec. 4), it follows that the contour line $\mathcal{E}(e, w) = \mathcal{E}_0$ stays entirely in the positive half-plane corresponding to $w > 0$ and $w_\star > 0$ is the unique (and only) intersection of this line with the w-axis, and hence the set of critical points. Since the w-axis corresponding to $w > 0$ is a set of a local minima (Eq. (7)), it follows that any initialization $(e_0, w_0)$ with $w_0 > 0$ converges to a local minimum.

Now suppose $-\frac{1}{\sqrt{2}} < w_0 < 0$ and $e_0 \in (-g(w_0), g(w_0))$, where $g(w_0) = \sqrt{w^2 - \log(-w) + \mathcal{E}_{\text{sad}}}$. Thus $|e_0| < g(w_0)$ and hence $e_0^2 - (w_0^2 - \log(-w_0)) = \mathcal{E}(e_0, w_0) = \mathcal{E}_0 < \mathcal{E}_{\text{sad}}$. Hence by Lemma 12, (iii), there is a unique intersection of the contour line $\mathcal{E}(e, w) = \mathcal{E}_0$ with the w-axis, which lies in the region $\left( -\frac{1}{\sqrt{2}}, 0 \right)$. Further note that this contour line cannot intersect with the global minima loci as it lies in the half-plane $w < -\frac{1}{\sqrt{2}}$, and hence its only intersection with the set of critical points is this segment of w-axis, which is precisely the set of local minima the GF initialized on this line would converge to.

Thus we have shown that any initialization in $\mathcal{I}_{\text{min}} \setminus \cup \{(e, w) : w = 0\}$ converges to a local minimum, the set of which exhausts all the set of local minima $\boldsymbol{\Theta}_{\text{min}}$ except for the origin. Below we will estbalish that any initialization on e-axis $= \{(e, w) : w = 0\}$ converges to the origin, implying $\mathcal{I}_{\text{min}}$ is the full set of initializations for which the limit is a local minimum.

**Initializations for saddle points, $\mathcal{I}_{\text{sad}}$.** It's straightforward to see that for any $\boldsymbol{\theta}_0 \in \mathcal{I}_{\text{sad}}$, $e_0^2 - (w_0^2 - \log(-w_0)) = \mathcal{E}(0, -\frac{1}{\sqrt{2}}) = \mathcal{E}_{\text{sad}}$. Since $-\frac{1}{\sqrt{2}} \leq w_0 < 0$, the point $(w, e) = (-\frac{1}{\sqrt{2}}, 0)$ is the only intersection of the contour line with the set of critical points, any initialization in $\mathcal{I}_{\text{sad}}$ converges to the saddle point. On the other hand, there also exists a contour line $e_0^2 - (w_0^2 - \log(-w_0)) = \mathcal{E}_{\text{sad}}$ for $w_0 < -\frac{1}{\sqrt{2}}$ that passes through $(-\frac{1}{\sqrt{2}}, 0) \in \mathbb{R}^2$ and further intersecting with the global minima loci $\boldsymbol{\Theta}_\star$. However, if we initialize on this line the flow escapes away from the saddle point and converges instead to a global minimum. To show this, it suffices to prove that $\frac{de_t}{dt} > 0$ and $\frac{dw_t}{dt} < 0$ if $e_0 > 0$ and $w_0 < -\frac{1}{\sqrt{2}}$, such that $(w_0, e_0)$ is close to the saddle point $(-\frac{1}{\sqrt{2}}, 0)$ (the case for $e_0 < 0$ is similar as the flow is symmetric in $e \in \mathbb{R}$). From Lemma 7 and the definition of the GF, we have that

$$\frac{de_t}{dt} = -\frac{\partial L}{\partial e}(e_0, w_0) = 2\mathbb{E}_X\left[(f_1 X + f_2)X\right] \cdot (1 - 2w_0^2))e_0$$

$$\frac{dw_t}{dt} = -\frac{\partial L}{\partial w}(e_0, w_0) = 4\mathbb{E}_X\left[(f_1 X + f_2)X\right] \cdot (-e_0^2 w_0).$$

So it suffices to show that $\mathbb{E}_X\left[(f_1 X + f_2)X\right] > 0$. To establish this, we have from Lemma 5 that

$$\mathbb{E}_X\left[(f_1 X + f_2)X\right] = \mathbb{E}[X](f_1 + f_2) = \pi_1\left(-\frac{f_2}{\pi_1} + f_2\right) = -\pi_0 \cdot f_2.$$

From the defintion of $f_2$ and the optimal bias $b_\star$ in Lemma 7 and Lemma 5 respectively, we obtain

$$f_2 = \sigma\left(b_\star - \frac{e_0^2}{2}\right) - p = \left(1 + \exp\left(-b_\star + \frac{e_0^2}{2}\right)\right)^{-1} - p$$

$$= \left(1 + \frac{2A}{\frac{p}{q} - 1 + \sqrt{\left(\frac{p}{q} - 1\right)^2 + 4 \cdot \frac{p}{q} \cdot A}}\right)^{-1} - p, \quad A \triangleq \exp(e_0^2(1 - 2w_0^2)).$$

When $e_0 = 0$, we have $A = 1$ and hence

$$f_2 = \left(1 + \frac{q}{p}\right)^{-1} - p = \frac{p}{p+q} - p = -\frac{p}{p+q}(p + q - 1) < 0, \tag{36}$$

where we used the fact that $p + q > 1$. Hence by continuity of $f_2$ in $e_0$, for $e_0$ sufficiently close to 0, $f_2 < 0$ which proves our claim about the direction of the flow close to the saddle point. By using the continuity of the flow, it follows that GF cannot converge to saddle point when initialized on this contour line for $w_0 < -\frac{1}{\sqrt{2}}$. Thus $\mathcal{I}_{\text{sad}}$ is the only set of initializations for convergence to $\Theta_{\text{sad}}$.

**Initializations for local maxima, $\mathcal{I}_{\text{max}}$.** If $p + q > 1$, we have from Thm. 1 that $\Theta_{\text{max}} = \left\{(e, w) \in \mathbb{R}^2 : e = 0, (1 + 2w|w|) < 0\right\} = \left\{(e, w) \in \mathbb{R}^2 : e = 0, w < -\frac{1}{\sqrt{2}}\right\}$. Thus for any $\boldsymbol{\theta}_0 \in \Theta_{\text{max}}$, $\frac{d\boldsymbol{\theta}_t}{dt} = 0$ for all $t \geq 0$ and hence $\boldsymbol{\theta}_{\text{lim}} = \boldsymbol{\theta}_0$. Further if we slightly perturb away from this set, from Eq. (36) it follows that the flow diverges and hence it's an unstable set of critical points (they are local maxima indeed). Thus the only set of initializations leading to local maxima are $\mathcal{I}_{\text{max}} = \Theta_{\text{max}}$.

**Initializations for the global minima, $\mathcal{I}_\star$.** Since the set of all critical points of $L$ is $\Theta_\star \cup \Theta_{\text{min}} \cup \Theta_{\text{max}} \cup \Theta_{\text{sad}}$, and the initializations in $\mathcal{I}_{\text{min}}$, $\mathcal{I}_{\text{sad}}$, and $\mathcal{I}_{\text{max}}$ converge to $\Theta_{\text{min}}$, $\Theta_{\text{sad}}$, and $\Theta_{\text{max}}$ respectively, it follows that the set of initializations for which the GF converges to global minima is $\mathcal{I}_\star = \mathbb{R}^2 \setminus (\mathcal{I}_{\text{min}} \cup \mathcal{I}_{\text{sad}} \cup \mathcal{I}_{\text{max}})$.

In fact, since the loci of the global minima lies in the half-plane correspondint to $w < -\frac{1}{\sqrt{2}}$ when $p + q > 1$, we can precisely determine the location of the global minimum for which the intersection occurs for any $\boldsymbol{\theta}_0 \in \mathcal{I}_\star$. Specifically, we can solve the pair of equations $\mathcal{E}(e, w) = e^2 - w^2 + \log(-w) = \mathcal{E}_0$ and $e^2(1 - 2w^2) = \log\frac{(1-p)(1-q)}{pq}$ which has a unique solution for $w < 0$ (upto a sign flip in $e$).

**(ii) $\boldsymbol{\theta}_0 \in$ e-axis $\Rightarrow \boldsymbol{\theta}_0 \in \mathcal{I}_{\text{min}}$:** If $\boldsymbol{\theta}_0 = (e_0, w_0) \in$ e-axis, we have that $w_0 = 0$ and hence $w_t = 0$ for all $t \geq 0$ (Lemma 9). Lemma 10-(i) also establishes that the iterates $(\boldsymbol{\theta}_t = (e_t, 0))_{t \geq 0}$ stay bounded on the e-axis and monotonically decrease. Since the origin is the only critical point of $L$ on the e-axis, and $\lim_{t \to \infty} \boldsymbol{\theta}_t = \boldsymbol{\theta}_{\text{lim}}$ exists, it follows that $\boldsymbol{\theta}_{\text{lim}} = (0, 0)$, a local minima. Thus $\boldsymbol{\theta}_0 \in \mathcal{I}_{\text{min}}$.

This concludes the proof for all the initializations $\boldsymbol{\theta}_0 \in \mathbb{R}^2$.

**Gaussian initialization $\mathcal{N}(0, \sigma^2 \boldsymbol{I}_2)$.** When $\boldsymbol{\theta}_0$ is initialized according to the standard Gaussian distribution $\mathcal{N}(0, \sigma^2 \boldsymbol{I}_2)$ with $\sigma^2 \ll \frac{1}{\sqrt{2}}$, we note that $\boldsymbol{\theta}_0$ lands in the set $\mathcal{I}_{\text{min}}$ with high probability. In fact, this probability can be made arbitrarily close to 1 depending on $\sigma^2$. Thus this initialization will lead to a local minimum convergence on the w-axis. $\qquad\square$

## F.2 Gradient flow dynamics for $p + q < 1$

**Theorem 8** (GF dynamics for $p + q < 1$). *Under the same setting as in Thm. 2 with $p + q < 1$, and any initialization $\boldsymbol{\theta}_0 \in \mathbb{R}^2$, the GF trajectory always converges to a $\boldsymbol{\theta}_{\text{lim}} \in \mathbb{R}^2$ which is a critical point of the loss $L$. More specifically, $\boldsymbol{\theta}_{\text{lim}}$ is*

(i) *a local minimum if*

$$\boldsymbol{\theta}_0 \in \mathcal{I}_{\text{min}} \triangleq \left\{(e, w) : w < -1/\sqrt{2}, e \in (-g(w), g(w)), g(w) = \sqrt{w^2 - \log(-w) + \mathcal{E}_{\text{sad}}}\right\},$$

(ii) *a saddle point if $\boldsymbol{\theta}_0 \in \mathcal{I}_{\text{sad}} \triangleq \left\{(e, w) : w \leq -1/\sqrt{2}, e = \pm\sqrt{w^2 - \log(-w) + \mathcal{E}_{\text{sad}}}\right\}$,*

(iii) *a local maximum if $\boldsymbol{\theta}_0 \in \mathcal{I}_{\mathrm{max}} \triangleq \{(e, w) : e = 0, \, w > -1/\sqrt{2}\}$,*

(iv) *and a global minimum if $\boldsymbol{\theta}_0 \in \mathbb{R}^2 \setminus (\mathcal{I}_{\mathrm{min}} \cup \mathcal{I}_{\mathrm{sad}} \cup \mathcal{I}_{\mathrm{max}})$.*

*Consequentely, if we use the standard initialization $\boldsymbol{\theta}_0 \sim \mathcal{N}(0, \sigma^2 \boldsymbol{I}_2)$ with $\sigma^2 \ll 1/\sqrt{2}$, $\boldsymbol{\theta}_{\mathrm{lim}}$ will be a global minimum.*

*Proof.* The proof for the case of $p + q < 1$ essentially follows the same steps as that of $p + q > 1$. If the initialization is not on the e-axis we use the energy equation to establish the convergence to the critical point at the intersection of the energy contour line with the critical set and if it starts on the e-axis, the only change is that it now converges to the global minimum instead of the origin as in the earlier case. This is due to the fact that origin turns out to be a local maximum when $p + q < 1$ and hence it's an unstable critical point (which can be established as in the proof of Thm. 2 for $\mathcal{I}_{\mathrm{max}}$). □

# G   Gradient flow analysis with attention

In this section, we analyze the learning dynamics of the transformer parameters $\boldsymbol{\theta} \in \mathbb{R}^3$ with the attention scalar $a \in \mathbb{R}$, i.e. $\boldsymbol{\theta} = (e, w, a) \in \mathbb{R}^3$. Similar to the analysis for $\boldsymbol{\theta} = (e, w) \in \mathbb{R}^2$, we first introduce the canonical parameterization including $a \in \mathbb{R}$, then analyze the corresponding loss function $L(\cdot)$ in terms of its gradients and critical points, and capitalize on it to study the gradient flow dynamics using the energy. We first start with the parameterization.

## G.1   Canonical parameterization with attention

**Embedding.** Recall from App. C that we let $\boldsymbol{e} = e \cdot \boldsymbol{\alpha}$ and $\boldsymbol{p}_n = -p \cdot \boldsymbol{\alpha}$ for all $n$ where $e > 0, p = \frac{e}{2}$ and $\boldsymbol{\alpha} \in \{\pm 1\}^d / \sqrt{d}$. This results in the embedding

$$\boldsymbol{x}_n = e\left(x_n - \frac{1}{2}\right)\boldsymbol{\alpha}.$$

**Attention.** Similarly, we recall from Eq. (21) that the attention output $\boldsymbol{y}_n$ is given by

$$\boldsymbol{y}_n = e\left(x_n - \frac{1}{2}\right)\boldsymbol{\alpha} + \langle \boldsymbol{v}, \boldsymbol{\alpha} \rangle \left(\sum_{i \in [n]} \mathrm{att}_{n,i} \cdot e\left(x_i - \frac{1}{2}\right)\right)\boldsymbol{\alpha}, \tag{37}$$

where

$$\mathrm{att}_{n,i} \triangleq \exp\left(\langle \boldsymbol{q}_n, \boldsymbol{k}_i \rangle / \sqrt{d}\right) \Big/ \left(\sum_{j \in [n]} \exp\left(\langle \boldsymbol{q}_n, \boldsymbol{k}_j \rangle / \sqrt{d}\right)\right), \quad \boldsymbol{q}_n = \boldsymbol{W}_Q \boldsymbol{x}_n, \quad \boldsymbol{k}_i = \boldsymbol{W}_K \boldsymbol{x}_i,$$

$$\boldsymbol{W}_Q^\top \boldsymbol{W}_K = (q^2 d)\,\boldsymbol{\alpha} \cdot \boldsymbol{\alpha}^\top \in \mathbb{R}^{d \times d}, \quad \text{for some } q \in \mathbb{R}.$$

Instead of the softmax, now we assume that the attention weights are linear in the scaled dot product, i.e.

$$\begin{aligned}
\mathrm{att}_{n,i} &= \frac{\langle \boldsymbol{q}_n, \boldsymbol{k}_i \rangle}{n\sqrt{d}} = \frac{1}{\sqrt{d}} \cdot \boldsymbol{x}_n^\top \boldsymbol{W}_Q^\top \boldsymbol{W}_K \boldsymbol{x}_n = \frac{q^2 d}{n\sqrt{d}} \cdot (\boldsymbol{x}_n^\top \boldsymbol{\alpha})(\boldsymbol{x}_i^\top \boldsymbol{\alpha}) \\
&\overset{(\|\boldsymbol{\alpha}\|^2 = d)}{=} \frac{q^2 d^3}{n\sqrt{d}} \cdot (e x_n - p)(e x_i - p) \\
&= \frac{q^2 d^{5/2}}{n} \cdot (e x_n - p)(e x_i - p) \\
&= \frac{q^2 d^{5/2} e^2}{n} \cdot \left(x_n - \frac{1}{2}\right)\left(x_i - \frac{1}{2}\right).
\end{aligned} \tag{38}$$

Note that the $1/n$ factor is to ensure normalization for the attention weights in Eq. (37). Now substituting Eq. (38) in Eq. (37), we obtain

$$\begin{aligned}
\boldsymbol{y}_n &= e\left(x_n - \frac{1}{2}\right)\boldsymbol{\alpha} + \langle \boldsymbol{v}, \boldsymbol{\alpha} \rangle \left(\sum_{i \in [n]} \mathrm{att}_{n,i} \cdot e\left(x_i - \frac{1}{2}\right)\right)\boldsymbol{\alpha} \\
&= \left[e\left(x_n - \frac{1}{2}\right) + \langle \boldsymbol{v}, \boldsymbol{\alpha} \rangle \left(\sum_{i \in [n]} \frac{1}{n} q^2 d^{5/2} e^2 (x_n - \frac{1}{2})(x_i - \frac{1}{2})\right) \cdot e\left(x_i - \frac{1}{2}\right)\right]\boldsymbol{\alpha} \\
&= \left[e\left(x_n - \frac{1}{2}\right)\left(1 + \langle \boldsymbol{v}, \boldsymbol{\alpha} \rangle q^2 d^{5/2} e^2 \left(x_i - \frac{1}{2}\right)^2\right)\right]\boldsymbol{\alpha} \\
&= \left[e\left(x_n - \frac{1}{2}\right)\left(1 + \underbrace{\langle \boldsymbol{v}, \boldsymbol{\alpha} \rangle q^2 d^{5/2} \frac{1}{4}}_{a} \cdot e^2 \cdot\right)\right]\boldsymbol{\alpha} \\
&= e\left(x_n - \frac{1}{2}\right)\left(1 + a e^2\right)\boldsymbol{\alpha},
\end{aligned}$$

where we used the fact that $(x_i - \frac{1}{2})^2 = \frac{1}{4}$ since $x_i \in \{0, 1\}$, and

$$a \triangleq \frac{\langle \boldsymbol{v}, \boldsymbol{\alpha} \rangle q^2 d^{5/2}}{4} \tag{39}$$

is the attention scalar. Note that this includes the scaling $\langle \boldsymbol{v}, \boldsymbol{\alpha} \rangle$ from the value matrix $\boldsymbol{W}_V$ and $q^2$ from the query-key dot product. Thus we succinctly have

$$\boldsymbol{y}_n = e \left( x_n - \frac{1}{2} \right) \left( 1 + ae^2 \right) \boldsymbol{\alpha}. \tag{40}$$

**Feed-forward**. For the feed-forward layer, we have that $\boldsymbol{W}_1 = \frac{|w|}{\sqrt{d}} \mathbf{1} \cdot \boldsymbol{\alpha}^\top \in \mathbb{R}^{4d \times d}$, $\boldsymbol{W}_2 = \frac{w}{\sqrt{d}} \boldsymbol{\alpha} \cdot \mathbf{1}^\top \in \mathbb{R}^{d \times 4d}$. Hence Eq. (40) implies

$$\boldsymbol{W}_1 \boldsymbol{y}_n = \frac{|w|}{\sqrt{d}} \mathbf{1} \cdot \boldsymbol{\alpha}^\top \left[ e \left( x_n - \frac{1}{2} \right) \left( 1 + ae^2 \right) \right] \boldsymbol{\alpha} = \frac{|w|}{\sqrt{d}} \left[ e \left( x_n - \frac{1}{2} \right) \left( 1 + ae^2 \right) \right] \mathbf{1}.$$

Thus,

$$
\begin{aligned}
\mathrm{ReLU}(\boldsymbol{W}_1 \boldsymbol{y}_n) &= \frac{|w|}{\sqrt{d}} \mathbf{1} \cdot \mathrm{ReLU} \left( \left[ e \left( x_n - \frac{1}{2} \right) \left( 1 + ae^2 \right) \right] \right) \\
&= \frac{|w|}{\sqrt{d}} \mathbf{1} \cdot e\, \mathrm{ReLU} \left( \left[ \left( x_n - \frac{1}{2} \right) \left( 1 + ae^2 \right) \right] \right) \\
&= \frac{|w|}{\sqrt{d}} \mathbf{1} \cdot e \left( \frac{x_n}{2} \mathrm{ReLU}(1 + ae^2) + \frac{1 - x_n}{2} \mathrm{ReLU}(-1 - ae^2) \right) \\
&= \frac{|w|}{2\sqrt{d}} \mathbf{1} \cdot e \left( x_n \left[ \mathrm{ReLU}(1 + ae^2) - \mathrm{ReLU}(-1 - ae^2) \right] + \mathrm{ReLU}\left( -1 - ae^2 \right) \right).
\end{aligned}
$$

Using $\mathrm{ReLU}(x) - \mathrm{ReLU}(-x) = x$ above,

$$\mathrm{ReLU}(\boldsymbol{W}_1 \boldsymbol{y}_n) = \frac{|w|}{2\sqrt{d}} \mathbf{1} \cdot e \left( x_n \left( 1 + ae^2 \right) + \mathrm{ReLU}\left( -1 - ae^2 \right) \right).$$

Hence,

$$
\begin{aligned}
\boldsymbol{W}_2 \mathrm{ReLU}(\boldsymbol{W}_1 \boldsymbol{y}_n) &= \frac{w}{\sqrt{d}} \boldsymbol{\alpha} \cdot \mathbf{1}^\top \frac{|w|}{2\sqrt{d}} \mathbf{1} \cdot e \left( x_n \left( 1 + ae^2 \right) + \mathrm{ReLU}\left( -1 - ae^2 \right) \right) \\
&= 2w|w|e \left( x_n \left( 1 + ae^2 \right) + \mathrm{ReLU}\left( -1 - ae^2 \right) \right) \boldsymbol{\alpha} \\
&= 2w|w|e \left( x_n \left( 1 + ae^2 \right) + \frac{(-1 - ae^2)}{2} + \frac{|1 + ae^2|}{2} \right) \boldsymbol{\alpha} \\
&= 2w|w|e \left( \left( x_n - \frac{1}{2} \right) \left( 1 + ae^2 \right) + \frac{|1 + ae^2|}{2} \right) \boldsymbol{\alpha}.
\end{aligned}
$$

Thus the embedding $\boldsymbol{z}_n$ is given by

$$
\begin{aligned}
\boldsymbol{z}_n &= \boldsymbol{y}_n + \boldsymbol{W}_2 \mathrm{ReLU}(\boldsymbol{W}_1 \boldsymbol{y}_n) \\
&= \left[ e \left( x_n - \frac{1}{2} \right) \left( 1 + ae^2 \right) \right] \boldsymbol{\alpha} + 2w|w|e \left( \left( x_n - \frac{1}{2} \right) \left( 1 + ae^2 \right) + \frac{|1 + ae^2|}{2} \right) \boldsymbol{\alpha} \\
&= e \left[ \left( x_n - \frac{1}{2} \right) \left( 1 + ae^2 \right) \left( 1 + 2w|w| \right) + w|w|(1 + ae^2)| \right] \boldsymbol{\alpha}.
\end{aligned}
$$

**Linear.** Since $\boldsymbol{e} = \boldsymbol{a} = e \cdot \boldsymbol{\alpha}$ due to weight-tying, the logits are given by

$$\mathrm{logit}_n(e, w, a, b) = \langle \boldsymbol{a}, \boldsymbol{z}_n \rangle + b = e^2 \left[ \left( x_n - \frac{1}{2} \right) \left( 1 + ae^2 \right) \left( 1 + 2w|w| \right) + w|w|(1 + ae^2)| \right] + b.$$

**Loss.** Denote $\boldsymbol{\theta} \triangleq (e, w, a) \in \mathbb{R}^3$. Similar to the case without $a$ (Eq. (14) and Lemma 2), the cross-entropy loss in our setting can be compactly written as

$$L(\boldsymbol{\theta}, b) = \frac{1}{N} \sum_{n \in [N]} \mathbb{E}[\ell_{\log} \left( (2x_{n+1} - 1) \cdot \mathrm{logit}_n(\boldsymbol{\theta}, b) \right)] = \mathbb{E}_{X,Y} \left[ \ell_{\log} \left( (2Y - 1) \cdot \mathrm{logit}_X(\boldsymbol{\theta}, b) \right) \right],$$

$$\tag{41}$$

where $\text{logit}_X(\boldsymbol{\theta}, b) \triangleq e^2 \left[ \left( X - \frac{1}{2} \right) \left( 1 + ae^2 \right) (1 + 2w|w|) + w|w(1 + ae^2)| \right] + b, (X, Y) \in \{0, 1\}^2$
are distributed according to $(X, Y) \sim (\boldsymbol{\pi}, \boldsymbol{P})$, i.e. $X$ is a Bernoulli random variable with $X \sim \boldsymbol{\pi} \equiv$
$\text{Bern}(p/(p+q))$ and $Y|X \sim \boldsymbol{P}(p, q)$, the Markov kernel. Further, using the convexity of $b$ in $L(\cdot, b)$,
we can consider the optimal bias $b_\star(\boldsymbol{\theta}) = \text{argmin}_{b \in \mathbb{R}} L(\boldsymbol{\theta}, b)$ in Eq. (41) to obtain the loss $L(\boldsymbol{\theta})$:

$$
\begin{aligned}
L(\boldsymbol{\theta}) &\triangleq L(\boldsymbol{\theta}, b_\star) = \mathbb{E}_{X,Y} \left[ \ell_{\log} \left( (2Y - 1) \cdot \text{logit}_X(\boldsymbol{\theta}, b_\star) \right) \right] \\
&= \mathbb{E}_{X,Y} \left[ \ell_{\log} \left( (2Y - 1) \cdot \left( e^2 \left[ \left( X - \frac{1}{2} \right) \left( 1 + ae^2 \right) (1 + 2w|w|) + w|w(1 + ae^2)| \right] + b_\star \right) \right) \right].
\end{aligned}
\tag{42}
$$

We derive the expression for the optimal bias $b_\star$ in the proof of Lemma 13 below in App. N.

## G.2 Analysis of the loss function $L(\boldsymbol{\theta})$ from Eq. (42)

Now we establish the gradients of the loss function.

**Lemma 13** (Gradient computation and optimal bias). *For any $\boldsymbol{\theta} = (e, w, a) \in \mathbb{R}^3$ and the next-token prediction loss $L(\boldsymbol{\theta})$ in Eq. (42), the gradients are given by*

$$
\begin{aligned}
\frac{\partial L}{\partial e} &= -\mathbb{E} \left[ (f_1 X + f_2) \left( X - \frac{1}{2} \right) \right] \cdot 2e \left( 1 + ae^2 \right) (1 + 2w|w|) \\
&\quad - \mathbb{E} \left[ (f_1 X + f_2) \left( X - \frac{1}{2} \right) \right] \cdot 2e^3 a \left( 1 + 2w|w| \right), \\
\frac{\partial L}{\partial w} &= -\mathbb{E} \left[ (f_1 X + f_2) \left( X - \frac{1}{2} \right) \right] \cdot 2e^2 \left( 1 + ae^2 \right) \left( |w| + \text{sign}(w)\, w \right), \\
\frac{\partial L}{\partial a} &= -\mathbb{E} \left[ (f_1 X + f_2) \left( X - \frac{1}{2} \right) \right] \cdot e^4 \left( 1 + 2w|w| \right),
\end{aligned}
$$

*where $X \in \{0, 1\}$ is a Bernoulli random variable with $X \sim \text{Bern}(p/(p+q))$, and*

$$
\begin{aligned}
f_1 &\triangleq 1 - p - q - \phi_1 + \phi_0, \quad f_2 \triangleq p - \phi_0, \\
\phi_1 &\triangleq \sigma \left( e^2 \left( \frac{1}{2} \left( 1 + ae^2 \right) (1 + 2w|w|) + w|w(1 + ae^2)| \right) + b_\star \right), \\
\phi_0 &\triangleq \sigma \left( e^2 \left( \frac{-1}{2} \left( 1 + ae^2 \right) (1 + 2w|w|) + w|w(1 + ae^2)| \right) + b_\star \right),
\end{aligned}
$$

*where the optimal bias $b_\star$ is obtained by solving $\pi_1 f_1 + f_2 = 0$.*

*Proof.* We defer to App. N. $\qquad \square$

**Theorem 9** (All critical points for linear attention in $\mathbb{R}^4$). *Let the input sequence be $\{x_n\}_{n=1}^N \sim (\boldsymbol{\pi}, \boldsymbol{P})$, the transformer parameters $\boldsymbol{\theta} = (e, w, b, a) \in \mathbb{R}^4$, and the next-token prediction loss $L(\cdot)$ be as in Eq. (41). Then for any $(p, q) \in (0, 1)^2$ with $p + q \neq 1$ and $N \in \mathbb{N}$,*

(i) *the set of all global minima is given by*

$$
\boldsymbol{\Gamma}_\star(p, q) \triangleq \{(e, w, b, a) \in \mathbb{R}^4 : e^2 w|w(1 + ae^2)| + b = \frac{1}{2} \log \frac{p(1 - q)}{q(1 - p)},
\tag{43}
$$

$$
e^2 \left( 1 + ae^2 \right) (1 + 2w|w|) = \log \frac{(1 - q)(1 - p)}{pq} \}
\tag{44}
$$

(ii) *a set of local minima is given by*

$$
\boldsymbol{\Gamma}_{\min}(p, q) \triangleq \left\{ \boldsymbol{\gamma}_{\min} = (e, w, b, a) \in \mathbb{R}^4 : e = 0, (p + q - 1)(1 + 2w|w|) > 0, b = \log \frac{p}{q} \right\},
$$

(iii) *a set of saddle points is*

$$\boldsymbol{\Gamma}_{\text{sad}}(p, q) \triangleq \left\{\boldsymbol{\gamma}_{\text{sad}} = (e, w, b, a) \in \mathbb{R}^4 : e = 0, (p + q - 1)(1 + 2w|w|) \leq 0, b = \log\frac{p}{q}\right\}.$$

(iv) *a set of stationary points is*

$$\boldsymbol{\Gamma}_{\text{station}}(p, q) \triangleq \left\{\boldsymbol{\gamma}_{\text{station}} = (e, w, b, a) \in \mathbb{R}^4 : e \neq 0, 1 + ae^2 = 0, 1 + 2w|w| = 0, b = \log\frac{p}{q}\right\},$$

*Thus the set of all critical points is*

$$\left\{\boldsymbol{\theta} \in \mathbb{R}^2 : \nabla L(\boldsymbol{\theta}) = 0\right\} = \boldsymbol{\Gamma}_{\star} \cup \boldsymbol{\Gamma}_{\text{min}} \cup \boldsymbol{\Gamma}_{\text{sad}} \cup \boldsymbol{\Gamma}_{\text{station}}. \tag{45}$$

*In addition, for any $\boldsymbol{\theta}_{\star} \in \boldsymbol{\Gamma}_{\star}, \boldsymbol{\theta}_{\text{min}} \in \boldsymbol{\Gamma}_{\text{min}}$ and $\boldsymbol{\theta}_{\text{sad}} \in \boldsymbol{\Gamma}_{\text{sad}}$, the loss values satisfy*

$$H(x_{n+1} \mid x_n) = L(\boldsymbol{\theta}_{\star}) < L(\boldsymbol{\theta}_{\text{min}}) = L(\boldsymbol{\theta}_{\text{max}}) = L(\boldsymbol{\theta}_{\text{sad}}) = H(x_{n+1}).$$

**Theorem 10** (All critical points in $\mathbb{R}^3$). *Let the input sequence be $\{x_n\}_{n=1}^N \sim (\boldsymbol{\pi}, \boldsymbol{P})$, the transformer parameters $\boldsymbol{\theta} = (e, w, a) \in \mathbb{R}^3$, and the next-token prediction loss $L(\cdot)$ be as in Eq. (42). Then for any $(p, q) \in (0, 1)^2$ with $p + q \neq 1$ and $N \in \mathbb{N}$,*

(i) *the set of all global minima is given by*

$$\boldsymbol{\Theta}_{\star}(p, q) \triangleq \{(e, w, a) \in \mathbb{R}^3 : e^2 \left(1 + ae^2\right)(1 + 2w|w|) = \log\frac{(1 - q)(1 - p)}{pq}\} \tag{46}$$

(ii) *a set of local minima is given by*

$$\boldsymbol{\Theta}_{\text{min}}(p, q) \triangleq \left\{(e, w, a) \in \mathbb{R}^3 : e = 0, (p + q - 1)(1 + 2w|w|) > 0\right\},$$

(iii) *a set of local maxima is given by*

$$\boldsymbol{\Theta}_{\text{min}}(p, q) \triangleq \left\{(e, w, a) \in \mathbb{R}^3 : e = 0, (p + q - 1)(1 + 2w|w|) < 0\right\},$$

(iv) *a set of saddle points is*

$$\boldsymbol{\Theta}_{\text{sad}}(p, q) \triangleq \left\{(e, w, a) \in \mathbb{R}^3 : \left(0, -1/\sqrt{2}, a\right)\right\}.$$

*Defining a set of stationary points $\boldsymbol{\Theta}_{\text{station}}(p, q) \triangleq \left\{(e, w, a) \in \mathbb{R}^3 : e \neq 0, 1 + ae^2 = 0, 1 + 2w|w| = 0\right\}$, the set of all critical points is*

$$\left\{\boldsymbol{\theta} \in \mathbb{R}^2 : \nabla L(\boldsymbol{\theta}) = 0\right\} = \boldsymbol{\Theta}_{\star} \cup \boldsymbol{\Theta}_{\text{min}} \cup \boldsymbol{\Theta}_{\text{max}} \cup \boldsymbol{\Theta}_{\text{sad}} \cup \boldsymbol{\Theta}_{\text{station}}. \tag{47}$$

*In addition, for any $\boldsymbol{\theta}_{\star} \in \boldsymbol{\Theta}_{\star}, \boldsymbol{\theta}_{\text{min}} \in \boldsymbol{\Theta}_{\text{min}}$, and $\boldsymbol{\theta}_{\text{sad}} \in \boldsymbol{\Theta}_{\text{sad}}$, the loss values satisfy*

$$H(x_{n+1} \mid x_n) = L(\boldsymbol{\theta}_{\star}) < L(\boldsymbol{\theta}_{\text{min}}) = L(\boldsymbol{\theta}_{\text{max}}) = L(\boldsymbol{\theta}_{\text{sad}}) = H(x_{n+1}).$$

**Remark 5.** While the remaining set of stationary points could be classified to global minima, local minima, etc., it's technically unclear what category the set of critical points $\boldsymbol{\Theta}_{\text{station}}(p, q)$ belong to, as the Hessian is undefined here. We would need to rely on local perturbation analysis to characterize these class of points.

We defer the proofs of the theorems to App. N.2.

### G.3 Gradient flow analysis

Analogous to the gradient flow analysis for $\boldsymbol{\theta} = (e, w) \in \mathbb{R}^2$ in App. F, we now study its countepart toegether with the attention scalar, i.e. $\boldsymbol{\theta} = (e, w, a) \in \mathbb{R}^3$. To this end, let $(\boldsymbol{\theta}_t)_{t \geq 0}$ be a $C^1$ curve in $\mathbb{R}^3$ governed by

$$\frac{\mathrm{d}\boldsymbol{\theta}_t}{\mathrm{d}t} = -\nabla L(\boldsymbol{\theta}_t), \quad \boldsymbol{\theta}_t = (e_t, w_t, a_t) \in \mathbb{R}^3, t \geq 0, \qquad \text{(GF-Attention)}$$

starting with a randomly initalized $\boldsymbol{\theta}_0$. We define the *energy function* $\mathcal{E}(\cdot, \cdot, \cdot)$ as

$$\mathcal{E}(e, w, a) \triangleq e^2 - (w^2 + \text{sign}(w) \cdot \log|w|) - 2a^2, \quad \forall (e, w, a) \in \mathbb{R}^3 \setminus \text{ea-plane}, \tag{48}$$

where ea-plane $\triangleq \{(e, w = 0, a)\}$. The following lemma presents the crucial result that the energy is constant along the flow in GF-Attention.

**Lemma 14** (Constant energy along the flow). *For any $(p, q) \in (0, 1)^2$ and initialization $\boldsymbol{\theta}_0 = (e_0, w_0, a_0) \in \mathbb{R}^3$, let $(\boldsymbol{\theta}_t)_{t \geq 0}$ be the corresponding* GF-Attention *trajectory starting from $\boldsymbol{\theta}_0$. If $w_0 \neq 0$, then the energy stays constant along the trajectory, i.e.*

$$\mathcal{E}(\boldsymbol{\theta}_t) = e_t^2 - (w_t^2 + \mathrm{sign}(w_t) \cdot \log |w_t|) - 2a_t^2 = \mathcal{E}(\boldsymbol{\theta}_0), \quad \forall t \geq 0. \tag{49}$$

*On the other hand, if $w_0 = 0$, $w_t = 0$ for all $t \geq 0$. Hence, if we initialize on* ea-plane *the trajectory always stays on the* ea-plane.

Now we characterize the convergence of the gradient flow.

**Lemma 15** (GF convergence). *Let $(\boldsymbol{\theta}_t)_{t \geq 0}$ be a continuously diferentiable* GF-Attention *trajectory starting from $\boldsymbol{\theta}_0$. Then for all initializations $\boldsymbol{\theta}_0 \in \mathbb{R}^3$,*

(i) *$(\boldsymbol{\theta}_t)_{t \geq 0}$ is bounded,*

(ii) *there exists a $\boldsymbol{\theta}_{\lim} \in \mathbb{R}^3$ such that $\lim_{t \to \infty} \boldsymbol{\theta}_t = \boldsymbol{\theta}_{\lim}$ and*

(iii) *$\lim_{t \to \infty} \|\nabla L(\boldsymbol{\theta}_t)\| = \|\nabla L(\boldsymbol{\theta}_{\lim})\| = 0$.*

*Hence $\boldsymbol{\theta}_{\lim}$ is a critical point of $L$.*

The following result characterizes the energy of the limit point.

**Lemma 16** (Energy at the limit point). *Consider the same setting as in Lemma 15. If $\boldsymbol{\theta}_0 \in \mathbb{R}^3 \setminus$ ea-plane, then $\mathcal{E}(\boldsymbol{\theta}_{\lim}) = \mathcal{E}(\boldsymbol{\theta}_0)$. Hence $\boldsymbol{\theta}_{\lim}$ lies at the intersection of the contour line $\mathcal{E}(e, w) = \mathcal{E}_0$ with the set of critical points of $L$ in $\mathbb{R}^3$.*

*On the other hand, if $\boldsymbol{\theta}_0 \in$ ea-plane, then $\boldsymbol{\theta}_{\lim} \in$ ea-plane.*

We defer the proofs of the lemmas to App. N.

### G.4 Role of standard initialization

The following picture enhances the behavior of the GF dynamics near the origin.

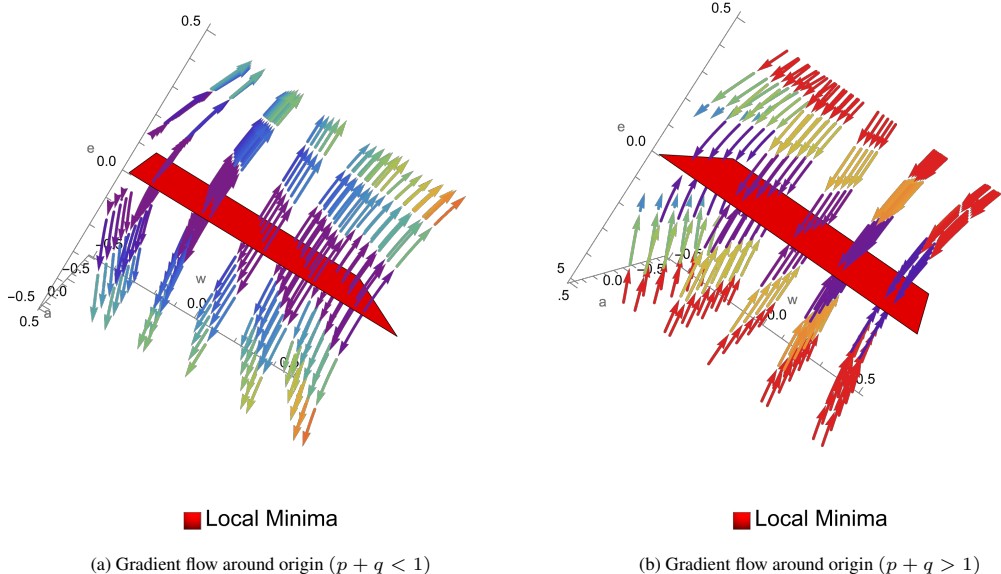

(a) Gradient flow around origin $(p + q < 1)$        (b) Gradient flow around origin $(p + q > 1)$

Figure 5: Gradient flow dynamics in $\mathbb{R}^3$, near the origin, for the transformer parameters with attention scalar $a$ (Sec. 4.1). The local minima are repellors for $p + q < 1$, while attracting for $p + q > 1$.

Based on the above theoretical results and empirical evidence, we conjecture the following theorem, whose informal proof we defer to App. N.5.

**Theorem 11** ( [Informal] Role of standard initialization for $p + q > 1$). *If we use the standard initialization $\boldsymbol{\theta}_0 \sim \mathcal{N}(0, \sigma^2 \boldsymbol{I}_3)$ with $\sigma^2 \ll 1/\sqrt{2}$ for the* GF-Attention, *$\boldsymbol{\theta}_{\lim}$ will be a local minimum with high probability.*

# H    Additional empirical results

## H.1    Gaussian initialization converges to low-rank

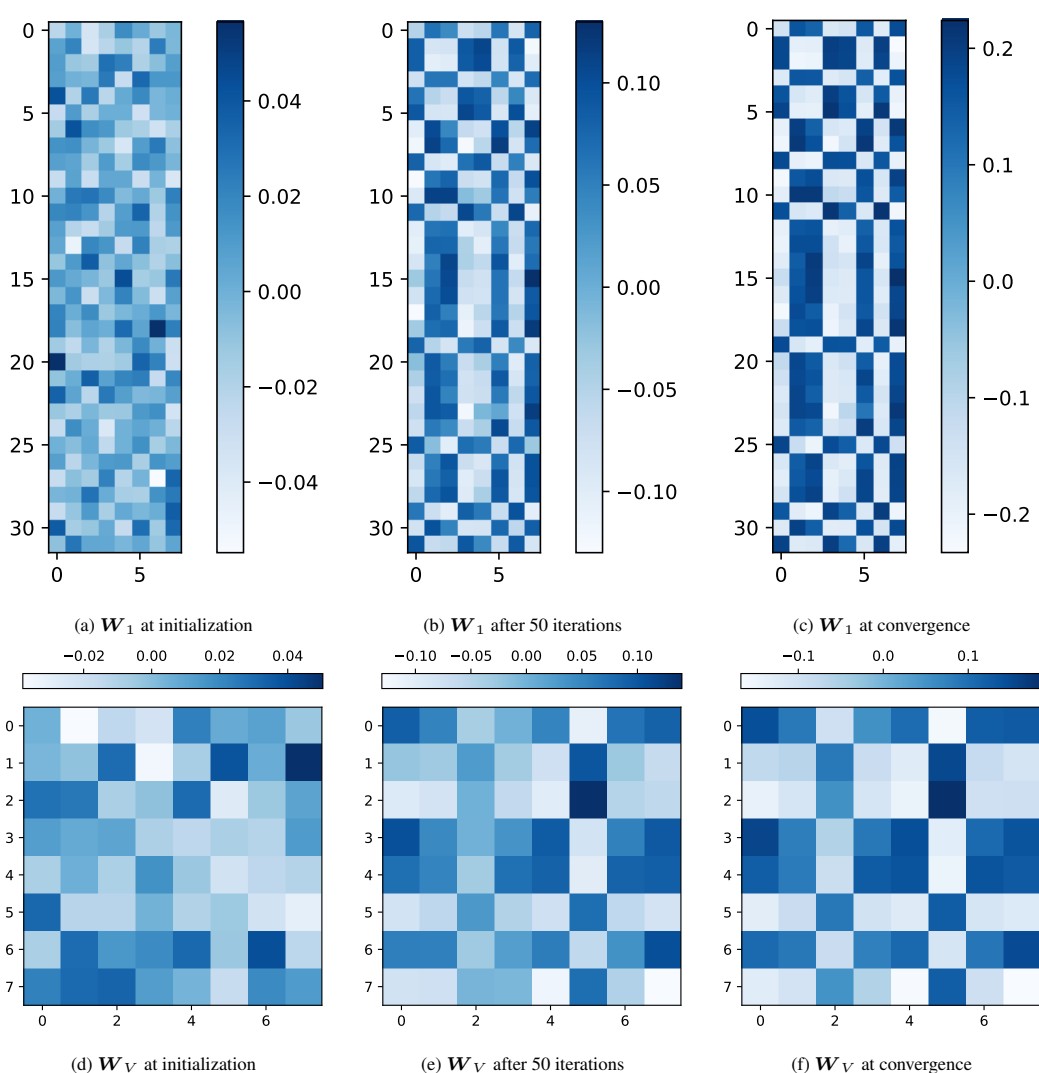

Figure 6: Evolution of parameters $W_1$ and $W_V$ across iterations, starting from a standard Gaussian initialization. At convergence, all the parameter matrices are approximately rank-one.

## H.2    Low-rank initialization stays low-rank

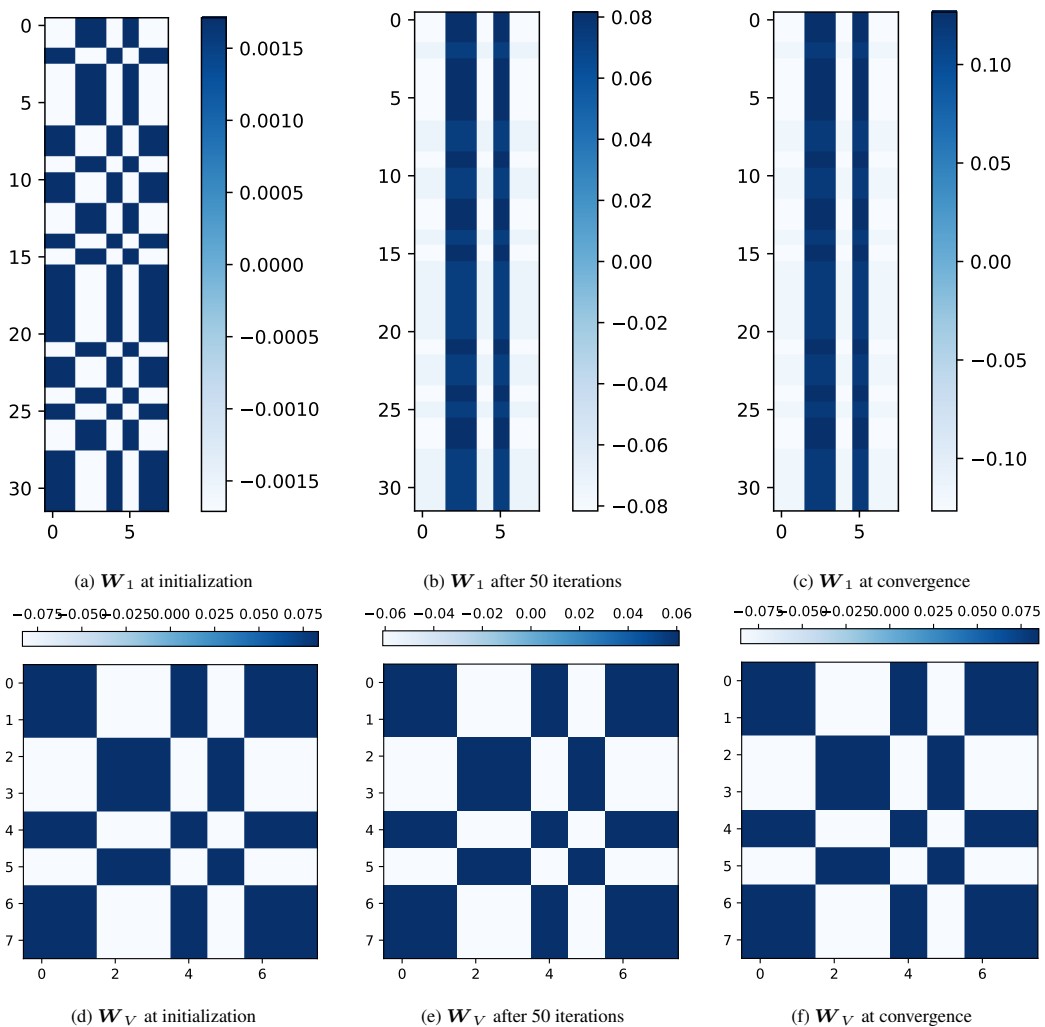

Figure 7: Evolution of parameters $\boldsymbol{W}_1$ and $\boldsymbol{W}_V$ across iterations, starting from a rank-one initialization. The parameters maintain a rank-one structure across the entire training.

# I  Model architecture and hyper-parameters

Table 1: Parameters in the transformer architecture with their shape.

| Parameter | Matrix shape |
|---|---|
| transformer.wte | $2 \times d$ |
| transformer.wpe | $N \times d$ |
| transformer.h.ln_1 | $d \times 1$ |
| transformer.h.attn.c_attn | $3d \times d$ |
| transformer.h.attn.c_proj | $d \times d$ |
| transformer.h.ln_2 | $d \times 1$ |
| transformer.h.mlp.c_fc | $4d \times d$ |
| transformer.h.mlp.c_proj | $d \times 4d$ |
| transformer.ln_f | $d \times 1$ |

Table 2: Settings and parameters for the transformer model used in the experiments.

| | |
|---|---|
| Dataset | $k$-th order binary Markov source |
| Architecture | Based on the GPT-2 architecture as implemented in [26] |
| Batch size | Grid-searched in $\{16, 50\}$ |
| Accumulation steps | 1 |
| Optimizer | AdamW ($\beta_1 = 0.9, \beta_2 = 0.95$) |
| Learning rate | 0.001 |
| Scheduler | Cosine |
| # Iterations | 8000 |
| Weight decay | $1 \times 10^{-3}$ |
| Dropout | 0 |
| Sequence length | Grid-searched in $\{512, 1024, 2048\}$ |
| Embedding dimension | Grid-searched in $\{4, 8, 16, 32, 64\}$ |
| Transformer layers | 1 |
| Attention heads | 1 |
| Repetitions | 3 or 5 |

# J Proofs of theorems in App. D

## J.1 Proof of Thm. 7

*Proof.* We characterize the set of global minima, local minima, and that of the saddle points individually.

**(i) Set of all global minima.** Let $\gamma_\star \in \mathbb{R}^3$ be arbitrary. From [23, Lemma 1], we have that $\gamma_\star$ is a global minimum for the loss $L(\cdot)$ in Eq. (23) if and only if its prediction probability satisfies $f_{\gamma_\star}(x_1^n) = \mathbb{P}(x_{n+1} = 1 \mid x_n)$, the Markov kernel. Since the input $\{x_n\}_{n=1}^N \sim (\boldsymbol{\pi}(p,q), \boldsymbol{P}(p,q))$, we have that

$$\mathbb{P}(x_{n+1} = 1 \mid x_n) = (1 - x_n)p + x_n(1 - q) = (1 - p - q)x_n + p. \tag{50}$$

On the other hand, by definition, from Eq. (3), $f_{\gamma_\star}(x_1^n) = \sigma\left(e^2(1 + 2w|w|)x_n + b - \frac{e^2}{2}\right)$, where $\gamma_\star = (e, w, b)$. Since $x_n \in \{0, 1\}$, this can be further simplified to

$$\begin{aligned}
f_{\gamma_\star}(x_1^n) &= \sigma\left(e^2(1 + 2w|w|)x_n + b - \frac{e^2}{2}\right) \\
&= x_n \cdot \sigma\left(e^2(1 + 2w|w|) + b - \frac{e^2}{2}\right) + (1 - x_n) \cdot \sigma\left(b - \frac{e^2}{2}\right) \\
&= x_n\left(\sigma\left(2e^2w|w| + b + \frac{e^2}{2}\right) - \sigma\left(b - \frac{e^2}{2}\right)\right) + \sigma\left(b - \frac{e^2}{2}\right).
\end{aligned} \tag{51}$$

Since both $f_{\gamma_\star}(x_1^n)$ and $\mathbb{P}(x_{n+1} = 1 \mid x_n)$ are linear functions of $x_n$, equating them for all values of $x_n \in \{0, 1\}$ implies that the respective coeffecients in these functions in Eq. (50) and Eq. (51) are also equal, i.e.

$$\sigma\left(b - \frac{e^2}{2}\right) = p,$$

$$\sigma\left(2e^2w|w| + b + \frac{e^2}{2}\right) - \sigma\left(b - \frac{e^2}{2}\right) = 1 - p - q,$$

and hence

$$\sigma\left(b - \frac{e^2}{2}\right) = p, \quad \sigma\left(2e^2w|w|\right) + b + \frac{e^2}{2} = 1 - q. \tag{52}$$

Since $\sigma(z) = y$ for $y \in (0, 1)$ implies $z = \log\frac{y}{1-y}$, Eq. (52) can be rewritten as

$$b - \frac{e^2}{2} = \log\frac{p}{1 - p}, \quad 2e^2w|w| + b + \frac{e^2}{2} = \log\frac{1 - q}{q}.$$

Using $2e^2w|w| + b + \frac{e^2}{2} = e^2(1 + 2w|w|) + b - \frac{e^2}{2} = e^2(1 + 2w|w|) + \log\frac{p}{1-p}$ in the second equality above, we obtain

$$\begin{aligned}
b - \frac{e^2}{2} &= \log\frac{p}{1 - p}, \\
e^2(1 + 2w|w|) &= \log\frac{1 - q}{q} + \log\frac{1 - p}{p} = \log\frac{(1 - p)(1 - q)}{pq}.
\end{aligned} \tag{53}$$

Thus $\gamma_\star \in \mathbb{R}^3$ is a global minimum for $L(\cdot)$ if and only if it satisfies Eq. (53) (note that it's already a critical point, as established in Thm. 6). Thus, the set of all global minimum $\Gamma_\star(p, q)$ is given by

$$\Gamma_\star(p, q) \triangleq \left\{\gamma_\star = (e, w, b) \in \mathbb{R}^3 : e^2(1 + 2w|w|) = \log\frac{(1 - p)(1 - q)}{pq}, \ b - \frac{e^2}{2} = \log\frac{p}{1 - p}\right\}.$$

Since the prediction $f_{\gamma_\star}(\cdot)$ equals the Markov kernel for any $\gamma_\star \in \Gamma_\star$, it follows from Thm. 4 (or [23, Lemma 1]) that $L(\gamma_\star) = H(x_{n+1} \mid x_n)$, the entropy rate of the Markov chain.

### (ii) Set of local minima and saddle points.

Define $\boldsymbol{\Gamma}_{\min}(p, q) \subseteq \mathbb{R}^3$ and $\boldsymbol{\Gamma}_{\text{sad}} \subseteq \mathbb{R}^3$ as follows:

$$\boldsymbol{\Gamma}_{\min}(p, q) \triangleq \left\{ \boldsymbol{\gamma}_{\min} = (e, w, b) \in \mathbb{R}^3 : e = 0, (p + q - 1)(1 + 2w|w|) > 0, b = \log \frac{p}{q} \right\},$$

$$\boldsymbol{\Gamma}_{\text{sad}}(p, q) \triangleq \left\{ \boldsymbol{\gamma}_{\text{sad}} = (e, w, b) \in \mathbb{R}^3 : e = 0, (p + q - 1)(1 + 2w|w|) \leq 0, b = \log \frac{p}{q} \right\}.$$

To show that $\boldsymbol{\Gamma}_{\min}$ is the set of all bad local minima for $L(\cdot)$, we first show that any $\boldsymbol{\gamma}_{\min} \in \boldsymbol{\Gamma}_{\min}$ is a bad local minimum and then show that every bad local minimum should belong to $\boldsymbol{\Gamma}_{\min}$. Similarly for $\boldsymbol{\Gamma}_{\text{sad}}$. We start with the local minima.

Let $\boldsymbol{\gamma}_{\min} = (e, w, b) \in \boldsymbol{\Gamma}_{\min}$. Recall that $\boldsymbol{\gamma}_{\min}$ is a stationary point (Thm. 6), i.e.

$$\nabla L(\boldsymbol{\gamma}_{\min}) = 0.$$

Rearragning the order of scalars and writing $\boldsymbol{\gamma}_{\min} = (b, e, w)$, from Lemma 4, the Hessian of the loss at $\boldsymbol{\gamma}_{\min}$ is

$$\nabla^2 L(\boldsymbol{\gamma}_{\min}) = \pi_0 \pi_1 \begin{bmatrix} 1 & 0 & 0 \\ 0 & 2(p + q - 1)(1 + 2w|w|) & 0 \\ 0 & 0 & 0 \end{bmatrix}. \tag{54}$$

By definition, $\boldsymbol{\gamma}_{\min} = (b, e, w)$ satisfies $(p + q - 1)(1 + 2w|w|) > 0$. Thus its Hessian in Eq. (54) has a block diagonal structure of the form $\begin{bmatrix} \boldsymbol{H}_{b,e} & 0 \\ 0 & 0 \end{bmatrix}$ where $\boldsymbol{H}_{b,e}$ has both the eigen values positive, and hence positive-definite. In other words, $\boldsymbol{\gamma}_{\min}$ is a local minimum for $L(\cdot)$ in the $(b, e) \in \mathbb{R}^2$ space for any fixed $w$ in the set. Interestingly, using the continuity argument and the fact that $L(b = \log \frac{p}{q}, e = 0, w)$ is constant in $w \in \mathbb{R}$, we can essentially follow the same steps as in proof of Theorem 2 in [23, Appendix B.3] (Thm. 5 above) to show that $\boldsymbol{\gamma}_{\min} = (b, e, w)$ is a also local minimum for $L(\cdot)$ in the full parameter space $\mathbb{R}^3$. This establishes that $\boldsymbol{\gamma}_{\min}$ is a local minimum for $L(\cdot)$.

For the saddle points, let $\boldsymbol{\gamma}_{\text{sad}} = (e, w, b) \in \boldsymbol{\Gamma}_{\text{sad}}$. We have that $\boldsymbol{\gamma}_{\text{sad}}$ is a stationary point (Thm. 6) and Lemma 4 implies its Hessian (after rearraging the order of scalars as above with $\boldsymbol{\gamma}_{\text{sad}} = (b, e, w)$) is:

$$\nabla^2 L(\boldsymbol{\gamma}_{\text{sad}}) = \pi_0 \pi_1 \begin{bmatrix} 1 & 0 & 0 \\ 0 & 2(p + q - 1)(1 + 2w|w|) & 0 \\ 0 & 0 & 0 \end{bmatrix}. \tag{55}$$

If $w \neq -\frac{1}{\sqrt{2}}$, $(p + q - 1)(1 + 2w|w|) < 0$ for any $\boldsymbol{\gamma}_{\text{sad}} \in \boldsymbol{\Gamma}_{\text{sad}}$, and hence the Hessian $\nabla^2 L(\boldsymbol{\gamma}_{\text{sad}})$ in Eq. (55) as both positive, negative, and zero eigen values. Thus $\boldsymbol{\gamma}_{\text{sad}}$ is a saddle point for $L(\cdot)$. Using a neighborhood argument, we can similarly argue for $w = \frac{1}{\sqrt{2}}$ to establish that it's also a saddle point. Now we compute the loss value.

For any $\boldsymbol{\gamma}_{\min} = (e, w, b) \in \boldsymbol{\Gamma}_{\min}$ or $\boldsymbol{\gamma}_{\text{sad}} = (e, w, b) \in \boldsymbol{\Gamma}_{\text{sad}}$, we have that $e = 0$ and $b = \log \frac{p}{q}$. Thus for $\boldsymbol{\gamma} = \boldsymbol{\gamma}_{\min}$ or $\boldsymbol{\Gamma}_{\text{sad}}$, the prediction probability in view of Eq. (3) is

$$\mathbb{P}_{\boldsymbol{\gamma}}(x_{n+1} = 1 \mid x_1^n) = \sigma\left(e^2(1 + 2w|w|)x_n + b - \frac{e^2}{2}\right) = \sigma(b) = \frac{p}{p + q} = \mathbb{P}(x_{n+1} = 1),$$

the marginal distribution. Substituting this equality in the definition of cross-entropy loss $L(\cdot)$ in Eq. (1) and the fact that $\mathbb{P}(x_{n+1} = 1) = \frac{p}{p+q} = \pi_1$, following the same steps as in [23, Appendix B.3], we obtain

$$L(\boldsymbol{\gamma}) = -\frac{1}{N} \sum_{n \in [N]} \mathbb{E}_{x_1^{n+1}}[x_{n+1} \cdot \log f_{\boldsymbol{\gamma}}(x_1^n) + (1 - x_{n+1}) \cdot \log(1 - f_{\boldsymbol{\gamma}}(x_1^n))]$$

$$= -\frac{1}{N} \sum_{n \in [N]} \mathbb{E}_{x_1^{n+1}}[x_{n+1} \cdot \log \pi_1 + (1 - x_{n+1}) \cdot \log \pi_0]$$

$$= \frac{1}{N} \sum_{n \in [N]} \left[ -\pi_1 \log \pi_1 - \pi_0 \log \pi_0 \right]$$

$$= H(\boldsymbol{\pi}) = H(x_{n+1}).$$

Thus $L(\boldsymbol{\gamma}_{\mathrm{min}}) = L(\boldsymbol{\gamma}_{\mathrm{sad}}) = H(x_{n+1})$. To see that $H(x_{n+1} \mid x_n) = L(\boldsymbol{\gamma}_\star) < L(\boldsymbol{\gamma}_{\mathrm{min}}) = L(\boldsymbol{\gamma}_{\mathrm{sad}}) = H(x_{n+1})$ for any global minimum $\boldsymbol{\gamma}_\star$, observe that the gap

$$L(\boldsymbol{\gamma}_{\mathrm{min}}) - L_\star = H(x_{n+1}) - H(x_{n+1}|x_n) = I(x_n; x_{n+1}) \geq 0,$$

where $I(x_n; x_{n+1})$ is the mutual information between $x_n$ and $x_{n+1}$ [11]. Hence the optimality gap equals zero if and only if the mutual information equals zero, which happens when $x_n$ and $x_{n+1}$ are independent, i.e. $\mathbb{P}\left(x_{n+1} = 1 \mid x_n\right)$ doesn't depend on $x_n$. Since $\mathbb{P}\left(x_{n+1} = 1 \mid x_n\right) = (1 - p - q)x_n + p$ from Eq. (50), this happens only when $p + q = 1$ which contradicts the theorem assumption that $p + q \neq 1$. Hence $H(x_{n+1} \mid x_n) = L(\boldsymbol{\gamma}_\star) < L(\boldsymbol{\gamma}_{\mathrm{min}}) = L(\boldsymbol{\gamma}_{\mathrm{sad}}) = H(x_{n+1})$.

Now we finally show that $\boldsymbol{\Gamma}_{\mathrm{min}}$ and $\boldsymbol{\Gamma}_{\mathrm{sad}}$ are the only set of bad local minima and saddle points respectively. Let $\boldsymbol{\gamma}$ is a bad local minimum for $L(\cdot)$. By definition, it's also a critical point. Recall from Thm. 6 that any stationary point $\boldsymbol{\gamma} = (e, w, b)$ for the loss $L(\cdot)$ satisfies that either $\boldsymbol{\gamma} \in \boldsymbol{\Gamma}_\star$, $\boldsymbol{\gamma} \in \boldsymbol{\Gamma}_{\mathrm{min}}$, or $\boldsymbol{\gamma} \in \boldsymbol{\Gamma}_{\mathrm{sad}}$. Clearly $\boldsymbol{\gamma} \notin \boldsymbol{\Gamma}_\star$, as $\boldsymbol{\Gamma}_\star$ is the set of all global minima. Similarly, $\boldsymbol{\gamma} \notin \boldsymbol{\Gamma}_{\mathrm{sad}}$ as every point in $\boldsymbol{\Gamma}_{\mathrm{sad}}$ is a saddle point for the loss $L(\cdot)$ as established above. Hence $\boldsymbol{\gamma} \in \boldsymbol{\Gamma}_{\mathrm{min}}$. Thus every bad local minimum in $\mathbb{R}^3$ belongs to $\boldsymbol{\Gamma}_{\mathrm{min}}$. This coupled with the fact above that $\boldsymbol{\Gamma}_{\mathrm{min}}$ is a set of bad local minima implies $\boldsymbol{\Gamma}_{\mathrm{min}}$ is indeed the set of all bad local minima. The proof for $\boldsymbol{\Gamma}_{\mathrm{sad}}$ is similar.

$\square$

## J.2 Proof of Thm. 6

*Proof.* Let $\boldsymbol{\gamma} = (e, w, b) \in \mathbb{R}^3$ be such that $\nabla L(\boldsymbol{\gamma}) = \left( \frac{\partial L}{\partial e}, \frac{\partial L}{\partial w}, \frac{\partial L}{\partial b} \right)^\top = 0$. By Lemma 3, we have

$$\frac{\partial L}{\partial e} = \mathbb{E}_X \left[ (f_1 X + f_2)(2X(1 + 2w|w|) - 1)) \right] \cdot e = 0, \tag{56}$$

$$\frac{\partial L}{\partial w} = \mathbb{E}_X \left[ (f_1 X + f_2)X \right] \cdot 4e^2|w| = 0, \tag{57}$$

$$\frac{\partial L}{\partial b} = \mathbb{E}_X \left[ f_1 X + f_2 \right] = 0, \tag{58}$$

where $X \sim \mathrm{Bern}(p/(p+q))$, $f_1 = \sigma \left( 2e^2 w|w| + b + \frac{e^2}{2} \right) + q - 1 - \sigma \left( b - \frac{e^2}{2} \right) + p$, and $f_2 = \sigma \left( b - \frac{e^2}{2} \right) - p$. Our goal is to now show that Eqs. (56)-(58) hold only if either $(e = 0, b = \log \frac{p}{q})$ or $(f_1 = 0, f_2 = 0)$. We consider two cases corresponding to $e = 0$ and $e \neq 0$.

(i): $e = 0$. If $e = 0$, we readily see that $\frac{\partial L}{\partial e} = \frac{\partial L}{\partial e} = 0$. Further, $f_1 = p + q - 1$ and $f_2 = \sigma(b) - p$. Hence, Eq. (58) implies that

$$\mathbb{E}_X \left[ f_1 X + f_2 \right] = (p + q - 1)\mathbb{E}[X] + \sigma(b) - p = (p + q - 1)\frac{p}{p + q} + \sigma(b) - p = \sigma(b) - \frac{p}{p + q} = 0,$$

which implies that $b = \log \frac{p}{q}$. Since $w \in \mathbb{R}$ is arbitrary, we see in this case that

$$\boldsymbol{\gamma} \in \left\{ (e, w, b) \in \mathbb{R}^3 : e = 0, (p + q - 1)(1 + 2w|w|) > 0, b = \log \frac{p}{q} \right\} \bigcup$$

$$\left\{ (e, w, b) \in \mathbb{R}^3 : e = 0, (p + q - 1)(1 + 2w|w|) < 0, b = \log \frac{p}{q} \right\},$$

$$= \boldsymbol{\Gamma}_{\mathrm{min}} \cup \boldsymbol{\Gamma}_{\mathrm{sad}}.$$

(ii): $e \neq 0$. Suppose $e \neq 0$. Here we show that $f_1 = f_2 = 0$ and hence $\boldsymbol{\gamma} \in \boldsymbol{\Gamma}_\star$. We consider two cases corresponding to $w \neq 0$ and $w = 0$. Let $w \neq 0$. Since both $e \neq 0$ and $w \neq 0$, and $X = X^2$ in Eq. (57) with $\mathbb{E}[X] = \pi_1 = \frac{p}{p+q} > 0$, we obtain that

$$\mathbb{E}_X[f_1 X^2 + f_2 X] = (f_1 + f_2)\mathbb{E}[X] = (f_1 + f_2)\pi_1 = 0,$$

and hence $f_1 + f_2 = 0$. Further Eq. (58) implies that $f_1\pi_1 + f_2 = 0$. Together, this implies $f_1 = f_2 = 0$.

Now suppose $w = 0$. From Eq. (58), we have that $\mathbb{E}[f_1 X + f_2] = f_1\pi_1 + f_2 = 0$. Since $e \neq 0$, Eq. (56) yields

$$\mathbb{E}_X \left[ (f_1 X + f_2)(2X - 1) \right] = 2\mathbb{E}_X \left[ (f_1 X + f_2)X \right] = 2\pi_1(f_1 + f_2) = 0.$$

So $f_1 = f_2 = 0$ in this case too. Thus we have showed that whenever $e \neq 0$, we have $f_1 = f_2 = 0$. Recalling the expressions for $f_1$ and $f_2$,

$$f_2 = \sigma\left(b - \frac{e^2}{2}\right) - p = 0 \Rightarrow b - \frac{e^2}{2} = \log\frac{p}{1-p}, \tag{59}$$

$$f_1 = \sigma\left(2e^2w|w| + b + \frac{e^2}{2}\right) + q - 1 - \sigma\left(b - \frac{e^2}{2}\right) + p = \sigma\left(2e^2w|w| + b + \frac{e^2}{2}\right) + q - 1 = 0,$$

and hence,

$$2e^2w|w| + b + \frac{e^2}{2} = \log\frac{1-q}{q}.$$

Substituting $b - \frac{e^2}{2} = \log\frac{p}{1-p}$ in the above equation,

$$2e^2w|w| + e^2 + b - \frac{e^2}{2} = 2e^2w|w| + e^2 + \log\frac{p}{1-p} = \log\frac{1-q}{q},$$

and thus,

$$e^2(1 + 2w|w|) = \log\frac{(1-p)(1-q)}{pq}. \tag{60}$$

In view of Eq. (59) and Eq. (60), we have that $\gamma = (e, w, b) \in \Gamma_\star$.

Together, we have shown that whenever $\nabla L(\gamma) = 0$, we have $\gamma \in \Gamma_\star \cup \Gamma_{\min} \cup \Gamma_{\text{sad}}$. Since $\Gamma_\star \cup \Gamma_{\min} \cup \Gamma_{\text{sad}} \subseteq \{\gamma : \nabla L(\gamma) = 0\}$, we are done. $\qquad\square$

# K   Proofs of technical lemmas in App. D

## K.1   Proof of Lemma 2

*Proof.* Recall from Eq. (14) that the cross-entropy loss $L(\cdot)$ is defined as

$$L(\gamma) = -\frac{1}{N}\sum_{n\in[N]} \mathbb{E}_{x_1^{n+1}}[x_{n+1} \cdot \log f_\gamma(x_1^n) + (1 - x_{n+1}) \cdot \log(1 - f_\gamma(x_1^n))], \tag{61}$$

where $f_\gamma(x_1^n) = \sigma(\text{logit}_n) = \sigma\left(e^2(1 + 2w|w|)x_n + b - \frac{e^2}{2}\right)$ from Eq. (3). For any $Y \in \{0, 1\}, Z \in \mathbb{R}$, using the fact that $1 - \sigma(Z) = \sigma(-Z)$, we have

$$-[Y\log\sigma(Z) + (1 - Y)\log(1 - \sigma(Z))] = -[Y\log\sigma(Z) + (1 - Y)\log\sigma(-Z)]$$

$$\stackrel{(a)}{=} Y\log(1 + \exp(-Z)) + (1 - Y)\log(1 + \exp(Z))$$

$$= \log(1 + \exp(-(2Y - 1)Z))$$

$$\stackrel{(b)}{=} \ell_{\log}((2Y - 1)Z), \tag{62}$$

where $(a)$ and $(b)$ follow from the definitions of sigmoid and the logistic functions: $\sigma(z) = \frac{1}{1+\exp(-z)}$, $\ell_{\log}(z) = \log(1 + \exp(-z))$ for $z \in \mathbb{R}$. Substituting $Y = x_{n+1} \in \{0, 1\}$ and $Z = \text{logit}_n \in \mathbb{R}$ in Eq. (62), Eq. (61) simplifies to

$$L(\gamma) = \frac{1}{N}\sum_{n\in[N]} \mathbb{E}[\ell_{\log}((2x_{n+1} - 1) \cdot \text{logit}_n)].$$

Since $\text{logit}_n$ is only a function of $x_n$, the above expectation is over the distribution of the pairs $(x_n, x_{n+1})$, which for all $n \in [N]$ have the same law as a pair of random variables $(X, Y)$ with $X \sim \boldsymbol{\pi} \equiv \text{Bern}(p/(p+q))$ and $Y|X \sim \boldsymbol{P}(p,q)$, the Markov kernel. Hence the above equality can be rewritten using the definition of $\text{logit}_n$ as

$$L(\boldsymbol{\gamma}) = \frac{1}{N} \sum_{n \in [N]} \mathbb{E}[\ell_{\log}((2x_{n+1}-1) \cdot \text{logit}_n)] = \mathbb{E}_{X,Y}\left[\ell_{\log}\left((2Y-1)\left(e^2(1+2w|w|)X + b - \frac{e^2}{2}\right)\right)\right].$$

$\square$

### K.2 Proof of Lemma 3

*Proof.* With $\boldsymbol{\gamma} = (e, w, b)$ and $\theta$ denoting either of the scalars $e, w,$ or $b$, we have from [23, Lemma 2] that the gradient of the loss $L(\cdot)$ is given by

$$\nabla_\theta L(\boldsymbol{\gamma}) = -\frac{1}{N} \sum_{n \in [N]} \mathbb{E}_{x_1^{n+1}}\left[(x_{n+1} - f_\theta(x_1^n)) \cdot \nabla_\theta \text{logit}_n\right], \tag{63}$$

where $f_\gamma(x_1^n) = \sigma(\text{logit}_n) = \sigma\left(e^2(1+2w|w|)x_n + b - \frac{e^2}{2}\right)$. Using the same argument as in the proof of Lemma 6, we can replace the expecatations in Eq. (63) with that of a pair of random variables $(X, Y)$ with $X \sim \boldsymbol{\pi} \equiv \text{Bern}(p/(p+q))$ and $Y|X \sim \boldsymbol{P}(p,q)$, the Markov kernel. That is,

$$\nabla_\theta L(\boldsymbol{\gamma}) = -\mathbb{E}_{X,Y}\left[\left(Y - \sigma\left(e^2(1+2w|w|)X + b - \frac{e^2}{2}\right)\right) \cdot \nabla_\theta\left(e^2(1+2w|w|)X + b - \frac{e^2}{2}\right)\right]. \tag{64}$$

Now we define the error term $\mathcal{E}(X,Y) \triangleq -\left(Y - \sigma\left(e^2(1+2w|w|)X + b - \frac{e^2}{2}\right)\right)$. Our goal is to show that $\mathbb{E}[\mathcal{E}(X,Y) \mid X] = f_1 X + f_2$, where $f_1 \triangleq \sigma\left(2e^2w|w| + b + \frac{e^2}{2}\right) + q - 1 - \sigma\left(b - \frac{e^2}{2}\right) + p$, and $f_2 \triangleq \sigma\left(b - \frac{e^2}{2}\right) - p$, which suffices to prove the lemma. To this end, using the fact that $X \in \{0, 1\}$, we have

$$\mathcal{E}(X,Y) = -\left(Y - \sigma\left(e^2(1+2w|w|)X + b - \frac{e^2}{2}\right)\right)$$

$$= -\left(Y - X \cdot \sigma\left(2e^2w|w| + b + \frac{e^2}{2}\right) - (1-X) \cdot \sigma\left(b - \frac{e^2}{2}\right)\right)$$

$$= -Y + X\left(\sigma\left(2e^2w|w| + b + \frac{e^2}{2}\right) - \sigma\left(b - \frac{e^2}{2}\right)\right) + \sigma\left(b - \frac{e^2}{2}\right).$$

Now taking the conditional expectation with respect to $X$ and using the fact that $\mathbb{E}[Y|X] = \mathbb{P}(Y=1 \mid X) = (1-p-q)X + p$ (since $Y|X \sim \boldsymbol{P}(p,q)$), we have

$$\mathbb{E}[\mathcal{E}(X,Y) \mid X] = -(1-p-q)X - p + X\left(\sigma\left(2e^2w|w| + b + \frac{e^2}{2}\right) - \sigma\left(b - \frac{e^2}{2}\right)\right) + \sigma\left(b - \frac{e^2}{2}\right)$$

$$= X\left(\sigma\left(2e^2w|w| + b + \frac{e^2}{2}\right) + q - 1 - \sigma\left(b - \frac{e^2}{2}\right) + p\right) + \sigma\left(b - \frac{e^2}{2}\right) - p$$

$$\overset{(a)}{=} f_1 X + f_2,$$

where $(a)$ follows from the definition of $f_1$ and $f_2$ above. Thus Eq. (64) simplifies to

$$\nabla_\theta L(\boldsymbol{\gamma}) = \mathbb{E}_X\left[(f_1 X + f_2) \cdot \nabla_\theta\left(e^2(1+2w|w|)X + b - \frac{e^2}{2}\right)\right].$$

Letting $\theta = e, w,$ and $b$ in the above equation, we finally obtain the individual gradients:

$$\frac{\partial L}{\partial e} = \mathbb{E}_X\left[(f_1 X + f_2)(2X(1+2w|w|) - 1))\right] \cdot e,$$

$$\frac{\partial L}{\partial w} = \mathbb{E}_X \left[ (f_1 X + f_2) X \right] \cdot 4e^2 |w|,$$

$$\frac{\partial L}{\partial b} = \mathbb{E}_X \left[ f_1 X + f_2 \right].$$

$\square$

### K.3  Proof of Lemma 4

*Proof.* Slightly changing the variable order, for any $\boldsymbol{\gamma} = (b, e, w) \in \mathbb{R}^3$, we define

$$\boldsymbol{H}(\boldsymbol{\gamma}) \triangleq \nabla^2 L(\boldsymbol{\gamma}) = \begin{bmatrix} \frac{\partial^2 L}{\partial b^2} & \frac{\partial^2 L}{\partial b \partial e} & \frac{\partial^2 L}{\partial b \partial w} \\[6pt] \frac{\partial^2 L}{\partial e \partial b} & \frac{\partial^2 L}{\partial e^2} & \frac{\partial^2 L}{\partial e \partial w} \\[6pt] \frac{\partial^2 L}{\partial w \partial b} & \frac{\partial^2 L}{\partial w \partial e} & \frac{\partial^2 L}{\partial w^2} \end{bmatrix} \in \mathbb{R}^{3 \times 3}. \tag{65}$$

Recall that for any $\boldsymbol{\gamma}_{\min} \in \boldsymbol{\Gamma}_{\min}$ and $\boldsymbol{\gamma}_{\text{sad}} \in \boldsymbol{\Gamma}_{\text{sad}}$, we have $e = 0$ and $b = \log \frac{p}{q}$. Now we compute the second derivatives of $L$ with respect to any $\boldsymbol{\gamma} = (b = \log \frac{p}{q}, e = 0, w)$. We start with the first derivatives. By Lemma 7, the gradients are

$$\begin{aligned} \frac{\partial L}{\partial b} &= \mathbb{E}_X \left[ f_1 X + f_2 \right], \\ \frac{\partial L}{\partial e} &= \mathbb{E}_X \left[ (f_1 X + f_2)(2X(1 + 2w|w|) - 1)) \right] \cdot e, \\ \frac{\partial L}{\partial w} &= \mathbb{E}_X \left[ (f_1 X + f_2) X \right] \cdot 4e^2 |w|, \end{aligned} \tag{66}$$

where

$$f_1 \triangleq \sigma \left( 2e^2 w|w| + b + \frac{e^2}{2} \right) + q - 1 - \sigma \left( b - \frac{e^2}{2} \right) + p,$$

$$f_2 \triangleq \sigma \left( b - \frac{e^2}{2} \right) - p.$$

From Eq. (66), we see that the second derivaties of $L$ depend on the first-derivatives of $f_1$ and $f_2$, which we now compute. Recall that the derivative of the sigmoid function obeys $\sigma'(z) = \sigma(z)(1 - \sigma(z)) = \sigma(z)\sigma(-z)$ for any $z \in \mathbb{R}$. Now the gradients of $f_1$ and $f_2$ with respect to $b, e$, and $w$ are

$$\frac{\partial f_1}{\partial b} = \sigma \left( 2e^2 w|w| + b + \frac{e^2}{2} \right) \sigma \left( -2e^2 w|w| - b - \frac{e^2}{2} \right) - \sigma \left( b - \frac{e^2}{2} \right) \sigma \left( -b + \frac{e^2}{2} \right),$$

$$\frac{\partial f_2}{\partial b} = \sigma \left( b - \frac{e^2}{2} \right) \sigma \left( -b + \frac{e^2}{2} \right),$$

$$\frac{\partial f_1}{\partial e} = (4ew|w| + e) \, \sigma \left( 2e^2 w|w| + b + \frac{e^2}{2} \right) \sigma \left( -2e^2 w|w| - b - \frac{e^2}{2} \right) + e \, \sigma \left( b - \frac{e^2}{2} \right) \sigma \left( -b + \frac{e^2}{2} \right),$$

$$\frac{\partial f_2}{\partial e} = (-e) \, \sigma \left( b - \frac{e^2}{2} \right) \sigma \left( -b + \frac{e^2}{2} \right),$$

$$\frac{\partial f_1}{\partial w} = (4e^2 \cdot w \text{sign}(w)) \, \sigma \left( 2e^2 w|w| + b + \frac{e^2}{2} \right) \sigma \left( -2e^2 w|w| - b - \frac{e^2}{2} \right),$$

$$\frac{\partial f_2}{\partial w} = 0.$$

Using the fact that $\sigma\left(\log\frac{p}{q}\right) = \frac{p}{p+q} = \pi_1$ and $\sigma\left(-\log\frac{p}{q}\right) = \frac{q}{p+q} = \pi_0$, the above gradients evaluated for any $\gamma = (b = \log\frac{p}{q}, e = 0, w)$ further reduce to

$$
\begin{aligned}
\left.\frac{\partial f_1}{\partial b}\right|_\gamma &= 0, \quad \left.\frac{\partial f_2}{\partial b}\right|_\gamma = \pi_0\pi_1, \\
\left.\frac{\partial f_1}{\partial e}\right|_\gamma &= 0, \quad \left.\frac{\partial f_2}{\partial e}\right|_\gamma = 0, \\
\left.\frac{\partial f_1}{\partial w}\right|_\gamma &= 0, \quad \left.\frac{\partial f_2}{\partial w}\right|_\gamma = 0.
\end{aligned}
\tag{67}
$$

Now substituting Eq. (67) when computing the second-derivatives of $L$ in Eq. (66), we obtain

$$
\begin{aligned}
\left.\frac{\partial^2 L}{\partial b^2}\right|_\gamma &= \mathbb{E}_X\left[\left.\frac{\partial f_1}{\partial b}\right|_\gamma X + \left.\frac{\partial f_2}{\partial b}\right|_\gamma\right] = \pi_0\pi_1, \\
\left.\frac{\partial^2 L}{\partial b\partial e}\right|_\gamma &= \mathbb{E}_X\left[\left.\frac{\partial f_1}{\partial e}\right|_\gamma X + \left.\frac{\partial f_2}{\partial e}\right|_\gamma\right] = 0, \\
\left.\frac{\partial^2 L}{\partial b\partial w}\right|_\gamma &= \mathbb{E}_X\left[\left.\frac{\partial f_1}{\partial w}\right|_\gamma X + \left.\frac{\partial f_2}{\partial w}\right|_\gamma\right] = 0, \\
\left.\frac{\partial^2 L}{\partial e^2}\right|_\gamma &= \mathbb{E}_X\left[(f_1 X + f_2)(2X(1 + 2w|w|) - 1))\right]\Big|_\gamma \\
&= \mathbb{E}_X\left[(2f_1(1 + 2w|w|) - f_1 + 2f_2(1 + 2w|w|)X - f_2\right]\Big|_\gamma \\
&= \mathbb{E}_X\left[(f_1(1 + 4w|w|) + f_2(2 + 4w|w|)X - f_2\right]\Big|_\gamma \\
&= (f_1(1 + 4w|w|) + f_2(2 + 4w|w|))\pi_1 - f_2\Big|_\gamma \\
&\overset{(a)}{=} ((p + q - 1)(1 + 4w|w|) - \pi_1(p + q - 1)(2 + 4w|w|))\pi_1 + \pi_1(p + q - 1) \\
&= \pi_1(p + q - 1)\left(1 + 4w|w| - \pi_1(2 + 4w|w|) + 1\right) \\
&\overset{(b)}{=} 2\pi_1\pi_0(p + q - 1)(1 + 2w|w|),
\end{aligned}
\tag{68}
$$

where $(a)$ follows from the fact that $f_1|_{\gamma} = p+q-1$, $f_2|_{\gamma} = \sigma(b)-p = \frac{p}{p+q}-p = \frac{-p}{p+q}(p+q-1) = -\pi_1(p+q-1)$ and $(b)$ from $1-\pi_1 = \pi_0$. Returning to the remaining second derivatives,

$$
\begin{aligned}
\frac{\partial^2 L}{\partial e \partial w}\bigg|_{\gamma} &= \frac{\partial}{\partial e}\left(\mathbb{E}[(f_1 X + f_2)X] \cdot 4e^2|w|\right)\bigg|_{\gamma} \\
&= \frac{\partial}{\partial e}\left(\mathbb{E}[(f_1 + f_2)X] \cdot 4e^2|w|\right)\bigg|_{\gamma} \\
&= \frac{\partial}{\partial e}\left((f_1 + f_2) \cdot 4\pi_1 e^2|w|\right)\bigg|_{\gamma} \\
&= \frac{\partial}{\partial e}\left(\left(\sigma\left(2e^2 w|w| + b + \frac{e^2}{2}\right) + q - 1\right) \cdot 4\pi_1 e^2|w|\right)\bigg|_{\gamma} \\
&= \left(\frac{\partial}{\partial e}\sigma\left(2e^2 w|w| + b + \frac{e^2}{2}\right)\right) 4\pi_1 e^2|w|\bigg|_{\gamma} \qquad\qquad (69) \\
&\quad + \left(\sigma\left(2e^2 w|w| + b + \frac{e^2}{2}\right) + q - 1\right) \cdot \frac{\partial}{\partial e}(4\pi_1 e^2|w|)\bigg|_{\gamma} \\
&= 0, \\
\frac{\partial^2 L}{\partial w^2}\bigg|_{\gamma} &= \frac{\partial}{\partial w}\left(\mathbb{E}[(f_1 X + f_2)X] \cdot 4e^2|w|\right) \\
&= \left(\frac{\partial}{\partial w}\mathbb{E}[(f_1 X + f_2)X] \cdot 4|w|\right)e^2\bigg|_{\gamma} \\
&= 0.
\end{aligned}
$$

Congregating all the second derivatives from Eq. (68) and Eq. (69) into the Hessian $\boldsymbol{H}(\gamma)$ in Eq. (65), we finally obtain

$$
\boldsymbol{H}(\boldsymbol{\gamma}) = \pi_0 \pi_1 \begin{bmatrix} 1 & 0 & 0 \\ 0 & 2(p+q-1)(1+2w|w|) & 0 \\ 0 & 0 & 0 \end{bmatrix}.
$$

$\square$

# L   Proofs of lemmas in App. E

## L.1   Proof of Lemma 5

*Proof.* Recall from Lemma 2 that for any $\boldsymbol{\theta} = (e, w) \in \mathbb{R}^2$ and $b \in \mathbb{R}$, we have

$$L(\boldsymbol{\theta}, b) = \mathbb{E}_{X,Y}\left[\ell_{\log}\left((2Y-1)\left(e^2(1+2w|w|)X + b - \frac{e^2}{2}\right)\right)\right].$$

Since $\ell_{\log}(\cdot)$ is a convex function, $Y \in \{0, 1\}$ and thus $2Y - 1 \in \{\pm 1\}$, the convexity of $L$ in $b$ follows from the following fact:

$$\frac{\partial^2 L}{\partial b^2} = \mathbb{E}_{X,Y}\left[\ell''_{\log}\left((2Y-1)\left(e^2(1+2w|w|)X + b - \frac{e^2}{2}\right)\right)\right] \geq 0.$$

To find the optimal $b_\star$, we set the gradient $\frac{\partial L}{\partial b} = 0$. Thus from Lemma 3, we obtain

$$\frac{\partial L}{\partial b} = \mathbb{E}_X\left[f_1 X + f_2\right] = 0,$$

$$f_1 = \sigma\left(2e^2 w|w| + b + \frac{e^2}{2}\right) + q - 1 - \sigma\left(b - \frac{e^2}{2}\right) + p, \; f_2 = \sigma\left(b - \frac{e^2}{2}\right) - p.$$

Substituting $E \triangleq e^2(1 + 2w|w|)$, $B \triangleq b - \frac{e^2}{2}$, and $\mathbb{E}[X] = \pi_1 = p/(p+q)$ in the above equations,

$$\pi_1\left(\sigma(E+B) + q - 1 - \sigma(B) + p\right) + \sigma(B) - p = \pi_1 \cdot \sigma(E+B) + \pi_0 \cdot \sigma(B) + \frac{p(p+q-1)}{p+q} - p = 0.$$

Further simplifying,

$$\pi_0 \cdot \sigma(B) = \pi_1 \cdot (1 - \sigma(E+B)) = \pi_1 \cdot \sigma(-E-B).$$

In other words,

$$\frac{(1+\exp(-B))^{-1}}{(1+\exp(E+B))^{-1}} = \frac{\pi_1}{\pi_0} = \frac{p}{q} \Rightarrow \frac{1 + \exp(E)\exp(B)}{1 + \exp(-B)} = \frac{p}{q}.$$

Defining $x \triangleq \exp(B)$ and $A \triangleq \exp(E)$, we thus obtain the following quadratic equation in $x$ and its corresponding roots:

$$Ax^2 - x\left(\frac{p}{q} - 1\right) - \frac{p}{q} = 0 \Rightarrow x = \frac{1}{2A}\left[\frac{p}{q} - 1 \pm \sqrt{\left(\frac{p}{q} - 1\right)^2 + 4 \cdot \frac{p}{q} \cdot A}\right].$$

Since $x > 0$, we take the root corresponding to the addition choice above and resubstituting $x = \exp(b - \frac{e^2}{2})$ and $A = \exp(e^2(1 + 2w|w|))$, we obtain the final expression for $b_\star$. In particular, if $e = 0$, it is easy to see that $A = 1$ and hence $x = \exp(b_\star) = \frac{p}{q}$, implying $b_\star = \log\frac{p}{q}$. Similarly, if $A = \frac{(1-p)(1-q)}{pq}$, it's straightforward to see that $\exp(b_\star - e^2/2) = \frac{p}{1-p}$ and hence $b_\star - e^2/2 = \log\frac{p}{1-p}$. $\qquad\square$

## L.2   Proof of Lemma 6

*Proof.* The proof directly follows from Lemma 2 by substituting $b = b_\star$. $\qquad\square$

## L.3   Proof of Lemma 7

*Proof.* By Danskin's theorem [12], it follows that for $b_\star = \operatorname{argmin}_{b \in \mathbb{R}} L(\boldsymbol{\theta}, b)$ and $L(\boldsymbol{\theta}) = L(\boldsymbol{\theta}, b_\star)$, we have $\nabla_{\boldsymbol{\theta}} L(\boldsymbol{\theta}) = \nabla_{\boldsymbol{\theta}} L(\boldsymbol{\theta}, b_\star)$. Using the gradient expressions of $L(\boldsymbol{\theta}, b)$ w.r.t $\boldsymbol{\theta}$ from Lemma 3, and using the fact that $\frac{\partial L}{\partial b} = \mathbb{E}[f_1 X + f_2]$ at $b = b_\star$, the claim follows. $\qquad\square$

## L.4   Proof of Lemma 8

*Proof.* Since $L(\boldsymbol{\theta}) = L(\boldsymbol{\theta}, b_\star)$ where $b_\star = \operatorname{argmin}_{b \in \mathbb{R}} L(\boldsymbol{\theta}, b)$, the identity in Eq. (32) about the Hessian of the loss $L$ with respect to $\boldsymbol{\theta}$ follows from the classical result of [29, Lemma 2.2] about second-derivatives of extremal-value functions. Finally Eq. (33) follows from substituting the full Hessian in $\mathbb{R}^{3\times 3}$ from Lemma 4 in this identity. $\qquad\square$

# M  Proofs of lemmas in App. F

## M.1  Proof of Lemma 9

*Proof.* Denote $(e_t, w_t) = (e, w)$ with the dependence on time implicitly assumed. Then by the definition of GF and the gradient expressions in Lemma 7, we have that

$$\frac{\mathrm{d}e}{\mathrm{d}t} = -\frac{\partial L}{\partial e}(\boldsymbol{\theta}_t) = -2\mathbb{E}[(f_1 X + f_2)X] \cdot (1 + 2w|w|)e, \tag{70}$$

$$\frac{\mathrm{d}w}{\mathrm{d}t} = -\frac{\partial L}{\partial w}(\boldsymbol{\theta}_t) = -\mathbb{E}[(f_1 X + f_2)X] \cdot 4e^2|w|. \tag{71}$$

Dividing Eq. (70) by $(1 + 2w|w|)e$ and Eq. (71) by $4e^2|w|$, we have

$$\frac{1}{1 + 2w|w|}\frac{\mathrm{d}e}{\mathrm{d}t} = \frac{1}{2e|w|}\frac{\mathrm{d}w}{\mathrm{d}t} \Rightarrow e\frac{\mathrm{d}e}{\mathrm{d}t} = \left(w + \frac{1}{2|w|}\right)\frac{\mathrm{d}w}{\mathrm{d}t}.$$

Noting that $\frac{\mathrm{d}}{\mathrm{d}t}(\text{sign}(w) \cdot \log|w|) = \frac{1}{|w|}\frac{\mathrm{d}w}{\mathrm{d}t}$, the above equation can be rewritten as

$$\frac{\mathrm{d}}{\mathrm{d}t}\left(e^2 - w^2 - \text{sign}(w) \cdot \log|w|\right) = 0.$$

Thus defining $\mathcal{E}(e, w) = e^2 - w^2 - \text{sign}(w) \cdot \log|w|$ for $w \neq 0$, the above equation implies $\mathcal{E}(\boldsymbol{\theta}_t) = \mathcal{E}(\boldsymbol{\theta}_0)$ for $\boldsymbol{\theta}_0 = (e_0, w_0)$ with $w_0 \neq 0$. On the other hand, it's easy to see that if $w_0 = 0$, Eq. (71) implies $\frac{\mathrm{d}w}{\mathrm{d}t} = 0$ at $t = 0$ and hence $w_t = 0$ for all $t \geq 0$. $\square$

## M.2  Proof of Lemma 10

*Proof.* To prove the convergence of the trajectory $(\boldsymbol{\theta}_t)_{t \geq 0}$, we use the classical result due to Łojasiewicz [1, Theorem 2.2] which gurantees the convergence of gradient flow for real analytic functions, as long as the trajectory is bounded. Hence we first show the boundedness of the trajectory.

(i) $(\boldsymbol{\theta}_t)_{t \geq 0}$ is bounded.

We consider the cases $\boldsymbol{\theta}_0 \in$ e-axis and $\boldsymbol{\theta}_0 \in \mathbb{R}^2 \setminus$ e-axis separately.

Let's suppose $\boldsymbol{\theta}_0 = (e_0, w_0) \in$ e-axis, i.e. $w_0 = 0$. Thus it follows from Lemma 9 that $w_t = 0$ for all $t \geq 0$. That is, the trajectory always stays on the e-axis and it suffices to track $(e_t)_{t \geq 0}$ and show that they are bounded. To this end, we show that if $e_0 > 0$, we have $\frac{\mathrm{d}e}{\mathrm{d}t} < 0$ and if $e_0 < 0$, we have $\frac{\mathrm{d}e}{\mathrm{d}t} > 0$ for all $t \geq 0$, which establishes our claim. We have from the GF and Lemma 7 that

$$\frac{\mathrm{d}e_t}{\mathrm{d}t} = -\frac{\partial L}{\partial e}(e_t, w_t = 0) = -2\mathbb{E}_X\left[(f_1 X + f_2)X\right]e_t = -2\pi_1(f_1 + f_2)e_t, \tag{72}$$

and

$$\begin{aligned}
f_1 + f_2 &= -\frac{f_2}{\pi_1} + f_2 = -\frac{\pi_0}{\pi_1}f_2 = -\frac{\pi_0}{\pi_1}\left[\sigma\left((b_\star)_t - \frac{e_t^2}{2}\right) - p\right] \\
&= -\frac{\pi_0}{\pi_1}\left[\left(1 + \exp\left(-(b_\star)_t + \frac{e_t^2}{2}\right)\right)^{-1} - p\right] \\
&= -\frac{\pi_0}{\pi_1}\left[\left(1 + \frac{2x_t}{\frac{p}{q} - 1 + \sqrt{\left(\frac{p}{q} - 1\right)^2 + 4 \cdot \frac{p}{q} \cdot x_t}}\right)^{-1} - p\right], \quad x_t \triangleq \exp(e_t^2). \tag{73}
\end{aligned}$$

Defining

$$g(x) \triangleq \frac{2x}{\frac{p}{q} - 1 + \sqrt{\left(\frac{p}{q} - 1\right)^2 + 4 \cdot \frac{p}{q} \cdot x}}, \tag{74}$$

and substituting Eq. (74) and Eq. (73) in Eq. (72), we obtain

$$\frac{\mathrm{d}e_t}{\mathrm{d}t} = 2\pi_0 \left( \frac{1}{1 + g(x_t)} - p \right) \cdot e_t. \tag{75}$$

Since $x_t = \exp(e_t^2) = \exp(-e_t^2)$, in view of Eq. (75), with out loss of generality, we can assume that $e_0 > 0$ and show that $\frac{\mathrm{d}e_t}{\mathrm{d}t} < 0$ for all $t \geq 0$. That is, the RHS Eq. (75) is negative. Note that $x_t \geq 1$ since $x_t = \exp(e_t^2)$ and $g(x_t) > 0$ since the denominator $\frac{p}{q} - 1 + \sqrt{\left( \frac{p}{q} - 1 \right)^2 + 4 \cdot \frac{p}{q} \cdot x_t} > \frac{p}{q} - 1 + \frac{p}{q} + 1 = 2 \cdot \frac{p}{q} > 0$. Further $g(1) = \frac{q}{p}$ and $\lim_{x \to \infty} g(x) = \infty$. If we show that $g(x)$ is increasing in $x$ for $x \geq 1$, it implies $\frac{1}{1+g(x)} - p < \frac{1}{1+g(1)} - p = \frac{1}{1+\frac{q}{p}} - p = -\frac{p}{p+q}(p+q-1) < 0$. Thus the gradient in Eq. (75) remains negative starting at $t = 0$ and hence the sequence $(e_t)_{t \geq 0}$ will be bounded. Now it remains to show $g(\cdot)$ is increasing, i.e. $g'(\cdot) > 0$. Defining $C = \left( \frac{p}{q} - 1 \right) / \left( 2\sqrt{\frac{p}{q}} \right)$ and $D = C^2$, we have that $g(x)$ upto a postive scaling is

$$g(x) = \frac{x}{C + \sqrt{x + D}}.$$

Hence

$$g'(x) = \frac{C + \sqrt{x + D} - \frac{x}{2\sqrt{x+D}}}{(C + \sqrt{x + D})^2}.$$

Thus it suffices to show that $h_1(x) \triangleq C + \sqrt{x + D} > h_2(x) \triangleq \frac{x}{2\sqrt{x+D}}$ for $x \geq 1$. Note that $h_1(1) - h_2(1)$ is given by

$$h_1(1) - h_2(1) = \frac{\frac{p}{q} - 1}{2\sqrt{\frac{p}{q}}} + \sqrt{1 + \left( \frac{\frac{p}{q} - 1}{2\sqrt{\frac{p}{q}}} \right)^2} - \frac{1}{2\sqrt{1 + \left( \frac{\frac{p}{q}-1}{2\sqrt{\frac{p}{q}}} \right)^2}}$$

$$= \sqrt{\frac{p}{q}} - \frac{\sqrt{\frac{p}{q}}}{1 + \frac{p}{q}}$$

$$> 0.$$

Now we show that $h'(x) > h'_2(x)$ for all $x \geq 1$ which implies that $h_1(x) > h_2(x)$ for all $x \geq 1$, thus establishing our claim. To this end, we have that

$$h'_1(x) - h'_2(x) = \frac{1}{2\sqrt{x + D}} - \frac{\sqrt{x + D} - \frac{x}{2\sqrt{x+D}}}{2(x + D)}$$

$$= \frac{x}{2\sqrt{x + D}(x + D)} > 0.$$

This proves our claim that $g(\cdot)$ is increasing and hence $(e_t)_{t \geq 0}$, and consequently $(\boldsymbol{\theta}_t)_{t \geq 0}$, is bounded when $\boldsymbol{\theta}_0 \in$ e-axis.

Now let's assume that $\boldsymbol{\theta}_0 = (e_0, w_0) \in \mathbb{R}^2 \setminus$ e-axis. Since $(\boldsymbol{\theta}_t)_{t \geq 0} \subseteq \mathbb{R}^2 \setminus$ e-axis, it follows that the loss $L(\cdot)$ is analytic on the trajectory (since the logistic function is analytic), and hence by [1, Theorem 2.2], it follows that $\lim_{t \to \infty} \|\boldsymbol{\theta}_t\|$ exists. Now we show that $\lim_{t \to \infty} \|\boldsymbol{\theta}_t\| \neq \infty$, which implies the desired result about boundedness. To show $\lim_{t \to \infty} \|\boldsymbol{\theta}_t\| \neq \infty$, we show that there exists a large $B > 0$ such that for any $\boldsymbol{\theta}_t = (e, w) \in \mathbb{R}^2$ with $\|(e, w)\| \geq B$, the velocity vector $\frac{\mathrm{d}\boldsymbol{\theta}}{\mathrm{d}t}$ points inwards into the ball of radius $B$ and thus the trajectory always stays inside this ball, and hence bounded. To establish this, let's denote $(e_t, w_t) = (e, w)$ with the dependence on time implicitly assumed. Then by the definition of GF and the gradient expressions in Lemma 7, we have that

$$\frac{\mathrm{d}e}{\mathrm{d}t} = -\frac{\partial L}{\partial e}(\boldsymbol{\theta}_t) = -2\mathbb{E}[(f_1 X + f_2)X] \cdot (1 + 2w|w|)e \tag{76}$$

$$\frac{\mathrm{d}w}{\mathrm{d}t} = -\frac{\partial L}{\partial w}(\boldsymbol{\theta}_t) = -\mathbb{E}[(f_1 X + f_2)X] \cdot 4e^2|w|, \tag{77}$$

where $f_1 = \sigma\left(2e^2 w|w| + b_\star + \frac{e^2}{2}\right) + q - 1 - \sigma\left(b_\star - \frac{e^2}{2}\right) + p$, and $f_2 = \sigma\left(b_\star - \frac{e^2}{2}\right) - p$ with $\pi_1 f_1 + f_2 = 0$. Given that only $\frac{de}{dt}$ flips in sign under the transformation $(e, w) \mapsto (-e, w)$, with out loss of generality we can assume $e > 0$. Now let's also assume $w > 0$. Thus, in view of Eq. (76) and GF, to show that the derivative points inwards, it suffices to show that $\mathbb{E}[(f_1 X + f_2)X] > 0$ for reasonably large $B$ with $\|(e, w)\| = B$. Similar to Eq. (73) and Eq. (74) above, using the relation $\pi_1 f_1 + f_2 = 0$, we obtain

$$\mathbb{E}[(f_1 X + f_2)X] = \pi_1(f_1 + f_2) = -\pi_0\left(\frac{1}{1 + g(x)} - p\right), \quad x \triangleq \exp(e^2(1 + 2w|w|)). \quad (78)$$

Using the fact that $g(x)$ is increasing for $x \geq 1$ with $\lim_{x \to \infty} g(x) = \infty$, and $|w| = w > 0$, we can chose a $B > 0$ such that for any $\|(e, w)\| \geq B$, in Eq. (78) we have $1/(1 + g(x)) < p$ and hence $\mathbb{E}[(f_1 X + f_2)X] > 0$. This finishes the proof of our claim. The proof for $w < 0$ is similar, where we make use of the fact that $\lim_{x \to 0} g(x) = 0$ to show $\mathbb{E}[(f_1 X + f_2)X] < 0$ for $e, w$ reasonably large.

(ii) $\lim_{t \to \infty} \boldsymbol{\theta}_t = \boldsymbol{\theta}_{\lim}$. Since the logistic function $\ell_{\log}(\cdot)$ is analytic, it follows from Lemma 6 that the loss $L(\boldsymbol{\theta})$ is analytic too whenever $w \neq 0$. On the other hand, when $w = 0$, it's easy to see that $L$ is an analytic function of $e \in \mathbb{R}$. By Lemma 9, we know that if $w_0 \neq 0$, $w_t \neq 0$ and if $w_0 = 0$, $w_t = 0$ for all $t \geq 0$. Thus the loss is analytic on the trajectory for all $t \geq 0$. Since the trajectory is bounded, it follows from Łojasiewicz's theorem [1, Theorem 2.2] that there exists a $\boldsymbol{\theta}_{\lim} \in \mathbb{R}^2$ such that $\lim_{t \to \infty} \boldsymbol{\theta}_t = \boldsymbol{\theta}_{\lim}$.

(iii) $\lim_{t \to \infty} \|\nabla L(\boldsymbol{\theta}_t)\| = \|\nabla L(\boldsymbol{\theta}_{\lim})\| = 0$. Since the trajectory is bounded, it follows from [2, Theorem 2] that the gradient converges to zero, i.e. $\lim_{t \to \infty} \|\nabla L(\boldsymbol{\theta}_t)\| = 0$. Since $\nabla L(\cdot)$ is a continuous function and $\lim_{t \to \infty} \boldsymbol{\theta}_t = \boldsymbol{\theta}_{\lim}$, we have $\lim_{t \to \infty} \|\nabla L(\boldsymbol{\theta}_t)\| = \|\nabla L(\boldsymbol{\theta}_{\lim})\| = 0$.

$\square$

## M.3 Proof of Lemma 11

*Proof.* Since the energy function $\mathcal{E}(\cdot, \cdot)$ in Eq. (48) is a continuous function in $\mathbb{R}^2 \setminus$ e-axis, and any trajectory $(\boldsymbol{\theta}_t)_{t \geq 0}$ with intialization $\theta \in \mathbb{R}^2 \setminus$ e-axis stays in $\mathbb{R}^2 \setminus$ e-axis for all $t \geq 0$ (Lemma 9), it follows that $\lim_{t \to \infty} \mathcal{E}(\boldsymbol{\theta}_t) = \mathcal{E}(\boldsymbol{\theta}_{\lim}) = \mathcal{E}(\boldsymbol{\theta}_0)$. As $\nabla L(\boldsymbol{\theta}_{\lim}) = 0$ from Lemma 10, it follows that $\boldsymbol{\theta}_{\lim}$ lies at the intersection of the contour line $\mathcal{E}(e, w) = \mathcal{E}_0$ with the set of critical points of $L$ in $\mathbb{R}^2$.

On the other hand, if $\boldsymbol{\theta}_0 \in$ e-axis, we have $\boldsymbol{\theta}_t \in$ e-axis from Lemma 9 for all $t \geq 0$. Hence $\boldsymbol{\theta}_{\lim} \in$ e-axis. $\square$

## M.4 Proof of Lemma 12

*Proof.* Recall that $f : \mathbb{R} \setminus \{0\} \to \mathbb{R}$, defined as $f(w) \triangleq \mathcal{E}(e = 0, w) = -(w^2 + \text{sign}(w) \cdot \log|w|)$. If $w < 0$, we have

$$f(w) = -(w^2 - \log(-w)), \quad f'(w) = -2w + \frac{1}{w}.$$

Hence $f'(w) \geq 0$ for $w \in (-\infty, -1/\sqrt{2}]$ and $f'(w) \leq 0$ for $w \in [-1/\sqrt{2}, 0)$ with $f'(-\frac{1}{\sqrt{2}}) = 0$. It's also straightforward to see that $\lim_{w \to -\infty} f(w) = -\infty$, $\lim_{w \to 0^-} f(w) = -\infty$, and $f(-1/\sqrt{2}) = \mathcal{E}_{\text{sad}}$ (by the definition of $f$). This establishes (i), (ii), and (iii).

On the other hand, for $w > 0$, we have $f(w) = -(w^2 + \log w)$ and $f'(w) = -(2w + 1/w)$. Hence $f$ is monotonically decreasing for $w > 0$ with $\lim_{w \to 0^+} f(w) = \infty$ and $\lim_{w \to \infty} f(w) = -\infty$. Note that $w = 0$ acts as an energy barrier since $\lim_{w \to 0^-} f(w) = -\infty$ whereas $\lim_{w \to 0^+} f(w) = \infty$. $\square$

# N  Proofs of lemmas in App. G

## N.1  Proof of Lemma 13

*Proof.* First we recall the loss with the bias $L(\boldsymbol{\theta}, b)$ from Eq. (41):
$$L(\boldsymbol{\theta}, b) = \mathbb{E}_{X,Y} \left[ \ell_{\log} \left( (2Y - 1) \cdot \text{logit}_X(\boldsymbol{\theta}, b) \right) \right],$$

where $\text{logit}_X(\boldsymbol{\theta}, b) = e^2 \left[ \left( X - \frac{1}{2} \right) \left( 1 + ae^2 \right) (1 + 2w|w|) + w|w(1 + ae^2)| \right] + b$ with $\boldsymbol{\theta} = (e, w, a)$. Using the fact that $\ell'_{\log}(z) = \sigma(z) - 1$ and $2Y - 1 \in \{\pm 1\}$, we have for any $\theta \in \{e, w, a, b\}$ that
$$\nabla_\theta L = \mathbb{E} \left[ (\sigma((2Y - 1) \cdot \text{logit}_X) - 1)(2Y - 1) \cdot \nabla_\theta \text{logit}_X \right] = \mathbb{E} \left[ (\sigma(\text{logit}_X) - Y) \cdot \nabla_\theta \text{logit}_X \right]. \tag{79}$$

Now we simplify $\sigma(\text{logit}_X)$ using the fact that $X \in \{0, 1\}$:
$$\begin{aligned}
\sigma(\text{logit}_X) &= \sigma \left( e^2 \left[ \left( X - \frac{1}{2} \right) \left( 1 + ae^2 \right) (1 + 2w|w|) + w|w(1 + ae^2)| \right] + b \right) \\
&= X \underbrace{\sigma \left( e^2 \left[ \frac{1}{2} \left( 1 + ae^2 \right) (1 + 2w|w|) + w|w(1 + ae^2)| \right] + b \right)}_{\triangleq \phi_1} \\
&\quad + (1 - X) \underbrace{\sigma \left( e^2 \left[ \frac{-1}{2} \left( 1 + ae^2 \right) (1 + 2w|w|) + w|w(1 + ae^2)| \right] + b \right)}_{\triangleq \phi_0} \\
&= X\phi_1 + (1 - X)\phi_0 \\
&= X(\phi_1 - \phi_0) + \phi_0.
\end{aligned}$$
Thus the gradients in Eq. (79) are given by
$$\begin{aligned}
\nabla_\theta L &= -\mathbb{E} \left[ (Y - X(\phi_1 - \phi_0) - \phi_0) \nabla_\theta \text{logit}_X \right] \\
&= -\mathbb{E}_X \left[ \mathbb{E}_{x_1^{n+1}} \left[ (\mathbb{E}[Y \mid X] - X(\phi_1 - \phi_0) - \phi_0) \nabla_\theta \text{logit}_X \right] \right] \\
&= -\mathbb{E}_X \left[ ((1 - p - q) X + p - X(\phi_1 - \phi_0) - \phi_0) \nabla_\theta \text{logit}_X \right] \\
&= -\mathbb{E}_X \left[ \left( \Big( \underbrace{1 - p - q - \phi_1 + \phi_0}_{f_1} \Big) X + \underbrace{p - \phi_0}_{f_2} \right) \nabla_\theta \text{logit}_X \right] \\
&= -\mathbb{E}_X \left[ (f_1 X + f_2) \nabla_\theta \text{logit}_X \right].
\end{aligned} \tag{80}$$

Now we compute the individual gradients with respect to $e, w, a$ and $b$. Recall that

$$\text{logit}_X = e^2 \left[ \left( X - \frac{1}{2} \right) \left( 1 + ae^2 \right) (1 + 2w|w|) + w|w(1 + ae^2)| \right] + b.$$

Thus,
$$\begin{aligned}
\nabla_e \text{logit}_X &= 2e \left[ \left( X - \frac{1}{2} \right) \left( 1 + ae^2 \right) (1 + 2w|w|) + w|w(1 + ae^2)| \right] \\
&\quad + e^2 \left[ 2ae \left( X - \frac{1}{2} \right) (1 + 2w|w|) + w\text{sign} \left( w \left( 1 + ae^2 \right) \right) (2ae) \right] \\
&= 2e \left( X - \frac{1}{2} \right) \left( 1 + ae^2 \right) (1 + 2w|w|) + 2ew|w(1 + ae^2)| \\
&\quad + 2e^3 a \left( X - \frac{1}{2} \right) (1 + 2w|w|) + 2e^3 aw\text{sign} \left( w \left( 1 + ae^2 \right) \right) \\
&= 2e \left( X - \frac{1}{2} \right) \left( 1 + ae^2 \right) (1 + 2w|w|) + 2e^3 a \left( X - \frac{1}{2} \right) (1 + 2w|w|) \\
&\quad + 2ew|w(1 + ae^2)| + 2e^3 aw\text{sign} \left( w \left( 1 + ae^2 \right) \right).
\end{aligned}$$

Substituting the above equation in Eq. (80), we obtain

$$\nabla_e L = -\left(\mathbb{E}\left[(f_1 X + f_2)\left(X - \frac{1}{2}\right)\right]\right) \cdot 2e\left(1 + ae^2\right)\left(1 + 2w|w|\right)$$
$$-\left(\mathbb{E}\left[(f_1 X + f_2)\left(X - \frac{1}{2}\right)\right]\right) \cdot 2e^3 a\left(1 + 2w|w|\right)$$
$$-\left(\mathbb{E}\left[(f_1 X + f_2)\right]\right) \cdot 2ew|w(1 + ae^2)|$$
$$-\left(\mathbb{E}\left[(f_1 X + f_2)\right]\right) \cdot 2e^3 aw\,\mathrm{sign}\left(w\left(1 + ae^2\right)\right).$$

Now we compute the derivative with respect to $w$.

$$\mathrm{logit}_X = e^2\left[\left(X - \frac{1}{2}\right)\left(1 + ae^2\right)\left(1 + 2w|w|\right) + w|w(1 + ae^2)|\right] + b$$
$$\Rightarrow \nabla_w \mathrm{logit}_X = 2e^2\left(X - \frac{1}{2}\right)\left(1 + ae^2\right)\left(|w| + \mathrm{sign}\left(w\right)w\right)$$
$$+ e^2\left[|w(1 + ae^2)| + w\left(1 + ae^2\right)\mathrm{sign}\left(w\left(1 + ae^2\right)\right)\right]$$
$$\Rightarrow \nabla_w L = -\left(\mathbb{E}\left[(f_1 X + f_2)\left(X - \frac{1}{2}\right)\right]\right)2e^2\left(1 + ae^2\right)\left(|w| + \mathrm{sign}\left(w\right)w\right)$$
$$-\left(\mathbb{E}\left[(f_1 X + f_2)\right]\right)e^2\left[|w(1 + ae^2)| + w\left(1 + ae^2\right)\mathrm{sign}\left(w\left(1 + ae^2\right)\right)\right].$$

Similarly, for $a$:

$$\mathrm{logit}_X = e^2\left[\left(X - \frac{1}{2}\right)\left(1 + 2w|w|\right) + w|w(1 + ae^2)|\right] + b$$
$$\Rightarrow \nabla_a \mathrm{logit}_X = e^4\left(X - \frac{1}{2}\right)\left(1 + ae^2\right)\left(1 + 2w|w|\right)$$
$$+ e^4 w^2\mathrm{sign}\left(w(1 + ae^2)\right)$$
$$\Rightarrow \nabla_a L = -\left(\mathbb{E}\left[(f_1 X + f_2)\left(X - \frac{1}{2}\right)\right]\right)e^4\left(1 + 2w|w|\right)$$
$$-\left(\mathbb{E}\left[(f_1 X + f_2)\right]\right)e^4 w^2\mathrm{sign}\left(w(1 + ae^2)\right).$$

Finally, since $\nabla_b \mathrm{logit}_X = 1$, it follows from Eq. (80) that

$$\nabla_b L = -\mathbb{E}\left[f_1 X + f_2\right]. \tag{81}$$

For the optimal $b_\star$, we have $\nabla_b L = 0$ and hence $\mathbb{E}\left[f_1 X + f_2\right] = 0$, simplifying the expressions for the gradients of $e, w$, and $a$ above. In fact, there exists a closed form expression for $b_\star$ in terms of $e, w$, and $a$. Recall from Eq. (80) that

$$-f_1 = \sigma(z_1) - \sigma(z_2) + p + q - 1,$$
$$-f_2 = \sigma(z_2) - p,$$
$$z_1 \triangleq e^2\left[\frac{1}{2}\left(1 + ae^2\right)\left(1 + 2w|w|\right) + w|w(1 + ae^2)|\right] + b, \tag{82}$$
$$z_2 \triangleq e^2\left[\frac{-1}{2}\left(1 + ae^2\right)\left(1 + 2w|w|\right) + w|w(1 + ae^2)|\right] + b.$$

Substituting Eq. (82) in Eq. (81) and setting the gradient to zero there, we have

$$-\mathbb{E}_X\left[f_1 X + f_2\right] = \left(\sigma(z_1) - \sigma(z_2) + p + q - 1\right)\mathbb{E}[X] + \sigma(z_2) - p$$
$$= \left(\sigma(z_1) - \sigma(z_2) + p + q - 1\right)\pi_1 + \sigma(z_2) - p = 0.$$

Simplifying,

$$
\begin{aligned}
(\sigma(z_1) - \sigma(z_2) + p + q - 1)\pi_1 &= p - \sigma(z_2) \\
\Rightarrow (\sigma(z_1) - 1)\pi_1 - \sigma(z_2)\pi_1 + p &= p - \sigma(z_2) \\
\Rightarrow (\sigma(z_1) - 1)\pi_1 &= \sigma(z_2)(\pi_1 - 1) \\
\Rightarrow \frac{\sigma(z_2)}{1 - \sigma(z_1)} &= \frac{\pi_1}{1 - \pi_1} = \frac{p}{q},
\end{aligned}
$$

Using the definition of the sigmoid function and rearranging,

$$
\begin{aligned}
\frac{1 + \exp(z_1)}{1 + \exp(-z_2)} &= \frac{p}{q} \\
\Rightarrow \exp(z_1) + 1 &= \frac{p}{q}\left(1 + \exp(-z_2)\right) \\
\Rightarrow \exp(2z_1) + \exp(z_1) &= \frac{p}{q}\exp(z_1) + \frac{p}{q}\exp(z_1 - z_2) \\
\Rightarrow (\exp(z_1))^2 + \exp(z_1)(1 - \frac{p}{q}) - \frac{p}{q}\cdot\exp(z_1 - z_2) &= 0.
\end{aligned} \tag{83}
$$

By definitions of $z_1$ and $z_2$ in Eq. (82), we have $z_1 - z_2 = e^2(1 + ae^2)(1 + 2w|w|)$ and thus $A \triangleq \exp(z_1 - z_2) = \exp(e^2(1 + ae^2)(1 + 2w|w|))$. Thus the quadratic equation in Eq. (83) simplifies to

$$
(\exp(z_1))^2 + \exp(z_1)(1 - \frac{p}{q}) - \frac{p}{q}\cdot A = 0.
$$

On solving the quadratic equation for $\exp(z_1)$:

$$
\begin{aligned}
\exp(z_1) &= \frac{1}{2}\left[\frac{p}{q} - 1 + \sqrt{\left(\frac{p}{q} - 1\right)^2 + 4\cdot\frac{p}{q}\cdot A}\right] \\
\Rightarrow z_1 &= \log\left(\frac{1}{2}\left[\frac{p}{q} - 1 + \sqrt{\left(\frac{p}{q} - 1\right)^2 + 4\cdot\frac{p}{q}\cdot A}\right]\right) \\
\Rightarrow b_\star &= \log\left(\frac{1}{2}\left[\frac{p}{q} - 1 + \sqrt{\left(\frac{p}{q} - 1\right)^2 + 4\cdot\frac{p}{q}\cdot A}\right]\right) \\
&\quad - e^2\left[\frac{1}{2}\left(1 + ae^2\right)\left(1 + 2w|w|\right) + w|w(1 + ae^2)|\right].
\end{aligned}
$$

Note that when $a = 0$ above, we recover the expression for the optimal bias in Lemma 5. This concludes the proof.

$\square$

## N.2 Proofs of Thm. 9 and Thm. 10

We prove Thm. 9 and Thm. 10 below. Note that Thm. 10 directly follows from the former by removing the bias $b$, since for any critical point $\gamma = (e, w, b, a) \in \mathbb{R}^4$ with $\nabla L(\gamma) = 0$, the bias $b$ is already the optimal one corresponding to $\theta = (e, w, a) \in \mathbb{R}^3$, i.e. $b = b_\star(\theta) = \operatorname{argmin}_{b\in\mathbb{R}} L(\theta, b)$. This is similar to the proof of Thm. 1, which follows from Thm. 7.

Now we prove Thm. 9.

*Proof.* We characterize the set of global minima first.

**Set of all global minima.** Let $\boldsymbol{\gamma}_\star \in \mathbb{R}^4$ be arbitrary. From [23, Lemma 1], we have that $\boldsymbol{\gamma}_\star$ is a global minimum for the loss $L(\cdot)$ in Eq. (23) if and only if its prediction probability satisfies $f_{\boldsymbol{\gamma}_\star}(x_1^n) = \mathbb{P}\left(x_{n+1} = 1 \mid x_n\right)$, the Markov kernel. Since the input $\{x_n\}_{n=1}^N \sim (\boldsymbol{\pi}(p,q), \boldsymbol{P}(p,q))$, we have that

$$\mathbb{P}\left(x_{n+1} = 1 \mid x_n\right) = (1 - x_n)p + x_n(1 - q) = (1 - p - q)x_n + p. \tag{84}$$

On the other hand, by definition, from Eq. (3), $f_{\boldsymbol{\gamma}_\star}(x_1^n) = \sigma\left(e^2\left[\left(x_n - \frac{1}{2}\right)\left(1 + ae^2\right)\left(1 + 2w|w|\right) + w|w(1 + ae^2)|\right] + b\right)$, where $\boldsymbol{\gamma}_\star = (e, w, b, a)$. Since $x_n \in \{0, 1\}$, this can be further simplified to

$$
\begin{aligned}
f_{\boldsymbol{\gamma}_\star}(x_1^n) &= \sigma\left(e^2\left[\left(x_n - \frac{1}{2}\right)\left(1 + ae^2\right)\left(1 + 2w|w|\right) + w|w(1 + ae^2)|\right] + b\right) \\
&= x_n\sigma\left(e^2\left[\frac{1}{2}\left(1 + ae^2\right)\left(1 + 2w|w|\right) + w|w(1 + ae^2)|\right] + b\right) \\
&\quad + (1 - x_n)\sigma\left(e^2\left[\frac{-1}{2}\left(1 + ae^2\right)\left(1 + 2w|w|\right) + w|w(1 + ae^2)|\right] + b\right) \\
&= x_n\sigma\left(e^2\left[\frac{1}{2}\left(1 + ae^2\right)\left(1 + 2w|w|\right) + w|w(1 + ae^2)|\right] + b\right) \\
&\quad - x_n\sigma\left(e^2\left[\frac{-1}{2}\left(1 + ae^2\right)\left(1 + 2w|w|\right) + w|w(1 + ae^2)|\right] + b\right) \\
&\quad + \sigma\left(e^2\left[\frac{-1}{2}\left(1 + ae^2\right)\left(1 + 2w|w|\right) + w|w(1 + ae^2)|\right] + b\right).
\end{aligned} \tag{85}
$$

Since both $f_{\boldsymbol{\gamma}_\star}(x_1^n)$ and $\mathbb{P}\left(x_{n+1} = 1 \mid x_n\right)$ are linear functions of $x_n$, equating them for all vallues of $x_n \in \{0, 1\}$ implies that the respective coeffecients in these functions in Eq. (84) and Eq. (85) are also equal, i.e.

$$
\begin{aligned}
1 - p - q &= \sigma\left(e^2\left[\frac{1}{2}\left(1 + ae^2\right)\left(1 + 2w|w|\right) + w|w(1 + ae^2)|\right] + b\right) \\
&\quad - \sigma\left(e^2\left[\frac{-1}{2}\left(1 + ae^2\right)\left(1 + 2w|w|\right) + w|w(1 + ae^2)|\right] + b\right), \\
p &= \sigma\left(e^2\left[\frac{-1}{2}\left(1 + ae^2\right)\left(1 + 2w|w|\right) + w|w(1 + ae^2)|\right] + b\right),
\end{aligned} \tag{86}
$$

and hence

$$
\begin{aligned}
\sigma\left(e^2\left[\frac{1}{2}\left(1 + ae^2\right)\left(1 + 2w|w|\right) + w|w(1 + ae^2)|\right] + b\right) &= 1 - q, \\
\sigma\left(e^2\left[\frac{-1}{2}\left(1 + ae^2\right)\left(1 + 2w|w|\right) + w|w(1 + ae^2)|\right] + b\right) &= p.
\end{aligned} \tag{87}
$$

Since $\sigma(z) = y$ for $y \in (0, 1)$ implies $z = \log \frac{y}{1-y}$, Eq. (87) can be rewritten as

$$
\begin{aligned}
e^2\left[\frac{1}{2}\left(1 + ae^2\right)\left(1 + 2w|w|\right) + w|w(1 + ae^2)|\right] + b &= \log \frac{1 - q}{q}, \\
e^2\left[\frac{-1}{2}\left(1 + ae^2\right)\left(1 + 2w|w|\right) + w|w(1 + ae^2)|\right] + b &= \log \frac{p}{1 - p}.
\end{aligned}
$$

Adding and subtracting the above two equations, we obtain

$$e^2 w |w(1 + ae^2)| + b = \frac{1}{2} \log \frac{p(1-q)}{q(1-p)},$$

$$e^2 \left(1 + ae^2\right) \left(1 + 2w|w|\right) = \log \frac{(1-q)(1-p)}{pq}. \tag{88}$$

Thus $\boldsymbol{\gamma}_\star \in \mathbb{R}^4$ is a global minimum for $L(\cdot)$ if and only if it satisfies Eq. (88). It's easy to see that $\boldsymbol{\gamma}_\star$ is already a critical point for $L$ as Eq. (86) is equivalent to $f_1 = f_2 = 0$ in Lemma 13. Thus, the set of all global minimum $\boldsymbol{\Gamma}_\star(p, q)$ is given by

$$\boldsymbol{\Gamma}_\star(p,q) \triangleq \{\boldsymbol{\gamma}_\star = (e, w, b, a) \in \mathbb{R}^4 : e^2 w |w(1 + ae^2)| + b = \frac{1}{2} \log \frac{p(1-q)}{q(1-p)},$$

$$e^2 \left(1 + ae^2\right) \left(1 + 2w|w|\right) = \log \frac{(1-q)(1-p)}{pq}\}.$$

Since the prediction $f_{\boldsymbol{\gamma}_\star}(\cdot)$ equals the Markov kernel for any $\boldsymbol{\gamma}_\star \in \boldsymbol{\Gamma}_\star$, it follows from Thm. 4 (or [23, Lemma 1]) that $L(\boldsymbol{\gamma}_\star) = H(x_{n+1} \mid x_n)$, the entropy rate of the Markov chain. Now we characterize the remaining set of stationary points.

**Non-global-min critical points.** For any critical point $\boldsymbol{\gamma} = (e, w, a, b) \in \mathbb{R}^4$, we have from the gradient expressions in Lemma 13 that (denoting $-f_1$ and $-f_2$ from the lemma as $f_1$ and $f_2$) respectively)

$$\frac{\partial L}{\partial b} = \mathbb{E}_X \left[ f_1 X + f_2 \right] = 0,$$

$$\frac{\partial L}{\partial e} = \mathbb{E}_X \left[ (f_1 X + f_2) \left( X - \frac{1}{2} \right) \right] 2e \left(1 + ae^2\right) \left(1 + 2w|w|\right)$$

$$\quad + \mathbb{E}_X \left[ (f_1 X + f_2) \left( X - \frac{1}{2} \right) \right] 2e^3 a \left(1 + 2w|w|\right) = 0, \tag{89}$$

$$\frac{\partial L}{\partial w} = \mathbb{E}_X \left[ (f_1 X + f_2) \left( X - \frac{1}{2} \right) \right] 2e^2 \left(1 + ae^2\right) \left(|w| + \text{sign}\,(w)\,w\right) = 0,$$

$$\frac{\partial L}{\partial a} = \mathbb{E}_X \left[ (f_1 X + f_2) \left( X - \frac{1}{2} \right) \right] e^4 \left(1 + 2w|w|\right) = 0.$$

From Eq. (89), we have that $\mathbb{E}_X \left[ f_1 X + f_2 \right] = 0$. If $\mathbb{E}_X \left[ (f_1 X + f_2) \left( X - \frac{1}{2} \right) \right] = \mathbb{E}[(f_1 X + f_2)X] = 0$, we have that $f_1 = f_2 = 0$, implying $\boldsymbol{\gamma}$ is a global minimum. Hence without loss of generality, assume that $\mathbb{E}_X \left[ (f_1 X + f_2) \left( X - \frac{1}{2} \right) \right] \neq 0$. Then we can partition the above set of equations into the following regions of stationarity:

(i) $\mathbb{E}_X \left[ f_1 X + f_2 \right] = 0, e = 0$,

(ii) $\mathbb{E}_X \left[ f_1 X + f_2 \right] = 0, e \neq 0, 1 + ae^2 = 0, 1 + 2w|w| = 0$.

Slightly changing the variable order, for any $\boldsymbol{\gamma} = (b, e, w, a) \in \mathbb{R}^4$, we define

$$\boldsymbol{H}(\boldsymbol{\gamma}) \triangleq \nabla^2 L(\boldsymbol{\gamma}) = \begin{bmatrix} \frac{\partial^2 L}{\partial b^2} & \frac{\partial^2 L}{\partial b \partial e} & \frac{\partial^2 L}{\partial b \partial w} & \frac{\partial^2 L}{\partial b \partial a} \\[6pt] \frac{\partial^2 L}{\partial e \partial b} & \frac{\partial^2 L}{\partial e^2} & \frac{\partial^2 L}{\partial e \partial w} & \frac{\partial^2 L}{\partial e \partial a} \\[6pt] \frac{\partial^2 L}{\partial w \partial b} & \frac{\partial^2 L}{\partial w \partial e} & \frac{\partial^2 L}{\partial w^2} & \frac{\partial^2 L}{\partial w \partial a} \\[6pt] \frac{\partial^2 L}{\partial a \partial b} & \frac{\partial^2 L}{\partial a \partial e} & \frac{\partial^2 L}{\partial a \partial w} & \frac{\partial^2 L}{\partial a^2} \end{bmatrix} \in \mathbb{R}^{4 \times 4}. \tag{90}$$

Recall that

$$f_1 = \sigma(z_1) - \sigma(z_2) + p + q - 1,$$
$$f_2 = \sigma(z_2) - p,$$
$$z_1 \triangleq e^2 \left[ \frac{1}{2} \left( 1 + ae^2 \right) \left( 1 + 2w|w| \right) + w|w(1 + ae^2)| \right] + b,$$
$$z_2 \triangleq e^2 \left[ \frac{-1}{2} \left( 1 + ae^2 \right) \left( 1 + 2w|w| \right) + w|w(1 + ae^2)| \right] + b.$$

From Eq. (89), we see that the second derivaties of $L$ depend on the first-derivatives of $f_1$ and $f_2$, which we now compute. Recall that the derivative of the sigmoid function obeys $\sigma'(z) = \sigma(z)(1 - \sigma(z)) = \sigma(z)\sigma(-z)$ for any $z \in \mathbb{R}$. Now the gradients of $f_1$ and $f_2$ with respect to $b, e, w$ and $a$ are

$$\frac{\partial f_1}{\partial b} = \sigma(z_1)\sigma(-z_1) - \sigma(z_2)\sigma(-z_2),$$

$$\frac{\partial f_2}{\partial b} = \sigma(z_2)\sigma(-z_2),$$

$$\frac{\partial f_1}{\partial e} = \sigma(z_1)\sigma(-z_1) \left\{ 2e \left[ \frac{1}{2} \left( 1 + ae^2 \right) \left( 1 + 2w|w| \right) + w|w(1 + ae^2)| \right] \right\}$$
$$+ \sigma(z_1)\sigma(-z_1) \left\{ 2ae^3 \left[ \frac{1}{2} \left( 1 + 2w|w| \right) + w|w|\mathrm{sign}(1 + 2w|w|) \right] \right\}$$
$$- \sigma(z_2)\sigma(-z_2) \left\{ 2e \left[ -\frac{1}{2} \left( 1 + ae^2 \right) \left( 1 + 2w|w| \right) + w|w(1 + ae^2)| \right] \right\}$$
$$- \sigma(z_2)\sigma(-z_2) \left\{ 2ae^3 \left[ -\frac{1}{2} \left( 1 + 2w|w| \right) + w|w|\mathrm{sign}(1 + 2w|w|) \right] \right\}$$

$$\frac{\partial f_2}{\partial e} = \sigma(z_2)\sigma(-z_2) \left\{ 2e \left[ -\frac{1}{2} \left( 1 + ae^2 \right) \left( 1 + 2w|w| \right) + w|w(1 + ae^2)| \right] \right\}$$
$$+ \sigma(z_2)\sigma(-z_2) \left\{ 2ae^3 \left[ -\frac{1}{2} \left( 1 + 2w|w| \right) + w|w|\mathrm{sign}(1 + 2w|w|) \right] \right\}$$
(91)

$$\frac{\partial f_1}{\partial w} = \sigma(z_1)\sigma(-z_1) \left\{ 2e^2 \left[ \frac{1}{2} \left( 1 + ae^2 \right) |w| + |w||1 + ae^2| \right] \right\}$$
$$- \sigma(z_2)\sigma(-z_2) \left\{ 2e^2 \left[ -\frac{1}{2} \left( 1 + ae^2 \right) |w| + |w||1 + ae^2| \right] \right\},$$

$$\frac{\partial f_2}{\partial w} = \left\{ 2e^2 \left[ -\frac{1}{2} \left( 1 + ae^2 \right) |w| + |w||1 + ae^2| \right] \right\},$$

$$\frac{\partial f_1}{\partial a} = \sigma(z_1)\sigma(-z_1) \left\{ e^4 \left[ \frac{1}{2} \left( 1 + 2w|w| \right) + w|w|\mathrm{sign} \left( 1 + ae^2 \right) \right] \right\}$$
$$- \sigma(z_2)\sigma(-z_2) \left\{ e^4 \left[ -\frac{1}{2} \left( 1 + 2w|w| \right) + w|w|\mathrm{sign} \left( 1 + ae^2 \right) \right] \right\},$$

$$\frac{\partial f_2}{\partial a} = \sigma(z_2)\sigma(-z_2) \left\{ e^4 \left[ -\frac{1}{2} \left( 1 + 2w|w| \right) + w|w|\mathrm{sign} \left( 1 + ae^2 \right) \right] \right\}.$$

Now we characterize the first set of critical points.

**(i) Stationary points with** $\mathbb{E}_X \left[ f_1 X + f_2 \right] = 0, e = 0$. When $e = 0$, we have that $z_1 = z_2 = b$. Hence,

$$f_1 = \sigma(b) + p + q - 1 - \sigma(b) = p + q - 1,$$
$$f_2 = \sigma(b) - p.$$

Thus, $\mathbb{E}_X\left[f_1 X + f_2\right] = (p+q-1)\,\mathbb{E}_X\left[X\right] + \sigma(b) - p = (p+q-1)\,\pi_1 + \sigma(b) - p = 0$. Rearranging and simplifying, $\sigma(b) = \frac{p}{p+q}$ and hence $b = \log\frac{p}{q}$. Using the fact that $\sigma\left(\log\frac{p}{q}\right) = \frac{p}{p+q} = \pi_1$ and $\sigma\left(-\log\frac{p}{q}\right) = \frac{q}{p+q} = \pi_0$, the above gradients evaluated for any $\gamma = (b = \log\frac{p}{q}, e = 0, w, a)$ further reduce to

$$\left.\frac{\partial f_1}{\partial b}\right|_\gamma = 0, \quad \left.\frac{\partial f_2}{\partial b}\right|_\gamma = \pi_0\pi_1,$$

$$\left.\frac{\partial f_1}{\partial e}\right|_\gamma = 0, \quad \left.\frac{\partial f_2}{\partial e}\right|_\gamma = 0,$$

$$\left.\frac{\partial f_1}{\partial w}\right|_\gamma = 0, \quad \left.\frac{\partial f_2}{\partial w}\right|_\gamma = 0,$$

$$\left.\frac{\partial f_1}{\partial a}\right|_\gamma = 0, \quad \left.\frac{\partial f_2}{\partial a}\right|_\gamma = 0. \tag{92}$$

Now substituting Eq. (92) when computing the second-derivatives of $L$ in Eq. (89), we obtain

$$\left.\frac{\partial^2 L}{\partial b^2}\right|_\gamma = \mathbb{E}_X\left[\left.\frac{\partial f_1}{\partial b}\right|_\gamma X + \left.\frac{\partial f_2}{\partial b}\right|_\gamma\right] = \pi_0\pi_1,$$

$$\left.\frac{\partial^2 L}{\partial b\partial e}\right|_\gamma = \mathbb{E}_X\left[\left.\frac{\partial f_1}{\partial e}\right|_\gamma X + \left.\frac{\partial f_2}{\partial e}\right|_\gamma\right] = 0,$$

$$\left.\frac{\partial^2 L}{\partial b\partial w}\right|_\gamma = \mathbb{E}_X\left[\left.\frac{\partial f_1}{\partial w}\right|_\gamma X + \left.\frac{\partial f_2}{\partial w}\right|_\gamma\right] = 0,$$

$$\left.\frac{\partial^2 L}{\partial b\partial a}\right|_\gamma = \mathbb{E}_X\left[\left.\frac{\partial f_1}{\partial a}\right|_\gamma X + \left.\frac{\partial f_2}{\partial a}\right|_\gamma\right] = 0,$$

$$\left.\frac{\partial^2 L}{\partial e^2}\right|_\gamma = \mathbb{E}_X\left[(f_1 X + f_2)\left(X - \frac{1}{2}\right)\right] 2\left(1 + ae^2\right)\left(1 + 2w|w|\right)\Big|_\gamma$$

$$+ \underbrace{\mathbb{E}_X\left[(f_1 X + f_2)\left(X - \frac{1}{2}\right)\right] 4e^2 a\left(1 + 2w|w|\right)\Big|_\gamma}_{0} \tag{93}$$

$$+ \underbrace{\mathbb{E}_X\left[\left(\left.\frac{\partial f_1}{\partial e}\right|_\gamma X + \left.\frac{\partial f_2}{\partial e}\right|_\gamma\right)\left(X - \frac{1}{2}\right)\right] 2\left(1 + ae^2\right)\left(1 + 2w|w|\right)}_{0}$$

$$= \mathbb{E}_X\left[(2f_1(1 + 2w|w|) - f_1 + 2f_2(1 + 2w|w|)X - f_2\right]\Big|_\gamma$$

$$= \mathbb{E}_X\left[(f_1(1 + 4w|w|) + f_2(2 + 4w|w|)X - f_2\right]\Big|_\gamma$$

$$= (f_1(1 + 4w|w|) + f_2(2 + 4w|w|))\pi_1 - f_2\Big|_\gamma$$

$$\overset{(a)}{=} ((p+q-1)(1 + 4w|w|) - \pi_1(p+q-1)(2 + 4w|w|))\pi_1 + \pi_1(p+q-1)$$

$$= \pi_1(p+q-1)\left(1 + 4w|w| - \pi_1(2 + 4w|w|) + 1\right)$$

$$\overset{(b)}{=} 2\pi_1\pi_0(p+q-1)(1 + 2w|w|),$$

where $(a)$ follows from the fact that $f_1|_\gamma = p+q-1$, $f_2|_\gamma = \sigma(b) - p = \frac{p}{p+q} - p = \frac{-p}{p+q}(p+q-1) = -\pi_1(p+q-1)$ and $(b)$ from $1 - \pi_1 = \pi_0$. Returning to the remaining second derivatives,

$$\frac{\partial^2 L}{\partial e \partial w}\bigg|_{\gamma} = \frac{\partial}{\partial e} \mathbb{E}_X \left[ (f_1 X + f_2) \left( X - \frac{1}{2} \right) \right] 4e^2 \left( 1 + ae^2 \right) |w| \bigg|_{\gamma}$$

$$= \mathbb{E}_X \left[ (f_1 X + f_2) \left( X - \frac{1}{2} \right) \right] \left( 8e \left( 1 + ae^2 \right) |w| + 8e^3 a |w| \right) \bigg|_{\gamma}$$

$$+ \mathbb{E}_X \left[ \left( \frac{\partial f_1}{\partial e} \bigg|_{\gamma} X + \frac{\partial f_2}{\partial e} \bigg|_{\gamma} \right) \left( X - \frac{1}{2} \right) \right] 4e^2 \left( 1 + ae^2 \right) (|w|)$$

$$= 0,$$

$$\frac{\partial^2 L}{\partial e \partial a}\bigg|_{\gamma} = \frac{\partial}{\partial e} \mathbb{E}_X \left[ (f_1 X + f_2) \left( X - \frac{1}{2} \right) \right] e^4 \left( 1 + 2w|w| \right) \bigg|_{\gamma}$$

$$= \mathbb{E}_X \left[ (f_1 X + f_2) \left( X - \frac{1}{2} \right) \right] 4e^3 \left( 1 + 2w|w| \right) \bigg|_{\gamma}$$

$$+ \mathbb{E}_X \left[ \left( \frac{\partial f_1}{\partial e} \bigg|_{\gamma} X + \frac{\partial f_2}{\partial e} \bigg|_{\gamma} \right) \left( X - \frac{1}{2} \right) \right] e^4 \left( 1 + 2w|w| \right)$$

$$= 0,$$

$$\frac{\partial^2 L}{\partial w^2}\bigg|_{\gamma} = \frac{\partial}{\partial w} \mathbb{E}_X \left[ (f_1 X + f_2) \left( X - \frac{1}{2} \right) \right] 4e^2 \left( 1 + ae^2 \right) |w| \bigg|_{\gamma}$$

$$= \mathbb{E}_X \left[ (f_1 X + f_2) \left( X - \frac{1}{2} \right) \right] 4e^2 \left( 1 + ae^2 \right) \operatorname{sign}(w) \tag{94}$$

$$+ \mathbb{E}_X \left[ \left( \frac{\partial f_1}{\partial w} \bigg|_{\gamma} X + \frac{\partial f_2}{\partial w} \bigg|_{\gamma} \right) \left( X - \frac{1}{2} \right) \right] 4e^2 \left( 1 + ae^2 \right) |w|$$

$$= 0,$$

$$\frac{\partial^2 L}{\partial w \partial a}\bigg|_{\gamma} = \frac{\partial}{\partial w} \mathbb{E}_X \left[ (f_1 X + f_2) \left( X - \frac{1}{2} \right) \right] e^4 \left( 1 + 2w|w| \right) \bigg|_{\gamma}$$

$$= \mathbb{E}_X \left[ (f_1 X + f_2) \left( X - \frac{1}{2} \right) \right] 2e^4 |w| \bigg|_{\gamma}$$

$$+ \mathbb{E}_X \left[ \left( \frac{\partial f_1}{\partial w} \bigg|_{\gamma} X + \frac{\partial f_2}{\partial w} \right) \bigg|_{\gamma} \right] e^4 \left( 1 + 2w|w| \right)$$

$$= 0,$$

$$\frac{\partial^2 L}{\partial a^2}\bigg|_{\gamma} = \frac{\partial}{\partial a} \mathbb{E}_X \left[ (f_1 X + f_2) \left( X - \frac{1}{2} \right) \right] e^4 \left( 1 + 2w|w| \right) \bigg|_{\gamma}$$

$$= \mathbb{E}_X \left[ (f_1 X + f_2) \left( X - \frac{1}{2} \right) \right] 2e^4 |w| \bigg|_{\gamma}$$

$$+ \mathbb{E}_X \left[ \left( \frac{\partial f_1}{\partial w} \bigg|_{\gamma} X + \frac{\partial f_2}{\partial w} \right) \bigg|_{\gamma} \right] e^4 \left( 1 + 2w|w| \right)$$

$$= 0.$$

Congregating all the second derivatives from Eq. (93) and Eq. (94) into the Hessian $\boldsymbol{H}(\boldsymbol{\gamma})$ in Eq. (90), we finally obtain

$$\boldsymbol{H}(\boldsymbol{\gamma}) = \pi_0 \pi_1 \begin{bmatrix} 1 & 0 & 0 & 0 \\ 0 & 2(p + q - 1)(1 + 2w|w|) & 0 & 0 \\ 0 & 0 & 0 & 0 \\ 0 & 0 & 0 & 0 \end{bmatrix},$$

which is identical to the Hessian obtained in the proof of Thm. 7 (App. J.2) for $e = 0$. Thus it follows that $\boldsymbol{\Gamma}_{\min}(p, q) \subseteq \mathbb{R}^4$ and $\boldsymbol{\Gamma}_{\mathrm{sad}} \subseteq \mathbb{R}^4$ defined below are a set of local minima and saddle points respectively:

$$\boldsymbol{\Gamma}_{\min}(p, q) \triangleq \left\{ \boldsymbol{\gamma}_{\min} = (e, w, b, a) \in \mathbb{R}^4 : e = 0, (p + q - 1)(1 + 2w|w|) > 0, b = \log \frac{p}{q} \right\},$$

$$\boldsymbol{\Gamma}_{\mathrm{sad}}(p, q) \triangleq \left\{ \boldsymbol{\gamma}_{\mathrm{sad}} = (e, w, b, a) \in \mathbb{R}^4 : e = 0, (p + q - 1)(1 + 2w|w|) \leq 0, b = \log \frac{p}{q} \right\}.$$

Now we focus on the remaining set of critical points.

**(ii) Stationary points with** $\mathbb{E}_X [f_1 X + f_2] = 0, e \neq 0, 1 + ae^2 = 0, 1 + 2w|w| = 0$. For this set of points, the Hessian remains undefined because $\frac{\partial f_1}{\partial e}, \frac{\partial f_2}{\partial e}, \frac{\partial f_1}{\partial a}, \frac{\partial f_2}{\partial a}$ do not exist (Eq. (91)). This non-existence arises since sign $\left(1 + ae^2\right)$ is undefine $1 + ae^2 = 0$. However, even in this scenario, when $e \neq 0, 1 + ae^2 = 0, 1 + 2w|w| = 0$, we have $z_1 = z_2 = b$. Hence,

$$f_1 = \sigma(b) + p + q - 1 - \sigma(b) = p + q - 1,$$
$$f_2 = \sigma(b) - p.$$

Thus the expectation term $\mathbb{E}_X [f_1 X + f_2] = (p + q - 1) \mathbb{E}_X [X] + \sigma(b) - p = (p + q - 1) \pi_1 + \sigma(b) - p = 0$. Simplifying, $\sigma(b) = \frac{p}{p+q}$, which implies $b = \log \frac{p}{q}$.

We could attempt to understand the characterization of the points on this manifold through local perturbation analysis. However, in this work, we classify them as stationary points and leave the comprehensive characterization for future research. This set of points $\boldsymbol{\Gamma}_{\mathrm{station}}(p, q) \subseteq \mathbb{R}^4$ is defined as

$$\boldsymbol{\Gamma}_{\mathrm{station}}(p, q) \triangleq \left\{ \boldsymbol{\gamma}_{\min} = (e, w, b, a) \in \mathbb{R}^4 : e \neq 0, 1 + ae^2 = 0, 1 + 2w|w| = 0, b = \log \frac{p}{q} \right\}.$$

This concludes the proof.

$\square$

### N.3 Proof of Lemma 14

*Proof.* Recall from Lemma 13 that for $\boldsymbol{\theta} = (e, w, a) \in \mathbb{R}^3$, we have

$$\frac{\partial L}{\partial e} = -\mathbb{E}\left[ (f_1 X + f_2)\left( X - \frac{1}{2} \right) \right] \cdot 2e\left(1 + ae^2\right)(1 + 2w|w|)$$

$$- \mathbb{E}\left[ (f_1 X + f_2)\left( X - \frac{1}{2} \right) \right] \cdot 2e^3 a \left(1 + 2w|w|\right),$$

$$\frac{\partial L}{\partial w} = -\mathbb{E}\left[ (f_1 X + f_2)\left( X - \frac{1}{2} \right) \right] \cdot 2e^2 \left(1 + ae^2\right)(|w| + \mathrm{sign}\left(w\right)w),$$

$$\frac{\partial L}{\partial a} = -\mathbb{E}\left[ (f_1 X + f_2)\left( X - \frac{1}{2} \right) \right] \cdot e^4 \left(1 + 2w|w|\right),$$

where $X \in \{0, 1\}$ is a Bernoulli random variable with $X \sim \mathrm{Bern}(p/(p + q))$, and

$$f_1 \triangleq 1 - p - q - \phi_1 + \phi_0, \quad f_2 \triangleq p - \phi_0,$$

$$\phi_1 \triangleq \sigma\left( e^2 \left( \frac{1}{2}\left(1 + ae^2\right)(1 + 2w|w|) + w|w(1 + ae^2)| \right) + b_\star \right),$$

$$\phi_0 \triangleq \sigma\left( e^2 \left( \frac{-1}{2}\left(1 + ae^2\right)(1 + 2w|w|) + w|w(1 + ae^2)| \right) + b_\star \right),$$

where the optimal bias $b_\star$ is obtained by solving $\pi_1 f_1 + f_2 = 0$. Using the definition of the gradient flow that $\dot{\boldsymbol{\theta}} = -\nabla L(\boldsymbol{\theta})$ for $\boldsymbol{\theta} = \boldsymbol{\theta}_t$, we have

$$\dot{w} = -\frac{L(\boldsymbol{\theta})}{\partial w} = \mathbb{E}\left[(f_1 X + f_2)\left(X - \frac{1}{2}\right)\right] \cdot 2e^2\left(1 + ae^2\right)\left(|w| + \text{sign}\left(w\right)w\right)$$

$$\Rightarrow \frac{\dot{w}}{e^2\left(|w| + \text{sign}\left(w\right)w\right)} = \mathbb{E}\left[(f_1 X + f_2)\left(X - \frac{1}{2}\right)\right]2\left(1 + ae^2\right). \tag{95}$$

Similarly for $a$,

$$\dot{a} = -\frac{L(\boldsymbol{\theta})}{\partial a} = \mathbb{E}\left[(f_1 X + f_2)\left(X - \frac{1}{2}\right)\right]e^4\left(1 + 2w|w|\right)$$

$$\Rightarrow \frac{\dot{a}}{e^4} = \mathbb{E}\left[(f_1 X + f_2)\left(X - \frac{1}{2}\right)\right]\left(1 + 2w|w|\right). \tag{96}$$

Likewise, for $e$,

$$\dot{e} = -\frac{L(\boldsymbol{\theta})}{\partial e} = \mathbb{E}\left[(f_1 X + f_2)\left(X - \frac{1}{2}\right)\right]2\left(1 + ae^2\right)e\left(1 + 2w|w|\right)$$

$$+ \mathbb{E}\left[(f_1 X + f_2)\left(X - \frac{1}{2}\right)\right]\left(1 + 2w|w|\right)2e^3 a. \tag{97}$$

By substituting the expressions of Eq. (95) and Eq. (96) into Eq. (97):

$$\dot{e} = \underbrace{\mathbb{E}\left[(f_1 X + f_2)\left(X - \frac{1}{2}\right)\right]2\left(1 + ae^2\right)e\left(1 + 2w|w|\right)}_{Eq.\ (95)}$$

$$+ \underbrace{\mathbb{E}\left[(f_1 X + f_2)\left(X - \frac{1}{2}\right)\right]\left(1 + 2w|w|\right)2e^3 a}_{Eq.\ (96)}. \tag{98}$$

Thus, we obtain

$$\dot{e} = \frac{\dot{w}}{2e^2\left(|w| + \text{sign}\left(w\right)w\right)}e\left(1 + 2w|w|\right) + \frac{\dot{a}}{e^4}2e^3 a. \tag{99}$$

On rearranging and simplifying:

$$e\dot{e} = \frac{\dot{w}}{\left(|w| + \text{sign}\left(w\right)w\right)}\left(1 + 2w|w|\right) + \dot{a}2a$$

$$\Rightarrow e\dot{e} = \frac{\dot{w}}{2\left(|w|\right)}\left(1 + 2w|w|\right) + 2a\dot{a}. \tag{100}$$

Integrating the above equation on both sides:

$$\int e\dot{e} = \int \frac{\dot{w}}{4\left(|w|\right)}\left(1 + 2w|w|\right) + \int 2a\dot{a}$$

$$\Rightarrow \frac{e^2(t)}{2} = \frac{\text{sign}(w(t)) \cdot \log|w(t)| + w(t)^2}{2} + a(t)^2 + \frac{c}{2} \tag{101}$$

$$\Rightarrow e^2(t) = \text{sign}(w(t)) \cdot \log|w(t)| + w(t)^2 + 2a(t)^2 + c.$$

Note that here $c \in \mathbb{R}$, is a constant that depends on the initial conditions. Thus the energy $\mathcal{E}(\boldsymbol{\theta}_t) = \mathcal{E}(\boldsymbol{\theta}_0)$ for $w_0 \neq 0$.

$\square$

### N.4 Proofs of Lemma 16, Lemma 15, and Thm. 3

*Proof.* We note that the proofs of Lemma 16, Lemma 15 directly follow from that of their counterparts Lemma 11 and Lemma 10 using Łojasiewicz's theorem to characterize the convergence of the gradient flow. Thm. 3 is a direct consequence of Lemma 16, Lemma 15. □

### N.5 Informal proof of Thm. 11

*Proof.* [Informal] For $\boldsymbol{\theta} = (e, w, a)$, recall from Lemma 13 that the derivative of the loss $L$ with respect to $e$ is

$$
\begin{aligned}
\frac{\partial L}{\partial e} = \ & \mathbb{E}_X \left[ (f_1 X + f_2) \left( X - \frac{1}{2} \right) \right] 2e \left( 1 + ae^2 \right) \left( 1 + 2w|w| \right) \\
& + \mathbb{E}_X \left[ (f_1 X + f_2) \left( X - \frac{1}{2} \right) \right] 2e^3 a \left( 1 + 2w|w| \right),
\end{aligned}
$$

where

$$
\begin{aligned}
f_1 &= \sigma(z_1) - \sigma(z_2) + p + q - 1, \\
f_2 &= \sigma(z_2) - p, \\
z_1 &\triangleq e^2 \left[ \frac{1}{2} \left( 1 + ae^2 \right) \left( 1 + 2w|w| \right) + w|w(1 + ae^2)| \right] + b, \\
z_2 &\triangleq e^2 \left[ \frac{-1}{2} \left( 1 + ae^2 \right) \left( 1 + 2w|w| \right) + w|w(1 + ae^2)| \right] + b.
\end{aligned}
$$

Assuming the initialization is very small, making any product of quantities in $\boldsymbol{\theta} = (e, w, a, b)$ much smaller than the individual quantities. Therefore, we can consider these products to be approximately zero. That is, $\forall x, y \in \boldsymbol{\theta}, x \geq xy \ \& \ y \geq xy \ \& \ xy \approx 0$. Hence,

$$
\begin{aligned}
z_1 &= b, \\
z_2 &= b, \\
f_1 &= p + q - 1, \\
f_2 &= \sigma(b) - p.
\end{aligned}
$$

Hence the gradient with respect to $e$ is

$$
\frac{\partial L}{\partial e} = 2\mathbb{E}_X \left[ (f_1 X + f_2) (X) \right] e. \tag{102}
$$

Simplifying the expectation term, $\mathbb{E}_X \left[ (f_1 X + f_2) X \right] = (f_1 + f_2)\pi_1 = f_1 \pi_1 - f_1 \pi^2 = (p + q - 1)(\pi_1 - \pi_1^2)$, where we used the fact that $b$ is optimal in the above equations, specifically where $f_1 \pi_1 + f_2 = 0$. Thus the gradient flow for the parameter $e$ is governed by

$$
\dot{e} = -\frac{\partial L}{\partial e} = -(p + q - 1)(\pi_1 - \pi_1^2)e \Rightarrow e = e_0 \exp(-(p + q - 1)(\pi_1 - \pi_1^2)t).
$$

Since $(p + q - 1)(\pi_1 - \pi_1^2) > 0, e \to 0$, which denotes it converges to the local minima.

□

