# OpenReview forum: "Local to Global: Learning Dynamics and Effect of Initialization for Transformers"
_NeurIPS.cc/2024/Conference — NeurIPS 2024 poster_

### Official Review · Reviewer_er5c · 2024-07-09

**Soundness:** 3
**Presentation:** 2
**Contribution:** 3
**Rating:** 5
**Confidence:** 3

**Summary:**

This paper focuses on training a single-layer linear attention transformer with low-rank parameterization on first-order Markov chain data. By reparameterization, they can reduce this problem to a 3-variable learning dynamics and comprehensively characterize the trajectory and local/global minimizers. This paper precisely characterizes the different convergence conditions for local/global convergence, highlighting the role of initialization. Empirically, on this specific data distribution, they verified the superiority of their special initialization compared to standard initialization.

**Strengths:**

This paper is the first result highlighting the role of initialization. Theoretically, they simplify the model to a low-rank parameterization with a fixed subspace for each parameter and rigorously characterize the landscape and trajectory. With this simplified model, all the possible stationary points can be found, and the dynamics can be calculated given different initializations. The simplification is somehow corroborated by the empirical evidence, and the analysis is complicated.

Surprisingly, empirically a random low-rank initialization can outperform the standard initialization on a linear transformer on the Markovian task. It implies the theoretical insights from the low-rank parameterization provide some guidelines for transformer initialization.

**Weaknesses:**

1. I am concerned with the generality of the insights on the role initialization. I appreciate that the authors make a great effort to theoretically analyze the Markov Chain with 2 states (i.e. $P\in R^{2\times 2}$) and do extensive experiments on this particular dataset with linear attention. However, the authors claim that ``we offer practical guidelines for initialization of transformer parameters and demonstrate their effectiveness."(Line 280-281) in both the abstract and the conclusion. Line 255 also claims "the corresponding insights are more general and apply more broadly to the general architecture.''. Since all the conclusions are based on this simplified setting, I don't think it can be characterized as a realistic guide for initialization. I think further experiments on the Markov data with more states and non-linear attention transformers can benefit this paper and make the results more general. I will increase the score if the results can be verified in a more general setting.

2. As a minor point, the dynamics analysis is restricted to a low-rank manifold. Though it is partially corroborated by the experiments that GD finally converges to the low-rank solution, it cannot guarantee that without this restriction, the parameter can stay on the manifold once initialized on the manifold. Nevertheless, the analysis is already complicated and this simplification is acceptable.

**Questions:**

1. Does the low-rank initialization scheme insight transfer to more general data distribution or architecture?

**Limitations:**

The dynamics analysis is restricted to certain subspaces. The attention pattern does not train (which is simplified as a trainable scaling factor $a$). The experiments cannot corroborate the general claim.

---

> ### Author Rebuttal · Authors · 2024-08-05
>
> We thank the reviewer for the helpful feedback and insightful comments. We address the individual questions below:
>
> - **Generality of the insights**: For the two-state Markov chain, while our analysis capitalizes on the canonical parameters and linear attention, our guidelines and insights from this setting readily generalize to non-linear soft-max attention. For instance, Figure 6 in the paper (Figure 1, 3 and 4 in the rebuttal) demonstrates the effectiveness of our initialization scheme for the full one-layer transformer model with soft-max attention (Lines 60-64). Similarly, the low-rank structure of the parameters, as shown in Figure 5 (Figure 2 in the rebuttal), is concerning the full general model. The lines 280-281 and 255 were meant to convey this message which we are happy to rewrite to make it more clear.
>
>      For multi-state Markov chains, we observe a similar phenomenon where, under standard Gaussian initialization, a single-layer transformer can converge to either local or global minima depending on the Markovian switching.  Specifically, in Figure 5 of the rebuttal, we consider a first-order Markov chain with state space $S = \\{ 0, 1, 2, 3, 4 \\} $ where any state $s \\in S$ stays in the same state with probability $1-p$ and switches uniformly random to either of the remaining states with probability $p/4$. This generalizes the binary-state Markov chain in the paper with $ p = q$. As illustrated in Figure 5 of the rebuttal, depending on the values of the switching probability $p$, the transformer parameters either converge to local-minima (unigram) or global minima (bi-gram). Further, Figure 6 of the rebuttal also reveals the low-rank structure of the parameters during training for the multi-state setting too. These interesting empirical observations highlight an exciting research opportunity to mathematically characterize the gradient dynamics for the multi-state Markov chains, similar to our two state analysis.
>
> - **Low rank manifold**: Indeed, using canonical parameterization to characterize the gradient descent dynamics of transformers is a standard tool in the literature. For instance, a recent work [1] employs a similar approach to show that two-layer transformers learn induction heads, assuming specific structure for the attention matrices in both layers, inspired by empirical observations (Figure 2, [1]).
>
> - **Limitations**: Please note that the attention parameter $a$ is also trained alongside the embedding and weight parameters $(e,w)$, and the corresponding analysis is highlighted in Section 4.1. All the figures in the paper reflect this setting, with no parameters being frozen or omitted.
> ----------------
>
> **References**
>
> [1] How Transformers Learn Causal Structure with Gradient Descent? https://arxiv.org/pdf/2402.14735

---

> > ### Comment · Reviewer_er5c · 2024-08-10
> >
> > Thanks for the detailed response. After careful consideration, I will maintain my score. Below, I provide specific reasons for each of the points.
> >
> > **Generality of the insights**: Regarding the softmax attention for $P\in \mathbb{R}^{2\times 2}$ mentioned in the paper, I believe that the initialization scheme in linear attention prevents the softmax transformers from converging to local minima. However, the exact architecture used in the experiments should be explicitly stated in the main paper. For the multi-state Markov Chain setting, since I was requested to look only at the first page, the current content does not sufficiently demonstrate general results that would illustrate the low-rank structure of the final minimizer or the effectiveness of your low-rank initialization. For completeness, I suggest including experiments that explore whether your low-rank initialization helps transformers not converge to local minima in multi-state Markov Chain tasks.
> >
> > **Low-rank manifold**:  I fully understand the difficulty of analyzing transformer dynamics, and I agree that reparameterizing the transformer to simplify the analysis is a reasonable approach. However, I must note that restricting the analysis to the ground-truth low-rank manifold with only a trainable scalar factor $a$ may be an oversimplification. Compared with the related works, [1] and other linear regression ICL papers only limit the trainable parameter by zeroing out some irrelevant parts of the matrices. They can then prove the gradient can help learn the correct ground truth, instead of limiting the feature on the ground-truth direction. I believe that if the low-rank parameters in the attention layer were trained (initialized or parameterized in a low-rank manner, but not necessarily aligned with $\alpha$), it would significantly enhance the contribution of this paper. It may be too challenging to analyze with so many trainable parameters, so the authors are free to consider or disregard this suggestion.
> >
> > **Limitations**: I have revised my previous inaccurate comment. My primary concern is that the attention features of the transformers are not learned, with the only trainable parameter being the scalar $a$. Specifically, the attention score $\texttt{attn}_{n, i}$ is not trained.

---

> ### Author Response · Authors · 2024-08-12
> **Attention features are also learnt. Insights for linear-attention generalize to soft-max. Low-rank vectors turn out not be relevant.**
>
> We thank the reviewer for interesting questions and apologize for not being able to accommodate the requested figures in the first page of the PDF. We address the concerns below.
>
> **Attention features are indeed learnt:** Note that with canonical parameterization, the attention score is given by $\texttt{attn}_{n,i} = a \cdot (x_n - \frac{1}{2}) (x_i - \frac{1}{2})$ (Eq. 45 in the Appendix). Since $a$ is a trainable parameter, the attention scores are also learned.
>
> **Linear vs. soft-max attention**: We are afraid that there has been a misunderstanding about our intialization and the transformer architecture. For all the experiments in the paper, we use the full general 1-layer transformer with non-linear soft-max attention, as described in lines 63-64 of the paper. For the theoretical analysis, we use the linear attention mechanism but with rest of the architecture intact. Using these insights derived from the linear setting, we demonstrate that our initialization scheme performs significantly better than standard Gaussian initialization even on the full soft-max model. This is not due to linear vs. soft-max but rather the actual loss landscape as illustrated in Figure 4 in the paper and theoretically characterized in Theorem 9 of Appendix G. Hence we respectully disagree with the claim that "...our intialization scheme in linear attention prevents the softmax transformers from converging to local minima...".
>
> **We do not limit low-rank parameters:** Please note that the low-rank parameters in the attention layer are not artificially restricted to be same as $\alpha$. The reason why they are also parameterized as $\alpha$ is because empirical evidence suggests that the transformer parameters always converge to this specific structure. Furthermore, this $\alpha$ turns out to be all $\pm 1$ vector. Hence, even after parameterizing $\alpha$ as a trainable vector in our setting, it turns out that the final loss function $L$ (in line 203) is independent of $\alpha$  (since $\|\alpha \|$ is constant, cf. Appendix B) and only depends on three scalar parameters $(e,w,a)$. Thus without loss of generality, it suffices to analyze these scalars.

---

> ### Comment · Reviewer_er5c · 2024-08-12
>
> **Attention features are indeed learned:** The definition of $a$ in the paper is somewhat confusing, as there are two different definitions provided. Between lines 111-112, $a=\langle v, \alpha\rangle$ is stated, while on line 200, $a=q^2d^{5/2}/4\langle v, \alpha\rangle$ is given. I believe the definition on lines 111-112 is a typo, but it seems to suggest that only the $V$ layer is being trained. This would imply that $\texttt{attn}$ is not learned and that the trainable parameter $a$ comes from $v$. It is a minor issue, since it is equivalent to put this scalar with $V$ or $\texttt{attn}$.
>
> **Linear vs. Softmax Attention:** I am puzzled by the authors' disagreement with my claim. My statement was intended to convey that "your initialization scheme, inspired by the linear model, indeed helps generalize to softmax attention and prevents convergence to local minima." But my concern remains: does this low-rank initialization (not necessarily rank-2) generalize to multi-state Markov models? Even within the extended rebuttal PDF, I don't see this experiment included. Furthermore, according to Figure 5, it is not strictly low-rank (and definitely not rank-1); the strict low-rank property only holds for the binary-state case in the PDF. I suggest that the authors include these details in future versions, and hopefully, all previous insights will generalize to the multi-state settings.
>
> **We do not limit low-rank parameters:** I fully understand the authors need to make these simplification. But I respectively disagree with the claim that "even after parameterizing $\alpha$ as a trainable vector in our setting, the loss L is constant", since the after training the norm of $\alpha$ will change. Also, it does not necessarily true that $x_n, W_Q,W_K$ all align with $\alpha$ after you train $\alpha$ with some random initialization. And the authors **limit** low-rank parameters, just not artificially. It is natural and common to simplify the network as a theoretical work but it is indeed a limitation.
>
> I will maintain my score and cannot strongly recommend acceptance.

---

> > ### Author Response · Authors · 2024-08-12
> >
> > Dear Reviewer er5c,
> >
> > Thanks for your prompt response. We address the individual comments below.
> >
> > **Attention features:** As you already noted, indeed the attention scalar $a$ captures the essence of both the attention mechanism and the value matrix and hence can be thought of as a trainable parameter that takes both of them into account, as also implied by its definition in Line 200. Sorry for the confusion through the typo in Line 112. Here we just wanted to motivate the definition of $a$ without getting into too much notation-heavy details. We will correct this.
> >
> > **Linear vs. soft-max attention:** Sorry if we misunderstood your claim. Our earlier disagreement was with the initialization scheme and not regarding the multi-state Markov chains. As the Figure 6 in the rebuttal suggests, while it's true that the matrices are not exactly rank-1, nonetheless they still converge to a relativley low-rank corresponding to rank-4. And as also highlighted in Figure 5 of the rebuttal, the input Markovian switching $p$ has an effect on model converging to local or global minima, albeit with a threshold different than $0.5$, which is the threshold for the binary case. This is not surpising given that this corresponds to multi-state setting. Together these observations suggest that while similar phenomena as in binary-setting happen also in multi-state setting, this warrants a more detailed study and corresponding gradient flow analysis. This is considerably more challenging than the binary case, but nonetheless an important and exciting venue of future research and currently outside the scope of our current paper. We will include a discussion and corresponding results for multi-state case in the revised version to provide more context to our work.
> >
> > **Low-rank parameters:** We agree with your assessment. We initialized all the low-rank vectors to be a single $\alpha$ because that's what happens in practice at convergence (one could also experimentally check how fast this happens from the beginning). But we agree that analyzing the full dynamics with different vectors is indeed an interesting direction of future research though considerably more challenging, as already revealed in our simplified setting.

---

### Official Review · Reviewer_Fte8 · 2024-07-12

**Soundness:** 3
**Presentation:** 4
**Contribution:** 2
**Rating:** 6
**Confidence:** 3

**Summary:**

The paper seeks to characterize how single layer transformers learn (two symbol) markov chains. The analysis relies on a reparametrization of the transformer that assumes the weight matrices are rank 1. The primary results are characterizing when gradient flow on this reparameterized transformer leads to global minimums in loss or other critical points, showing how in some cases a standard gaussian initialization will lead to a local minimum instead of the global minimum.

**Strengths:**

The paper is generally well written and readable, with well explained theory (in the main body at least, I have not checked the appendix proofs carefully) and high quality visualizations. I like the step by step presentation of the architecture.

Theoretically analyzing the training dynamics and initializations of transformers is definitely significant, and currently somewhat rare to see even in toy settings such as this one.

Post Rebuttal: I believe the paper is technically very solid, with my main concern still being the contribution. I did not raise my confidence only because I am less confident that the contribution quite reaches "moderate to high", but I am more confident (4/5) in the other aspects of the paper.

**Weaknesses:**

1. The canonical parameterization is a rather strong assumption (reducing the model to just three parameters) and so it needs strong results to justify it. For me, the results in this paper are close to strong enough, but some strengthening of them would be appreciated (short of removing assumptions).

It would be nice more evidence showing how the analysis on the canonical parameterization could extend to more general settings. Ideally, theoretical analyses of toy systems should be paired with empirical and heuristic evidence that they either a) approximately explain the dynamics of more realistic systems, or b) yield new insights that can be empirically verified on more realistic systems.

I believe that the paper is primarily trying to show the first claim (a), with the experiments showing the transformers initialized with rank 1 weights stay rank 1, but I think this can benefit from some additional evidence. For example, it would be great to show that the learning dynamics of the canonical parameterization are similar to that of a single layer transformer (with softmax attention and no constant parameters) with a rank one initialization.

2. Figure 6 might be trying to address the second claim (b), but I think it can be improved. Having W_1 and W_2 be constant implies that “our initialization” is more of a reparametrization than an initialization, making the comparison not feel apples to apples.

3. For figures 5 and 7, while I agree that qualitatively the matrices look rank 1, some quantitative measure of rank (like nuclear norm) would be good to graph (especially with error bars for multiple runs).

4. While it is mentioned that experiments were repeated 5 times, it might be nice to put into the paper some of these results, Figure 6 stood out as a place where it would be good and not too difficult to add error bars or additional runs. Additionally, details of how replications were done would be good to mention (such as how random seeds were chosen). The code as of now is a bit messy, and could do with some reorganizing and removing unnecessary files (like untitled.ipynb).

**Questions:**

1. How is figure 6 testing the hypothesis that “for p + q > 1, any small initialization around zero would lead to a local minima” (line 247). I feel like it's more of comparing a small initialization to your parametrization (which is still a worthy endeavor). I’d imagine that testing the stated hypothesis would require trying many different small initializations and showing that they lead to local minima (Given the nature of the problem, I imagine it could be feasible to test initializations in a grid and create something like an empirical analogue to corroborate figure 1d?). Also, a natural follow-up question (or null hypothesis) is can some standard initialization that isn’t small and/or isn’t around zero work just as well as your initialization.

2. Also, it took me a bit to understand the definition of the attention matrix (attn) (along with Wq and Wk, they seem to only only be mentioned in section 4.1 in the main body, though I may have missed something). Are they abstracted away from earlier in the paper for simplicities sake, or some other reason?

**Limitations:**

The discussion of limitations in the conclusion section (which I mentioned specifically because of the response given in the checklist, lines 1200-1201) is a bit too brief. It would be good to have all of the assumptions made for the main theorems in one place, especially those implicit and not stated in the theorem statements, including the rank one and related assumptions, linear attention, and gradient flow (which is in the theorem statements, but good to mention because transformers are not normally trained with gradient flow).

---

> ### Author Rebuttal · Authors · 2024-08-05
>
> We thank the reviewer for the helpful feedback and suggestions to improve the paper and the code which we are planning to incorporate in the revised version. We refer to the global response regarding our results across repeated trials for various $(p,q)$ and the measure of low-rankness. We address the individual questions below.
>
>
> - **Our insights carry over to the full general model**: While our theoretical analysis uses canonical parameterization, our insights and guidelines extend to the general single-layer transformer with soft-max attention (Lines 60-64). For example, Figure 6 demonstrates the effectiveness of our initialization scheme for the full one-layer transformer model. Similarly, the low-rank structure of the parameters, shown in Figure 5 in the paper (Figure 2 and 6 in the rebuttal PDF), pertains to this general model.
>
> -  **Canonical parameters exhibit the same dynamics as the full model**: We strongly believe that the training dynamics of the canonical model closely resemble those of a full single-layer transformer with rank-1 initialization. Due to the time constraints, we currently lack empirical evidence for this. However, we will share this comparison during the discussion phase.
>
> - **$W_1$ and $W_2$ are not constants**: We are afraid that there has been a misunderstanding about the initialization in Figure 6. While $W_1$ and $W_2$ are both initialized constant, they are trained alongside all other parameters  $\theta$ in the single-layer transformer with non-linear soft-max attention (Lines 60-64). Thus, Figure 6 of the paper (Figure 1, 3 and 4 in the rebuttal PDF) effectively demonstrates the success of our initialization scheme and the broad applicability of our insights.
>
> - **Q.1**: In Figure 6 in the paper, our standard initialization scheme is to let $e_0=1$ and $w_0 = -1 $ which falls in the region $ \mathcal{I}_\ast $ (see Figure 1d) and converges to the global minimum as predicted by our theory. To test our hypothesis on a wider range of initializations, we try the following set of initializations which concur with our theoretical predictions (Figure 3 of the rebuttal): we let $(e_0,w_0) \in \mathcal{N}(\mu, 0.02) $ with $ \mu \in \{ (-1,-1), (0, -0.5), (0,0.5), (0, 0)  \} $, with $ p=0.5$ and $ q=0.8$. As highlighted in Figure 3 of the rebuttal, we observe that the initializations around  $ (-1,-1)$ leads to global minimum convergence whereas the rest  converge to local minima, as predicted by Figure 1d in the paper.
>
> - **Q.2**: Yes indeed. We wanted to highlight the main ideas behind our gradient-flow analysis using canonical parameterization and hence chose to omit the attention parameters in the beginning for the ease of exposition and clarity.
>
> -----------------------------------------

---

> > ### Comment · Reviewer_Fte8 · 2024-08-08
> > **Question**
> >
> > I was wondering (after seeing figure 6 in the rebuttal pdf), if theres any chance your initialization scheme help in the multi state markov chain setting (I know that this might be beyond the scope of this work, but I am curious)?
> >
> > I also wanted to ask if you plan for the discussion of limitations in the conclusion to be any different in the camera ready version than the current version?

---

> > > ### Author Response · Authors · 2024-08-09
> > > **Limitation section and new figures**
> > >
> > > Dear reviewer,
> > >
> > > - **Canonical vs. low-rank**: Thanks for your prompt response and thought provoking questions. In our earlier response, due to time constraints , we couldn't attach the figure showcasing the similar training profiles of canonical and low-rank model but we now have the corresponding empirical results and the figure that illustrate this phenomenon. We are currently checking with the ACs about how best to share these results as the official rules forbid us from sharing links. We'll keep you updated.
> > >
> > > - **Multi-state**: In this setting as well, there would exist analogous good and bad initialization regions as highlighted for the binary case in Figs. 1 and 4 in our paper. While empirically we can verify if our scheme still works here as is, however we have to rederive and precisely determine the corresponding basins of attraction for the local and global minima in the multi-state setting separately. Also the Markovian switching condition $p+q>1$ would also have to correspondigly modified to accommodate more states. We already see a glimpse of this in Figure 5 of the rebuttal for five-state Markov chain: $p=0.6$ still converges to global minima unlike the binary-state setting where $p+q = 0.6 + 0.6 >1$ would have led to local minima convergence.
> > >
> > > - **Limitations and assumptions**: Indeed, in our revised version we will add two separate subsections. The first one outliining all the corresponding assumptions and the setting for our results. Then the second discusses the limitations of our results and potential future directions to address them.
> > >
> > > Once again, we would like to thank the reviewer for their constructive efforts to improve the paper which we really appreciate. We are wondering if they are willing to change their score if their concerns are addressed. We remain at your disposal for any further questions.

---

> ### Author Response · Authors · 2024-08-12
> **Discussion period ending soon**
>
> Dear Reviewer Fte8,
>
> We sincerely appreciate the time you have taken to provide valuable feedback for our work. As we are getting closer to the end of the discussion period, could you let us know if our responses above have adequately addressed your concerns? We remain at your disposal for any further questions.
>
> If you agree that our responses to your reviews have addressed the concerns you listed, we kindly ask that you consider whether raising your score would more accurately reflect your updated evaluation of our paper. Thank you again for your time and thoughtful comments!
>
> Sincerely,
> The Authors

---

> > ### Comment · Reviewer_Fte8 · 2024-08-14
> >
> > Sorry for the delay, I have updated my score.

---

### Official Review · Reviewer_JG7D · 2024-07-13

**Soundness:** 4
**Presentation:** 4
**Contribution:** 3
**Rating:** 6
**Confidence:** 3

**Summary:**

The paper investigates how transformers learn first-order Markov chains, focusing on the role of parameter initialization and providing a comprehensive analysis of the learning dynamics. It proves that transformer parameters can converge to global or local minima based on initialization and data properties, and supports these theoretical findings with empirical evidence.

**Strengths:**

1. This paper studies the training dynamics of transformers in learning Markov chains and study the effects of initialization of parameters, which is a meaningful direction which have not been fully explored.
2. The paper is well-structured and well-written, offering clear explanations and visualizations that enhance comprehension.
3. The empirical studies are well-designed and the results convincingly support the theoretical findings.

**Weaknesses:**

1. The paper examines first-order Markov chains with a rank-1 input sequence and considers single-layer linear attention. This general setting might be overly simplified.
2. Can the initialization guidelines provided be extended to more complex transformer models, such as those with additional layers?
3. While the empirical results support the findings, it is not intuitively obvious why the optimal parameters exhibit low rank. Could you offer a more detailed explanation?
4. A minor suggestion: the formulation $att_{n,i}= . . .$ should be included in the main body of the paper rather than just in the appendix, as attention is a crucial component of the transformer structure.

**Questions:**

Please see the weakness.

**Limitations:**

Yes.

---

> ### Author Rebuttal · Authors · 2024-08-04
>
> We thank the reviewer for the helpful feedback and insightful comments. We will update the revised version with information about the attention in the main text. We address the individual questions below.
>
> - **Rank-1 input sequence……:** Please note that the input is a first-order Markov chain, without the assumption of it being rank-one. We use linear attention for ease of theoretical analysis, but our guidelines and insights generalize to non-linear soft-max attention. For instance, Figure 6 in the paper demonstrates the effectiveness of our initialization scheme for the full one-layer transformer model (Lines 60-64). Similarly, the low-rank structure of the parameters, as shown in Figure 5 in the paper (Figure 2 and 6 in the rebuttal PDF), applies to the full general model.
>
> - **Initialization guidelines**: As our analysis shows, understanding the theoretical foundations of transformer training and initialization, even for shallow models, is challenging. Our paper, focusing on shallow transformers, is a crucial step toward optimizing deeper models. Recent research [1] employs a similar reparameterization technique to study gradient descent dynamics in a simplified two-layer attention-only transformer, elucidating the induction head mechanism. However, unlike our work, they do not explore how initialization impacts convergence to local or global minima. Given the nascent and evolving understanding of deeper transformers and higher-order Markov chains [1,2], exploring similar initialization guidelines for large-scale models is an exciting avenue for future research.
>
> - **Low rank parameters**: Recent evidence [3,4] suggests that SGD with weight decay inherently leads to rank minimization, resulting in low-rank parameters at convergence. Although these studies focus on feed-forward neural networks, we believe a similar phenomenon occurs in transformers as well..
>
> -------------------------------------------------------------------------------------------------------------------------
> **References**
> - [1] How Transformers Learn Causal Structure with Gradient Descent? https://arxiv.org/pdf/2402.14735
> - [2] Transformers on Markov Data: Constant Depth Suffices, https://arxiv.org/abs/2407.17686v1
> - [3] Characterizing the Implicit Bias of Regularized SGD in Rank Minimization, https://arxiv.org/pdf/2206.05794
> - [4] Rank Diminishing in Deep Neural Networks: https://openreview.net/forum?id=tIqzLFf3kk

---

> > ### Comment · Reviewer_JG7D · 2024-08-12
> > **Thank you for the rebuttal.**
> >
> > Thank you for the reply. Some of my concerns have been addressed. Hence, I will keep my original score.

---

### Official Review · Reviewer_DieV · 2024-07-14

**Soundness:** 3
**Presentation:** 2
**Contribution:** 3
**Rating:** 5
**Confidence:** 3

**Summary:**

This paper investigates the learning dynamics of transformer models, specifically focusing on first-order Markov chains and single-layer transformers. The authors aim to understand how transformers learn Markov chains and the impact of initialization on their training outcomes. They provide a comprehensive theoretical analysis, demonstrating that transformer parameters trained on next-token prediction loss can converge to either global or local minima, depending on the initialization and properties of the Markovian data. The paper also offers empirical evidence to support their theoretical findings and proposes guidelines for initializing transformer parameters effectively.

**Strengths:**

-  The paper provides a detailed theoretical framework that explains the conditions under which transformer parameters converge to global or local minima, filling a gap in the understanding of transformer learning dynamics.
- The theoretical findings are supported by empirical experiments, which strengthens the credibility and applicability of the results.
- The authors highlight the critical role of initialization in the training process and offer practical guidelines for initializing transformer parameters, which can be valuable for practitioners looking to optimize transformer training.

**Weaknesses:**

- One important weakness of this study is its limited scope. The study focuses on single-layer transformers and first-order Markov chains, which may limit the generalizability of the findings to more complex models and data structures.
- While thorough, the theoretical analysis may be difficult for practitioners without a strong mathematical background to fully grasp and apply. Maybe the presentation can be changed and made more intuitive to also target the audience who does not have a strong mathematical background.
- The empirical validation could benefit from a broader range of experiments, including different types of datasets and more complex transformer architectures.

**Questions:**

The paper does not address the scalability of the proposed initialization guidelines for large-scale transformer models, which are commonly used in practice. Can the authors provide an insight on this?

**Limitations:**

As I mentioned in weaknesses section, the most important limitation of this study is its generalizability. The study focuses on single-layer transformers and first-order Markov chains, and it is unknown if this method is generalizable to more complex models and data structures.

---

> ### Author Rebuttal · Authors · 2024-08-04
>
> We thank the reviewer for the constructive feedback and helpful suggestions to improve the paper. We will add a separate section with prerequisite background details in the revised version. We refer to the common response for experiments on multi-state Markov chains. We address the individual concerns below.
>
> **Scalability**: Understanding the theoretical foundations of transformer training and initialization, even for shallow models, remains a challenging task, which our work aims to address. While this paper focuses on shallow transformers, it represents a crucial step toward understanding optimization in deeper models. For instance, recent research [1] employs a similar reparameterization technique to study gradient descent dynamics in a simplified two-layer attention-only transformer, elucidating how they learn the induction head mechanism. However, unlike our work, they do not investigate how initialization impacts convergence to local or global minima. Given the nascent and evolving understanding of deeper transformers and higher-order Markov chains [1,2], exploring similar initialization guidelines for large-scale models is an exciting avenue for future research.
>
> ---------------------------------------------------------------------------
>
> **References**
>
> - [1] How Transformers Learn Causal Structure with Gradient Descent? https://arxiv.org/pdf/2402.14735
> - [2] Transformers on Markov Data: Constant Depth Suffices, https://arxiv.org/abs/2407.17686v1

---

> > ### Author Response · Authors · 2024-08-12
> > **Discussion period ending soon: call for action**
> >
> > Dear Reviewer DieV,
> >
> > We sincerely appreciate the time you have taken to provide valuable feedback for our work. As we are getting closer to the end of the discussion period, could you let us know if our responses above have adequately addressed your concerns? We remain at your disposal for any further questions.
> >
> > Sincerely,
> > The Authors

---

> > > ### Comment · Reviewer_DieV · 2024-08-12
> > >
> > > Dear Authors, I carefully read your responses and the other reviewers' reviews. I agree to the reviewer er5c on their comments on the generality of the insights. Additionally, my concerns regarding generalizability and the scalability of the proposed method sustain. That's why I will maintain my score.

---

> ### Author Response · Authors · 2024-08-12
> **Generality of the insights**
>
> Dear Reviewer DieV,
>
> Thanks for the insightful questions and keeping the score. We are afraid that there has been a misunderstanding about the generality of the insights, which we addressed in the global response and also individually to the Reviewer er5c just now. We are reposting them here for the sake of completeness.
>
> **Generality of the insights:**  For all the experiments in the paper, we use the full general 1-layer transformer with non-linear soft-max attention, as described in lines 63-64 of the paper. For the theoretical analysis, we use the linear attention mechanism but with rest of the architecture intact. Using these insights derived from the linear setting, we demonstrate that our initialization scheme performs significantly better than standard Gaussian initialization even on the full soft-max model. Since the same soft-max attention used for both schemes empirically, our superiority is not due to linear vs. soft-max but rather due to capitalizing on the actual loss landscape as illustrated in Figure 4 in the paper and theoretically characterized in Theorem 9 of Appendix G. Hence we respectully disagree with the Reviewer er5c's claim that "...our intialization scheme in linear attention prevents the softmax transformers from converging to local minima...".
>
> **Attention features are indeed learnt:** Note that with canonical parameterization, the attention score is given by $\texttt{attn}_{n,i} = a \cdot (x_n - \frac{1}{2}) (x_i - \frac{1}{2})$ (Eq. 45 in the Appendix). Since $a$ is a trainable parameter, the attention scores are also learned.
>
>
> **We do not limit low-rank parameters:** Please note that the low-rank parameters in the attention layer are not artificially restricted to be same as $\alpha$. The reason why they are also parameterized as $\alpha$ is because empirical evidence suggests that the transformer parameters always converge to this specific structure. Furthermore, this $\alpha$ turns out to be all $\pm 1$ vector. Hence, even after parameterizing $\alpha$ as a trainable vector in our setting, it turns out that the final loss function $L$ (in line 203) is independent of $\alpha$  (since $\|\alpha \|$ is constant, cf. Appendix B) and only depends on three scalar parameters $(e,w,a)$. Thus without loss of generality, it suffices to analyze these scalars.

---

### Official Review · Reviewer_LVoR · 2024-07-25

**Soundness:** 4
**Presentation:** 3
**Contribution:** 3
**Rating:** 6
**Confidence:** 3

**Summary:**

The paper theoretically studies the effect of initialization on the gradient dynamics of a single-layer transformer trained on Markov data. It considers a simplified model by: (1) using a binary input alphabet and (2) reducing the many parameters of the single-layer transformer to just two or three scalar parameters. The authors identify the critical points of this reduced set of parameters and compare them to the stable point under its gradient flow dynamics. This is used to determine whether an initialization will converge to global or local optima. This informs choices of initialization that will reliably converge to global optima. This is validated in a single-layer transformer where the reduction of weights is not performed. This experiment is also used to validate the reductions made for theoretical analysis.

**Strengths:**

This paper studies the dynamics of a simplified transformer model under next-token prediction loss. This extends previous work, which just studied the static landscape of the loss. In its study of dynamics, this paper also newly considers questions about initialization and its role in convergence toward global minima.

The paper is written clearly and generally supports its claims, highlighting the questions asked and key contributions.

**Weaknesses:**

It is not clear how much results vary across repeated trials. The authors state that they repeated experiments 3 to 5 times, but the results across replications are not presented in the figures or the text. For instance, in Figure 6, did every repeated experiment follow the same blue and red curves? Similarly, while the authors show the weights from a single run on Figure 5, they do not provide a numerical assessment of the validity of their low-rank approximations. Another weakness is the simplified data model, which is just a binary alphabet. The authors seem to inherit this from previous analytical work, but a discussion of how this restricts results and real-world applicability is missing.

**Questions:**

1. How much do results vary across replications? The authors could easily answer this by showing different runs in the same figure or stating measures of uncertainty. If possible, the authors could also consider more values of p and q.
2. How reliable is the low-rank approximation? It would be beneficial to quantify the low-rank structure of W1 and Wv so that it is easy to understand how consistent this approximation is in general.
3. Do similar empirical results hold with larger alphabets? More specifically, do similar low-rank approximations still hold, and do these initializations still work better?
4. The authors remark that they observe $a \approx 0$, so the attention term is often not relevant. Is this a consequence of the simplified data model? It would be nice to comment more on this observation.
5. Figures 2 and 3 should be subfigures.

**Limitations:**

The authors are fair in the scope of their claims, though it would be good to emphasize the use of reduction/reparameterization earlier in the paper or abstract (this is not made clear until the end of page 3, and this reduces the theoretical scope of their claims). Alternatively, the authors could provide stronger evidence to justify the universality of these approximations (precise quantification of low-rank structure, and more replications).

---

> ### Author Rebuttal · Authors · 2024-08-05
>
> We thank the reviewer for the helpful feedback and insightful comments. We refer to the global response regarding our results across repeated trials for various $(p,q)$ and the measure of low-rankness. We address the individual concerns below.
>
> - **Binary and large alphabet**: This paper primarily focuses on the binary alphabet to analytically characterize the phenomena reported in [1], specific to the binary setting. However, we observe a similar low-rank structure for transformer parameters in a first-order Markov chain on a larger alphabet (see Figure 6 in the rebuttal PDF). Consistent with the binary setting, parameters initialized as low-rank remain low-rank throughout training, as shown in Figure 6 .
> Theoretically, while our gradient-flow analysis can be generalized to the multi-alphabet setting, it is challenging to precisely characterize the phase portraits, as in Figure 2 of the paper, due to the increased number of parameters. This presents an intriguing avenue for future research.
>
> - **$ a \approx 0 $**: This results from the fact that for first-order Markov chains, the Markov kernel $ \mathbb{P}(x\_{n+1} = 1  \mid x_1^n) = x_n (1-p-q) + p $ is a simple linear function of the last symbol x_n​. Consequently, this function can be represented by a single-layer transformer using only the skip connection in the attention layer, without relying on the attention mechanism. Therefore, $ a \approx 0 $  becomes a viable solution, which we leverage for the gradient-flow analysis without attention. Figure 7 in the rebuttal empirically corrobarates this fact. As a side note, there are also non-zero attention coefficients that represent global minima, as shown in Figure 4 and discussed in Section 4.1 of the paper.
> ----------------------------------------------------------------
> **References:**
>
> [1] Attention with Markov: A Framework for Principled Analysis of Transformers via Markov Chains: https://arxiv.org/abs/2402.04161

---

> ### Comment · Reviewer_LVoR · 2024-08-14
>
> Dear Authors,
>
> Thank you for your responses to the other reviewers and myself, and for responding
>
> __Low-rank structure__: Thank you for simulating Markov chains with more data. Looking at Figure 6 (despite exceeding the page limit), it appears to be the case that the rank of $W_1$ and $W_V$ is |S| - 1, based on the binary and 5-state examples given (I assume Figure 6 used the same alphabet size as Figure 5, but this does not appear to have been explicitly stated). This further seems to imply that for more plausible settings, where alphabets are very large, we should initialize the weight matrices to be full rank. While I am not as concerned as other reviewers about the analytical simplifications of reducing assuming parameters to be low-rank at initialization for the sake of tractability, I find it hard to see how the implications, as presented in section 5, are very impactful.
>
> Nevertheless, I appreciate that the paper is technically strong and precise. If the assumptions and limitations are made clearer throughout the text (which the authors have promised to do), I believe this paper could be a nice, small contribution towards analyzing transformers, but I cannot presently strongly recommend acceptance. So, I maintain my score.

---

### Author Rebuttal · Authors · 2024-08-06

## **Generality of the insights: Our insights and conclusions hold for all pairs of $(p,q)$, across repeated trials, and for non-linear soft-max attention**

We thank the reviewers for the constructive feedback. We address the common concerns here regarding the error bars across repeated trials, experimental results for various $p$ and $q$, a metric for the low-rankness, and linear vs. soft-max attention.

### **Error bars**

In the paper, we reported results averaged across three to five trials but omitted the error bars due to an oversight. The rebuttal PDF includes all figures with error bars, confirming that our conclusions remain valid. For instance, Figures 1, 3 and 4 in the rebuttal shows that our initialization scheme consistently converges to a global minimum across multiple trials while standard Gaussian initialization gets stuck at the local minima.

### **Experimental results for various $(p,q)$**

We would like to emphasize that our theoretical results hold for  all values of $(p,q)$ with the regions $p+q <1 $ and $p+q > 1$ exhibiting fundamentally different behaviors as highlighted in respective Theorems 7 and 2, and Figure 1. For the empirical results in the paper, especially for Figure 6, we chose a single pair $(p,q)= (0.5, 0.8)$ in the region $p+q>1$ for the ease of exposition only. Complementing this result, we further empirically demonstrate in Figure 4 of the rebuttal that our initialization scheme consistently outperforms the standard Gaussian one for various pairs of $(p,q) \in  \{ (0.6, 0.6), (0.7, 0.9)   \} $ all satisfying $p+q>1$.

### **Low-rank metric**

Indeed, complementing our visual illustration of the low-rank matrices in the paper, we can also mathematically quantify this property via the following metric: we compute the relative energy contained in their top singular value compared to their overall singular values, i.e. $ \frac{\sigma\_1^2}{\sum_{i} \sigma_i^2 }  $. The closer this fraction is to one, the closer the corresponding matrix is to being rank-one. Tracking this metric training for various matrices, Figure 2 in the rebuttal highlights that while random initialization training eventually reaches rank-one parameters, initializing at rank-one always retains that structure during training. This figure corresponds to the same setting as Figure 5 in the manuscript.

### **Linear vs. soft-max attention, canonical parameters vs. full-transformer**

We are afraid that there has been a misunderstanding about the setting in the paper regarding linear and soft-max attention. For the two-state Markov chain, while our theoretical analysis capitalizes on the canonical parameters and linear attention, our guidelines and insights from this setting readily generalize to the full single-layer transformer with non-linear soft-max attention (Lines 60-64). For instance, Figure 6 in the paper (Figure 1 in the rebuttal) demonstrates the effectiveness of our initialization scheme for the full one-layer transformer model with soft-max attention across various values of $(p,q)$ and repeated trials. Similarly, the low-rank structure of the parameters, as shown in Figure 2 and 6 of the rebuttal (Figure 5 in the paper), pertains to the full general model.

---

> ### Comment · Area_Chair_Pe5t · 2024-08-10
> **Overlong PDF**
>
> The PDF in this global response is longer than the 1-page limit. Reviewers are requested to look only at the first page per the review policy.

---

> ### Author Response · Authors · 2024-08-11
> **Apologies**
>
> Dear Area Chair Pe5t,
>
> We apologize for overflowing the 1-page figure limit. Unfortunately, despite our best efforts we couldn't contain the experimental results requested by reviewers in less than a page.
>
> Once again, we apologize for the inconvenience.
>
> Regards,
> The Authors

---

### Author Response · Authors · 2024-08-14
**Summary of review and discussion**

We thank all the reviewers for their constructive feedback, engaging discussion during the rebuttal, and valuable time. We are pleased that all reviewers found our paper to be well-written, insightful, clear, and precise, with solid theoretical contributions supported by robust experimental results and visualizations. We are also delighted that they found our initialization analysis to be insightful and impactful (_reviewers DieV and er5c_)

The following list summarizes the additions that we will make to the final vision of our manuscript. These additions are based on the comments and questions of the reviewers during the official review, and are already made available in our rebuttal.

---------------------------------------------------------------------------------------------------------------------------------
1. **Assumptions and generality of the insights:**  We believe there maybe a misunderstanding about the generality of the insights obtained through our analysis in the current paper. In the modified version, we will include a separate section highlighting the specific setting used for our theoretical analysis (_canonical_) and the architecture used for the experiments (_full general model_). We will also include additional experiments for all $(p,q)$ highlighting the generality of our results. We thank all the reviewers for this suggestion.

2. **Error-bars and low-rank metric:** We will modify all the existing curves averaged over runs with the shaded error bars, as shared in the global response below. We will also quantify and better illustrate the low-rankness of matrices using the low-rank metric we discussed in the rebuttal. We thank reviewers LVoR and Fte8 for these suggestions.

3. **Multi-state Markov chains:** We will add an additional section about multi-state Markov chains discussing how one could obtain similar initialization guidelines in this scenario. In addition to the empirical results highlighting the role of Markovian switching and the number of states  on the loss landscape, as inlcuded in the rebuttal PDF, this entails a clear theoretical analysis of the loss landscape first and the associated learning dynamics. We thank reviewers LVoR and er5c for the suggestion.

4. **Limitations and scalability:** Being one of the first works to precisely characterzie the role of intialization in learning dynamics of transformers, our work opens new doors for similar analyses for deeper architectures and multi-state settings. We will add a separate section outlining the current limitations of our approach and outline some important research directions to extend to more complicated settings. We thank reviewers LVoR, DieV, and JG7D for this suggestion.

---

### Decision · Program_Chairs · 2024-09-25

**Decision:**

Accept (poster)

**Comment:**

This submission presents a theoretical analysis of the learning dynamics of a simple transformer architecture on a simple Markov chain data model. Reviewers praised the technical correctness of the work and the clarity. Most concerns from reviewers are about the relevance of assumptions for more practical settings. However, these assumptions have been clearly explained in the submission and responses, and theoretical work in this area is difficult and needed, so I recommend acceptance.